# SHIELD: Multi-task Multi-distribution Vehicle Routing Solver with Sparsity and Hierarchy

Yong Liang Goh [1 2 3 4]   Zhiguang Cao [5]   Yining Ma [6]   Jianan Zhou [7]   Mohammed Haroon Dupty [1]   Wee Sun Lee [1]

## Abstract

Recent advances toward foundation models for routing problems have shown great potential of a unified deep model for various VRP variants. However, they overlook the complex real-world customer distributions. In this work, we advance the Multi-Task VRP (MTVRP) setting to the more realistic yet challenging Multi-Task Multi-Distribution VRP (MTMDVRP) setting, and introduce SHIELD, a novel model that leverages both *sparsity* and *hierarchy* principles. Building on a deeper decoder architecture, we first incorporate the Mixture-of-Depths (MoD) technique to enforce sparsity. This improves both efficiency and generalization by allowing the model to dynamically select nodes to use or skip each decoder layer, providing the needed capacity to adaptively allocate computation for learning the task/distribution specific and shared representations. We also develop a context-based clustering layer that exploits the presence of hierarchical structures in the problems to produce better local representations. These two designs inductively bias the network to identify key features that are common across tasks and distributions, leading to significantly improved generalization on unseen ones. Our empirical results demonstrate the superiority of our approach over existing methods on 9 real-world maps with 16 VRP variants each.

[1]School of Computing, National University of Singapore, Singapore [2]Institute of Data Science, National University of Singapore, Singapore [3]Grabtaxi Holdings Pte Ltd, Singapore [4]Grab-NUS AI Lab, Singapore [5]School of Computing and Information Systems, Singapore Management University, Singapore [6]Laboratory for Information and Decision Systems, Massachusetts Institute of Technology, Cambridge MA, United States [7]College of Computing and Data Science, Nanyang Technological University, Singapore. Correspondence to: Yong Liang Goh <gyl@u.nus.edu>.

*Proceedings of the 42$^{nd}$ International Conference on Machine Learning*, Vancouver, Canada. PMLR 267, 2025. Copyright 2025 by the author(s).

## 1. Introduction

Combinatorial optimization problems (COPs) appear in many real-world applications, such as logistics (Cattaruzza et al., 2017) and DNA sequencing (Caserta & Voß, 2014), and have historically attracted significant attention (Bengio et al., 2021). A key example of COPs is the Vehicle Routing Problem (VRP), which asks: *Given a set of customers, what is the optimal set of routes for a fleet of vehicles to minimize overall costs while satisfying all constraints?* Traditionally, they are solved with exact or approximate solvers. However, these solvers rely heavily on expert-designed heuristic rules which limit its efficiency. Recently, the emerging Neural Combinatorial Optimization (NCO) community has been increasingly focused on developing novel neural solvers for VRPs based on deep (reinforcement) learning (Kool et al., 2018; Kwon et al., 2020; Bogyrbayeva et al., 2024). These solvers learn to construct solutions autoregressively, improving efficiency and reducing the need for domain knowledge.

Motivated by the recent breakthroughs in foundation models (Floridi & Chiriatti, 2020; Touvron et al., 2023; Achiam et al., 2023), a notable trend in the NCO community is the push towards developing a unified neural solver for handling multiple VRP variants, known as the Multi-Task VRP (MTVRP) setting (Liu et al., 2024; Zhou et al., 2024; Berto et al., 2024). These solvers are trained on multiple VRP variants and show impressive zero-shot generalization to new tasks. Compared to single-task solvers, unified solvers offer a key advantage: there is no longer a need to construct different solvers or heuristics for each specific problem variant. However, despite the importance of the MTVRP setup, it does not fully capture real-world industrial applications, as the underlying distributions are assumed to be uniform, lacking the structural properties of real-world data.

This work extends the MTVRP framework to real-world scenarios by incorporating realistic distributions (Goh et al., 2024). Consider a logistics company operating across multiple cities/countries, each with a fixed set of $M$ locations governed by its geographical layout. When a subset of $V$ orders arises, the problem is reduced to serving only those customers. To model this, we generate realistic distributions by selecting smaller subsets of $V$ from the fixed set of $M$ locations such that the geographical characteristics

of $M$ are retained. This transforms MTVRP into the Multi-Task Multi-Distribution VRP (MTMDVRP), a novel and challenging setting that, to our knowledge, has not been explored in the literature.

Nevertheless, MTMDVRP poses unique challenges for learning unified neural VRP models. First, beyond managing the diverse constraints of MTVRP, the model must further learn to handle arbitrary, distribution-specific layouts. Unfortunately, task-related contexts are often interdependent with distribution-related contexts during decision-making (e.g., selecting the next node), adding further complexity. Meanwhile, beyond traditional cross-distribution setups, our approach samples instances from an underlying distribution that captures more practical, real-world patterns. To perform well in the MTMDVRP setting, the model must capture both task-specific and distribution-related contexts when selecting the next node. One promising way to achieve this is to enable the model to dynamically process nodes, allowing it to allocate computational focus to the most critical nodes. Additionally, to be generalizable, the model must be sufficiently regularized to prevent over-fitting.

To this end, we introduce **S**parsity & **H**ierarchy **in E**fficiently **L**ayered **D**ecoder (SHIELD) to address the above challenges with two key innovations. First, SHIELD leverages *sparsity* by incorporating a customized Mixture-of-Depths (MoD) approach (Raposo et al., 2024) to the NCO decoders. While adding more decoder layers can improve predictive power, the autoregressive nature of neural VRP solver significantly hampers efficiency. In contrast, our MoD is designed to dynamically adjust the proper computational depth (number of decoder layers) based on the decision context. This allows it to adaptively allocate computation for learning the task/distribution specific and shared representations while acting as a regularization mechanism to prevent over-fitting by possibly reducing redundant computations. Secondly, we employ a clustering mechanism that considers *hierarchy* during node selection by forcing the learning of a small set of key representations of unvisited nodes, enabling *sparse* and compact modelling of the complex decision-making information. Together, these two designs encourage the model to learn compact, simple, generalizable representations by limiting computational budgets, effectively enhancing generalization across tasks and distributions. This paper highlights the following contributions:

- We propose Multi-Task Multi-Distribution VRP (MT-MDVRP), a novel, more realistic, yet challenging scenario that better represents the real-world industry.
- We present SHIELD, a neural solver that leverages *sparsity* through a customized NCO decoder with MoD layers and *hierarchy* through context-based cluster representation. Both contributions reduce computation and parameters, acting as effective regularizers, thereby

leading to a more generalizable neural VRP solver.

- We demonstrate SHIELD's impressive in-distribution and generalization benefits via extensive experiments across 9 real-world maps and 16 VRP variants, achieving state-of-the-art performance compared to existing unified neural VRP solvers.

## 2. Preliminaries

**CVRP and its Variants.** The CVRP is defined as an instance of $N$ nodes in a graph $\mathcal{G} = \{\mathcal{V}, \mathcal{E}\}$, where the depot node is denoted as $v_0$, customer nodes are denoted as $\{v_i\}_{i=1}^N \in \mathcal{V}$, and edges are defined as $e(v_i, v_j) \in \mathcal{E}$ between nodes $v_i$ and $v_j$ such that $i \neq j$. Every customer node has a demand $\delta_i$, and each vehicle has a maximum capacity limit $Q$. For a given problem, the final solution (tour) can be presented as a sequence of nodes with multiple sub-tours. Each sub-tour represents a vehicle's path, starting and ending at the depot. As a vehicle visits a customer node, the demand is fulfilled and subtracted from the vehicle's capacity. A solution is considered feasible if each customer node is visited exactly once, and the total demand in a sub-tour does not exceed the capacity limit of the vehicle. In this paper, we consider the nodes defined in Euclidean space within a unit square $[0, 1]$, and the overall cost of a solution, $c(\cdot)$, is calculated via the total Euclidean distance of all sub-tours. The objective is to find the optimal tour $\tau^*$ such that the cost is minimized, given by $\tau^* = \operatorname{argmin}_{\tau \in \Phi} c(\tau|\mathcal{G})$ where $\Phi$ defines the set of all possible solutions.

We define the following practical constraints that are integrated with CVRP: (1) *Open route (O)*: The vehicle is no longer required to return to the depot after visiting the customers; (2) *Backhaul (B)*: The demand on some nodes can be negative, indicating that goods are loaded into the vehicle. Practically, this mimics the pick-up scenario. Nodes with positive demand $\delta_i > 0$ are known as linehauls, and nodes with negative demand $\delta_i < 0$ are known as backhauls. Routes can have a mixed sequence of linehauls and backhauls without strict precedence; (3) *Duration Limit (L)*: Each sub-tour is upper bounded by a threshold limit on the total length; (4) *Time Window (TW)*: Each node $v_i$ is defined with a time window $[w_i^o, w_i^c]$, signifying the open and close times of the window, and $s_i$ the service time at a node. A customer can only be served if the vehicle arrives within the time window, and the total time taken at the node is the service time. If a vehicle arrives earlier, it has to wait until $w_i^o$. All vehicles have to return to the depot before $w_0^c$.

**Neural Constructive Solvers.** Neural constructive solvers are typically parameterized by a neural network, where a policy, $\pi_\theta$, is trained by reinforcement learning to construct a solution sequentially (Kool et al., 2018; Kwon et al., 2020). Generally, these solvers employ an encoder-decoder architecture and are trained as sequence-to-sequence mod-

els (Sutskever, 2014). The probability of a sequence can be factorized as $p_\theta(\tau|\mathcal{G}) = \prod_{t=1}^{T} p_\theta(\tau_t|\mathcal{G}, \tau_{1:t-1})$. The encoder stacks multiple transformer layers to extract node embeddings, while the decoder generates solutions autoregressively using a contextual embedding $\mathbf{h}_{(c)}$. To decide on the next node, the attention mechanism produces attention scores used for decision-making (Vaswani, 2017). The contextual vectors $\mathbf{h}_{(c)}$ serves as query vectors $\mathbf{Q}$, while the keys, $\mathbf{K}$, is the set of $N$ node embeddings. This is mathematically represented as

$$a_j = \begin{cases} U \cdot \text{TANH}(\frac{\mathbf{Q}\mathbf{K}^\top}{\sqrt{\text{DIM}}}) & j \neq \tau_{t'}, \forall t' < t \\ -\infty & \text{otherwise} \end{cases} \quad (1)$$

where $U$ is a clipping function and DIM the dimension of the latent vector. These attention scores are then normalized using a softmax function to generate the probability distribution: $p_i = p_\theta(\tau_t = i|s, \tau_{1:t-1}) = \frac{e^{a_j}}{\sum_j e^{a_j}}$. Invalid moves, such as previously visited nodes, are managed using a mask during this process. Finally, given a baseline function $b(\cdot)$, the policy is trained with the REINFORCE algorithm (Williams, 1992) and gradient ascent, with the expected return $J$ and the reward of each solution $R$ (i.e., the negative length of the solution tour): $\nabla_\theta J(\theta) \approx \mathbb{E}\left[(R(\tau^i) - b^i(s))\nabla_\theta \log p_\theta(\tau^i|s)\right]$. We leave additional details about the architecture in Appendix C.

**Mixture-of-Experts.** Previous work (Liu et al., 2024) demonstrated the ability of state-of-the-art transformers such as POMO (Kwon et al., 2020) to generalize across MTVRP instances. More recently, Zhou et al. improved upon this architecture using Mixture-of-Experts (MoE). An MoE layer consists of $m$ experts $\{E_1, E_2, ..., E_m\}$, whereby each expert is a feed-forward MLP. A gating network $G$ produces a scalar score based on an input token $x$, which decides how the inputs are distributed to the experts. The layer's output can be defined as $\text{MOE}(x) = \sum_{j=1}^{m} G(x)_j E_j(x)$. The gating network selects the top-k experts to prevent computation from exploding. For MV-MoE, MoE layers are inserted in each transformer block, allowing each token to use $k$ experts. Additionally, a hierarchical gate is introduced in the decoder at the problem level to learn whether or not to use experts at each decoding step.

# 3. Methodology

## 3.1. MTVRP and MTMDVRP Setup

Formally, the optimization objective of an MTVRP instance is given by

$$\min(c(X)) = \mathbb{E}_{k \sim \mathcal{K}}\left[\sum_{s \in \mathcal{S}} \sum_{p_i \in s} d(p_i, p_{i+1})\right] \quad (2)$$

where $\mathcal{K}$ is the set of all tasks, $\mathcal{S}$ the set of all sub-tours in an instance, $p_i$ the $i$-th node in the sequence of $s$, and $d(\cdot, \cdot)$ the Euclidean distance function. For the MTMDVRP in this paper, we expand on the MTVRP scenarios in (Liu et al., 2024; Zhou et al., 2024). The $x_i$ and $y_i$ coordinates for the instances are now sampled from a known underlying distribution of points. This enables the samples to mimic most of the structural distributions and patterns available in the problem. The optimization objective is now given by

$$\min(c(X)) = \mathbb{E}_{q \sim \mathcal{Q}}\left[\mathbb{E}_{k \sim \mathcal{K}}\left[\sum_{s \in \mathcal{S}} \sum_{p_i \in s} d(p_i, p_{i+1})\right]\right] \quad (3)$$

where $\mathcal{Q}$ is the set of all distributions. The following practical scenario can visualize our MTMDVRP: assume a logistics company X deploys a deep learning model to solve multiple known variants for its current business. In an ideal world, it would have access to all forms of logistics problems generated across all possible structured distributions in the world, whereby a country map $q \in \mathcal{Q}$. Realistically, company X only has historical data in some tasks and presence in a handful of countries, such that $q' \in \mathcal{Q}'$, whereby $\mathcal{Q}' \subset \mathcal{Q}$, meaning that it only has data drawn from a subset of distributions in $\mathcal{Q}$. Likewise, it has only faced a subset of tasks such that $k' \in \mathcal{K}', \mathcal{K}' \subset \mathcal{K}$. Based on this historical data, company X can train a single model using $\mathcal{Q}'$ and $\mathcal{K}'$. Now, if company X wishes to expand its presence to other parts of the world, it would see new data samples from new distributions and meet new tasks that were not present in the training set. Thus, it would be highly beneficial for company X to be able to apply its model readily. To do so, the model has to be robust to the task and distribution deviation simultaneously, suggesting strong generalization properties across these two aspects.

**Challenges of MTMDVRP.** While adding distributions may seem straightforward, it introduces significant complexity. *The model must learn representations that capture task and distribution contexts when selecting the next node to visit.* Unfortunately, these are often interdependent, which complicates decision-making. For example, in a skewed map such as EG7146 in Figure 4 of Appendix S, the task complexity is closely tied to the geographic layout. The depot's position significantly impacts the solution; a depot near clustered customer nodes is less complex to solve than one located in a sparse region with distant customer nodes. Balancing shared and task/distribution-specific representations is more complicated, as the model must generalize across a broader space to be useful across tasks and distributions.

For our setup, we adopt the following feature set. At each epoch, we are faced with a problem instance $i$ such that $\mathcal{S}_i = \{x_i, y_i, \delta_i, w_i^o, w_i^c\}$, where $x_i$ and $y_i$ are the respective coordinates, $\delta_i$ the demand, $w_i^o$ and $w_i^c$ the respective opening and closing times of the time window. This is passed

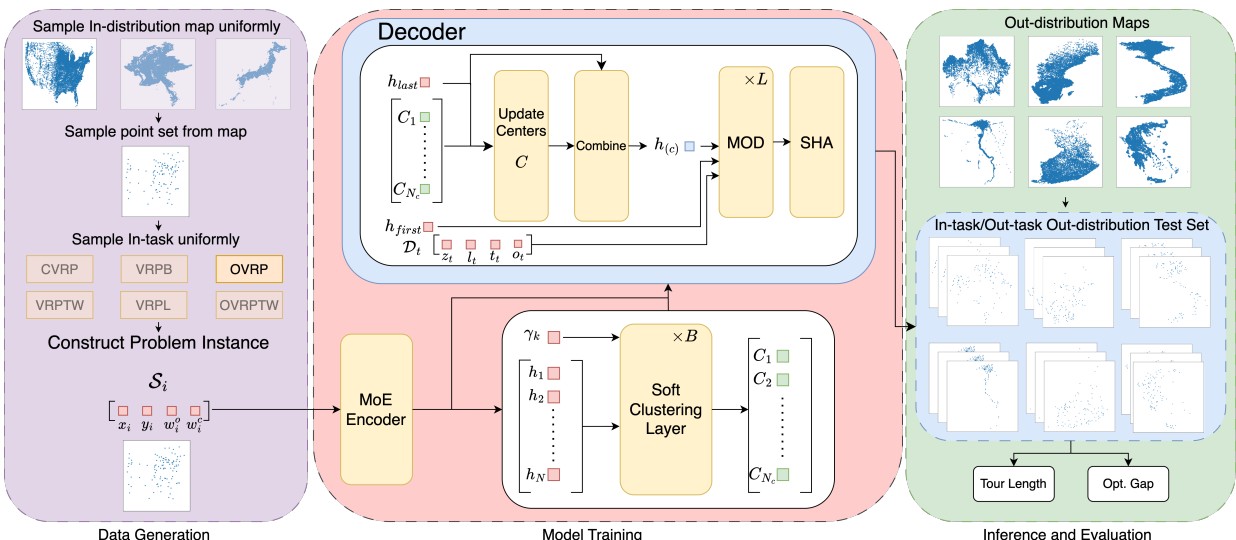

*Figure 1.* Overall proposed approach for MTMDVRP. First, in-distribution maps are sampled uniformly, and a set of points is sampled from the map. Next, a task is sampled uniformly from the in-task set. These form a batch of problem instances and are passed to the network. SHIELD encompasses an MoE encoder, a context-based clustering layer, and the MoD decoder. The decoder is applied autoregressively to in-task/out-task out-distribution instances where the optimality gap is calculated using known solvers.

through the encoder, resulting in a set $\mathbf{H}$ of $d$-dimensional embeddings. At the $t$-th decoding step, the decoder receives this set of embeddings $\mathbf{H}$, the clustering embeddings $\mathbf{C}$, and a set of dynamic features $\mathcal{D}_t = \{z_t, l_t, t_t, o_t\}$, where $z_t$ denotes the remaining capacity of the vehicle, $l_t$ the length of the current partial route, $t_t$ the current time step, and $o_t$ indicates if the route is an open route or not.

### 3.2. Regularization by compute and generalization

To further address the generalization aspect of foundation models for NCO, we present the perspective of adaptive computing motivated by the Vapnik-Chervonenkis (VC) dimension concept. The VC dimension is a traditional analysis in statistical learning that aims to quantify the complexity of an algorithm (e.g. a neural network) and its learning capacity. In particular, a high VC-dim indicates a more complex model, allowing for greater capacity for representation at the expense of greater sample complexity and a higher tendency for over-fitting. Likewise, a low VC-dim indicates a simpler model, suggesting inadequate representation power or possibly more substantial generalization due to its simplicity.

**Theorem.** *Let $\{\mathcal{C}_{k,n} : k, n \in \mathbb{N}\}$ be a set of concept classes where the test of membership of an instance $c$ in a concept $C$ consists of an algorithm $\mathcal{A}_{k,n}$ taking $k + n$ real inputs representing $C$ and $c$, whose runtime is $t = t(k, n)$, and which returns the truth value $c \in C$. The algorithm $\mathcal{A}_{k,n}$ is allowed to perform conditional jumps (conditioned on equality and inequality of real values) and execute the standard arithmetic operations on real numbers $(+, -, \times, /)$ in constant time. Then VC-dim$(\mathcal{C}_{k,n}) = O(kt)$.*

The above theorem (taken from Theorem 2.3 (Goldberg & Jerrum, 1993)) shows that for algorithms consisting of multivariate polynomials, such as neural nets, the VC-dim of the algorithm $\mathcal{A}_{k,n}$, where $k$ is the number of parameters and $n$ the number of input features, is polynomial in terms of its compute runtime $t$ and number parameters, giving us a complexity of $O(kt)$. While the Theorem is not strictly applicable to networks containing exponential functions, it suggests that the amount of compute can potentially serve as a regularizer. *Based on these observations, we hypothesize that one can alter the generalization performance of a neural network by adjusting the number of parameters and the total computation used (and hence its runtime).*

We propose an adaptive learning approach that regulates the complexity of the network as an appropriate architecture for generalization. Our customized MoD approach enforces *sparsity* through learning reduced network depths and lighter computation per token. We regularize the model to learn generalizable representations across tasks/distributions by constraining the network's total compute. Additionally, a clustering mechanism forces the network to condense information. By limiting the number of parameters (and hence the number of clusters) to a handful, we enforce *sparsity* the mechanism. In a Multi-Task Multi-Distribution scenario, we posit that these encourage the network to efficiently generalize by balancing the computational budget for task-specific information while leaving common information to be learned across other tasks or distributions, allowing for

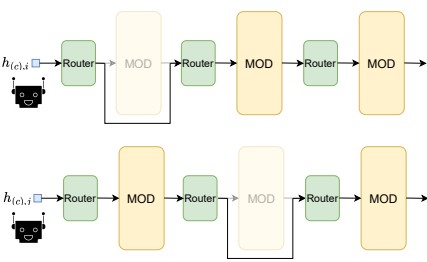

*Figure 2.* Token is routed differently for each agent depending on the router.

efficient generalization across tasks and distributions.

### 3.3. Going deeper but sparser

Our proposed architecture is shown in Figure 1. To increase the predictive power of the MVMoE, one can easily hypothesize that increasing the number of parameters would be necessary. However, due to the autoregressive nature of decoding, this quickly becomes computationally expensive. Instead, we propose the integration of the Mixture-of-Depths (MoD) (Raposo et al., 2024) approach into the decoder. Given a dense transformer layer and $N$ tokens, MoD selects the top $\beta$-th percentile of tokens to pass through the transformer layer. In contrast, the remaining unselected tokens are routed around the layer with a residual connection around the layers, avoiding the need to compute all $N$ attentional scores. Formally, the layer can be represented as

$$\mathbf{h}_i^{l+1} = \begin{cases} r_i^l f_i(\tilde{\mathbf{H}}^l) + \mathbf{h}_i^l & \text{if } r_i^l > P_\beta(\mathbf{r}^l) \\ \mathbf{h}_i^l & \text{if } r_i^l < P_\beta(\mathbf{r}^l) \end{cases} \quad (4)$$

where $r_i = \mathbf{W}_\theta^\top \mathbf{h}_i^l$ is router score given for token $i$ at layer $l$, $W_\theta$ is learnable parameters in the router that converts a $d$-dimensional embedding into a scalar score, $\mathbf{r}^l$ the set of all router scores at layer $l$, $P_\beta(\mathbf{r}^l)$ the $\beta$-th percentile of router scores, and $\tilde{\mathbf{H}}$ the subset of tokens in the $\beta$-th percentile. In this work, we apply token-level routing on contextual vectors $\mathbf{h}_{(c)}$, whereby each token is passed through the router, and the top $\beta$-th percentile tokens are selected and form the query embeddings. Each transformer layer still receives all $N$ node embeddings that serve as key and value embeddings, and a mask to determine whether a node has been visited. By controlling $\beta$, we control the sparsity of the architecture by limiting the total number of query tokens that are processed. This means that the network must learn to identify which *current locations* are more important to be processed, as shown in Figure 2.

### 3.4. Contextual clustering

Apart from sparsity in compute, we introduce hierarchy and sparsity in the form of representation. Goh et al. (2024) first showed that one can apply a form of soft-clustering

to summarize the set of unvisited cities into a handful of representations. This is then used to guide agents, providing crucial information about the groups of nodes left in the problem, which is highly useful for structured distributions.

In addition to structured distributions, the MTMDVRP has underlying commonalities among its tasks. As such, we hypothesize that nodes and their associated task features can be grouped. While spatial structure can typically be measured in Euclidean space, it is not so straightforward for tasks and its features. Thus, an EM-inspired soft clustering algorithm in latent space provides a sensible approach to this problem. We first define a set of $\mathbf{C} \in \mathrm{R}^{N_c \times d}$ representations, such that $N_c$ of these denote the number of cluster centers. The soft clustering algorithm poses the forward pass of the attention layer as an estimation of the E-step, and the re-estimation of $\mathbf{C}$ using the weighted sum of the learnt attention weights as the M-step. Repeated passes through this layer simulate a roll-out of a pseudo-EM algorithm. Effectively, the network learns to transform the initial cluster centroids into the final centroid embeddings.

In this work, we introduce context prompts to capture the task dependencies for the soft clustering algorithm. Ideally, for the same spatial graph, if the task at hand is different, the clustering mechanism should be sufficiently flexible to accommodate the various intricacies of the task. Prompts are a reasonable approach, as they provide helpful task information for LLMs (Radford et al., 2019). Specifically, we construct contextual prompts as latent representations given by $\alpha_k = \mathbf{W}_\theta^\top \gamma_k$ where $\mathbf{W}_\theta$ is a set of learnable parameters that transform the constraints to latent representations, and $\gamma_k$ is a one-hot encoded vector of constraints for task $k$, such that each feature corresponds to a constraint. In this work, we have $\gamma_k = [\gamma_k^1, \gamma_k^2, \gamma_k^3, \gamma_k^4]$, where $\gamma_k^1$ denotes *open*, $\gamma_k^2$ denotes *time-window*, $\gamma_k^3$ denotes *route length*, and $\gamma_k^4$ denotes *backhaul* constraints. By designing prompts to operate in the latent space, we thus enable the model to learn to stitch together these constraints, allowing for flexible modeling of tasks that it has not seen during training. Now, this vector is passed onto the clustering layer:

$$\hat{\mathbf{h}}_i = \mathbf{W}_H \mathbf{h}_i, \hat{\mathbf{c}}_j = \mathbf{W}_C[\mathbf{c}_j, \alpha_d], \quad (5)$$

$$\psi_{i,j} = \text{SOFTMAX}(\frac{\hat{\mathbf{h}}_i \hat{\mathbf{c}}_j^\top}{\sqrt{\text{DIM}}}), \mathbf{c}_j = \sum_i \psi_{i,j} \mathbf{h}_i \quad (6)$$

whereby $\mathbf{W}_H$ and $\mathbf{W}_C$ are weight matrices, $[\cdot]$ denotes the concatenation operation, $\Psi$ the set of all mixing coefficients $\psi_{i,j}$, $\hat{\mathbf{c}}_j$ the learnable initial cluster center representation, $\hat{\mathbf{h}}_i$ the input node embeddings, and $\mathbf{c}_j$ the final cluster representation as a weighted sum of input embeddings after multiple passes. Essentially, Equation 5 is repeated $B$-times. The overall process can be viewed in Algorithm 1 in Appendix D. The output of these cluster centroids is fed to the decoder and serves as additional information for the decoding pro-

cess. At each step, we update clusters by taking a weighted subtraction of visited nodes, given by

$$\mathbf{h}_{(c)} = W_{\text{COMBINE}}[\mathbf{h}_{\text{LAST}}^L, \mathbf{c}_1, \mathbf{c}_2, ..., \mathbf{c}_{N_c}] + \mathbf{h}_{\text{FIRST}}^L, \quad (7)$$

$$\mathbf{c}_j' = \mathbf{c}_j - (\psi_{i,j} * \mathbf{h}_i), \forall j \in N_c \quad (8)$$

## 4. Experiments

We conform to a similar problem setup in (Liu et al., 2024; Zhou et al., 2024), using a total of 16 VRP variants with 5 constraints, as described in section 2. All experiments run on a single A100-80Gb GPU.

**Datasets.** We utilize nine country maps[1]: USA13509, JA9847, BMM33708, KZ9976, SW24978, EG7146, FI10639, GR9882. Dataset details are in Appendix E.

**Task Setups.** We define the following: (1) *in-task* refers to the six tasks that the models are trained on: CVRP, OVRP, VRPB, VRPL, VRPTW, OVRPTW; (2) *out-task* refers to the ten tasks that the models are not trained on: OVRPB, OVRPL, VRPBL, VRPBTW, VRPLTW, OVRPBL, OVRPBTW, OVRPLTW, VRPBLTW, OVRP-BLTW; (3) *in-distribution* refers to the three distributions that the models observe during training: USA13509, JA9847, BM33708; (4) *out-distribution* refers to the six distributions that the models do not observe during training: KZ9976, SW24978, VM22775, EG7146, FI10639, GR9882.

**Neural Constructive Solvers.** We compare the following unified solvers focused on generalization: (1) POMO-MTVRP which applies POMO to the MTVRP setting (Liu et al., 2024); (2) MVMoE that extends POMO to include MoE layers (Zhou et al., 2024); (3) MVMoE-Light, a variant of MVMoE with an additional hierarchical gate in the decoder (Zhou et al., 2024); (4) MVMoE-Deeper whereby we increase the depth of MVMoE to have the same number of layers in the decoder as SHIELD so that both models have similar capacity; (5) SHIELD-MoD where we train our model only with MoD layers and without the clustering; (6) SHIELD, our proposed model of MoD and clustering.

**Hyperparameters.** We use the ADAM optimizer to train all neural solvers from scratch on $20,000$ instances per epoch for $1,000$ epochs. All models plateau at this epoch, and the relative rankings do not change with further training. At each training epoch, we sample a country from the in-distribution set, followed by a subset of points from the distribution and a problem from the in-task set, as shown in Figure 1. For SHIELD, we use 3 MoD layers in the decoder and only allow 10% of tokens per layer. The number of clusters is set to $N_c = 5$, with $B = 5$ iterations of soft clustering. The encoder consists of 6 MoE layers. We

provide full details of the hyperparameters in Appendix I.

**Performance Metrics.** We sample 1,000 test examples per problem for each country map and solve them using traditional solvers. We use HGS (Vidal, 2022) for CVRP and VRPTW instances and Google's OR-tools routing solver (Furnon & Perron) for the rest. For neural solving, each sample is augmented 8 times following Kwon et al. (2020), and we report the tour length and optimality gap (compared to the traditional solver) of the best solution found across these augmentations, whereby smaller values indicate better performance. We provide details of solver settings, augmentation, and optimality gap in Appendix H.

### 4.1. Empirical Results

**Main Results.** Table 1 presents the average tour length (Obj) and optimality gap (Gap) across the respective tasks (in-task/out-task) and distributions (in-dist/out-dist), with details in Tables 14 to 22. SHIELD demonstrates significantly stronger predictive capabilities and outperforms all other neural solvers across all tasks and distributions.

We can view MVMoE-Deeper as a model that processes each token heavily with multiple layers, while MVMoE is a model that processes each token only once. SHIELD is thus a middle point that learns how to adapt the processing according to the token and problem state. Consequently, this suggests that overprocessing (MVMoE-Deeper) and underprocessing (MVMoE) nodes can be problematic in building an efficient foundation model. As shown, increasing the depth of the decoder to MVMoE-Deeper improves its overall performance, especially in the in-task in-distribution case. Unfortunately, the autoregressive nature quickly renders the model untrainable on MTMDVRP100. Instead, if we replace these dense layers with sparse ones (as in SHIELD), the model is now trainable on larger problems and sees significant improvement in task and distribution generalization. These aspects also highlight the positive effects of regularization by reducing compute and parameters.

Table 1 also highlights the positive effect of contextual clustering, particularly in problems with 100 nodes. The benefits are most evident in the model's generalization across tasks and distributions. Summarizing the larger set of points helps the model identify key points in route construction.

**Model Complexities.** Table 4 in Appendix F displays each model's total number of parameters. To quantify complexity, we measure the average number of floating operations (FLOPs) for a single-pass through the encoder and one decoding step. Note that we use only one decoding step as inferior neural solvers will require more steps to solve the problem and thus increase its overall compute budget. As shown, MVMoE has an increased number of FLOPs compared to the original POMO-MTVRP. For our model, both

---

[1] https://www.math.uwaterloo.ca/tsp/world/countries.html

*Table 1.* Overall performance of models trained on 50 node and 100 node problems. **Bold** scores indicate best performing models in their respective groups. The scores and optimality gaps presented are averaged across their respective groups. Underlined results indicate the SHIELD-equivalent model for MVMoE, while *italicized* results indicate the SHIELD-equivalent model of MVMoE-Deeper.

| | | MTMDVRP50 | | | | | | MTMDVRP100 | | | | | |
| | | In-dist | | | Out-dist | | | In-dist | | | Out-dist | | |
| | Model | Obj | Gap | Time | Obj | Gap | Time | Obj | Gap | Time | Obj | Gap | Time |
|---|---|---|---|---|---|---|---|---|---|---|---|---|---|
| In-task | Solver | 5.8773 | - | 74.72s | 6.1866 | - | 72.89s | 9.0468 | - | 194.00s | 9.6506 | - | 187.89s |
| | POMO-MTVRP | 6.0778 | 3.5079% | 2.65s | 6.4261 | 3.9911% | 2.76s | 9.4123 | 4.0824% | 8.13s | 10.1147 | 5.0253% | 8.20s |
| | MVMoE | _6.0557_ | _3.1479%_ | _3.65s_ | _6.3924_ | _3.5071%_ | _3.67s_ | _9.3722_ | _3.5969%_ | _10.97s_ | _10.0827_ | _4.6855%_ | _11.30s_ |
| | MVMoE-Light | 6.0666 | 3.3595% | 3.41s | 6.4045 | 3.6860% | 3.43s | 9.3987 | 3.9088% | 10.04s | 10.1027 | 4.8979% | 10.46s |
| | MVMoE-Deeper | *6.0337* | *2.7343%* | *9.03s* | *6.3677* | *3.1333%* | *9.03s* | *OOM* | *OOM* | *OOM* | *OOM* | *OOM* | *OOM* |
| | SHIELD-MoD | 6.0220 | 2.5041% | 5.40s | 6.2933 | 2.9517% | 5.38s | 9.3453 | 2.5443% | 17.59s | 9.9800 | 3.5255% | 17.66s |
| | SHIELD-400Ep | _6.0597_ | _3.1495%_ | _6.14s_ | _6.3830_ | _3.2730%_ | _6.11s_ | _9.3785_ | _3.5993%_ | _19.90s_ | _10.0559_ | _4.3562%_ | _20.27s_ |
| | SHIELD-600Ep | *6.0333* | *2.7089%* | *6.15s* | *6.3653* | *2.9993%* | *6.09s* | *9.3194* | *2.9498%* | *19.88s* | *10.0113* | *3.8262%* | *20.28s* |
| | SHIELD | **6.0136** | **2.3747%** | **6.13s** | **6.2784** | **2.7376%** | **6.11s** | **9.2743** | **2.4397%** | **19.93s** | **9.9501** | **3.1638%** | **20.25s** |
| Out-task | Solver | 5.4513 | - | 78.00s | 5.7941 | - | 75.70s | 8.7852 | - | 160.90s | 9.4545 | - | 160.44s |
| | POMO-MTVRP | 5.8611 | 7.6284% | 2.83s | 6.2556 | 8.0311% | 2.70s | 9.4304 | 8.1068% | 8.39s | 10.2056 | 8.8907% | 8.46s |
| | MVMoE | _5.8328_ | _7.1553%_ | _3.81s_ | _6.2196_ | _7.5174%_ | _3.73s_ | _9.3811_ | _7.4092%_ | _11.13s_ | _10.1665_ | _8.5140%_ | _11.44s_ |
| | MVMoE-Light | 5.8466 | 7.4996% | 3.46s | 6.2346 | 7.8236% | 3.50s | 9.4173 | 7.9110% | 10.27s | 10.1945 | 8.8620% | 10.75s |
| | MVMoE-Deeper | *5.8207* | *6.7924%* | *9.40s* | *6.2136* | *7.2962%* | *9.45s* | *OOM* | *OOM* | *OOM* | *OOM* | *OOM* | *OOM* |
| | SHIELD-MoD | 5.7902 | 6.2672% | 5.47s | 6.2238 | 6.6155% | 5.48s | 9.2740 | 6.0296% | 17.75s | 10.0349 | 6.9029% | 17.79s |
| | SHIELD-400Ep | _5.8290_ | _7.1064%_ | _6.23s_ | _6.2085_ | _7.2927%_ | _6.21s_ | _9.3499_ | _6.9578%_ | _19.88s_ | _10.1202_ | _7.8332%_ | _20.15s_ |
| | SHIELD-600Ep | *5.8039* | *6.6539%* | *6.19s* | *6.1823* | *6.8736%* | *6.22s* | *9.3105* | *6.4308%* | *19.91s* | *10.0765* | *7.2549%* | *20.11s* |
| | SHIELD | **5.7779** | **6.0810%** | **6.20s** | **6.1570** | **6.3520%** | **6.20s** | **9.2400** | **5.6104%** | **19.92s** | **9.9867** | **6.2727%** | **20.18s** |

SHIELD-MoD and SHIELD have increased parameters and FLOPs due to the number of decoder layers. Interestingly, compared to MVMoE-Deeper (which also has three layers of decoder), we reduce the FLOP budget per step by imposing sparsity on the network. By constraining the compute budget, we effectively regularize the model and improve its generalization capabilities.

**Generalization of SHIELD.** To further evaluate the generalization capability of SHIELD, Table 1 shows its performance at earlier checkpoints, epochs 400 (SHIELD-400Ep) and epochs 600 (SHIELD-600Ep), that match the In-Task In-Distribution performance of MVMoE and MVMoE-Deeper, respectively. Our SHIELD counterparts show superior generalization across tasks and distributions, cementing its capability and flexibility as a general foundation model.

### 4.2. Ablation and Analyses

We discuss key observations and ablation studies here, and provide full tables and further details in Appendices J to R.

**Effect of Sparsity.** To examine the effect of sparsity, we train models with varying capacities of the MoD layer on MTMDVRP50. The results are shown in Table 2. Specifically, as the sparsity moves from 10% to 20%, the model's bias improves—the in-task performance improves slightly, while the out-distribution performs begins to degrade. Increasing the number of tokens further improves the in-task in-distribution optimality gaps, but we see a decline in performance for out-task and out-distribution settings. This

degradation continues with the 40% model, where overall performance deteriorates. The results indicate that sparsity is crucial in generalization across task and distribution.

**Effect of Clustering.** In the latent space, the soft clustering mechanism facilitates information exchange among dynamic clusters, enabling the model to capture high-level, generalizable features from neighboring hidden representations. This improves the model's understanding of the node selection process and enhances decision-making. Limiting the number of clusters reduces the number of parameters and promotes abstraction, which encourages the model to focus on broadly applicable patterns rather than overfitting task-specific details. In contrast, too many clusters dilute this effect, leading to over-segmentation and reduced generalization as the model prioritizes more complex patterns over shared structures. Table 3 supports this, whereby we vary the number of cluster centers in the model. Thus, maintaining sparsity in this aspect is crucial as well.

**Importance of Multi-Distribution.** To verify that our architecture improves overall, we trained and tested all models on the conventional MTVRP setting using the *uniform distribution*. Table 13 in Appendix Q showcases the performance of all models. Here, we see that while the gaps between the models are less significant once we remove the varied distributions, SHIELD is still clearly the better-performing model. This indicates the difficulty of a multi-distribution scenario – having varied structures with multiple tasks is more complex. Since our architecture is more flexible, it generalizes better in the MTMDVRP scenario.

*Table 2.* Performance of SHIELD with varying levels of sparsity on MTMDVRP50. As more nodes are processed the model's bias improves, but generalization degrades.

| | Model | In-dist | | Out-dist | |
|---|---|---|---|---|---|
| | | Obj | Gap | Obj | Gap |
| In-task | SHIELD (10%) | 6.0136 | 2.3747% | **6.2784** | **2.7376%** |
| | SHIELD (20%) | 6.0055 | 2.2268% | 6.3578 | 2.8442% |
| | SHIELD (30%) | **6.0033** | **2.1948%** | 6.3656 | 2.9608% |
| | SHIELD (40%) | 6.0131 | 2.3450% | 6.3718 | 3.0507% |
| | MVMoE-Deeper (100%) | 6.0337 | 2.7343% | 6.3677 | 3.1333% |
| Out-task | SHIELD (10%) | 5.7779 | 6.0810% | **6.1570** | **6.3520%** |
| | SHIELD (20%) | **5.7772** | **6.0327%** | 6.1671 | 6.4654% |
| | SHIELD (30%) | 5.7991 | 6.4241% | 6.1732 | 6.5603% |
| | SHIELD (40%) | 5.8068 | 6.5770% | 6.1862 | 6.7831% |
| | MVMoE-Deeper (100%) | 5.8206 | 6.7924% | 6.2136 | 7.2962% |

Next, we apply the trained models to the MTMDVRP test set and tabulate the results in Table 11 in Appendix O. Since all models are only trained on uniform data, they are unsuitable to be applied to more structured forms of data. Instead, if the model is exposed to some structure during training, it performs better in generalization to new distributions.

**Sparse Encoder.** Table 7 in Appendix K studies the impact of sparsity in the encoder. We replace encoder layers with MoD layers of capacity of 10% and find that the model's performance degrades significantly, even after doubling the number of layers.

This shows that the MoE encoder plays a crucial role in the architecture. In MTMDVRP, the encoder processes diverse multi-task contexts and learns meaningful representations from various task contexts which feature combinations of constraints. For example, CVRPTW combines capacity and time window constraints, while CVRPBLTW further adds backhaul and linehaul constraints. MoE is well-suited for the encoder as it leverages specialized expert subnetworks to handle the shared and combinatorial patterns in the inputs.

In contrast, the decoder in MTMDVRP focuses on sequential solution construction with adaptive computation. While some node selections are straightforward, others require finer granularity and greater computational/reasoning capacity – especially when dealing with clustered distributions or complex constraint-distribution interactions in MTMD-VRP. Thus, dynamic control over depth and computation is essential. MoD naturally addresses this need by adaptively allocating resources across decoder layers. Together, their synergy enhances the model's ability to capture context-dependent, adaptive fine-grained decisions for MTMDVRP.

**Alternative Sparse Attention Approaches.** Apart from studying the effect of sparsity in the encoder, we investigate similar sparse attention approaches such as INViT (Fang et al., 2024). Essentially, INViT proposes to only attend to the k-Nearest Neighbors (k-NN) during solution construction, as attention to all nodes introduces an aliasing ef-

*Table 3.* Ablation study for the number of clusters in SHIELD on MTMDVRP50. Keeping the number of clusters low, and thus having a sparser approach, is beneficial to the model.

| | Model | In-dist | | Out-dist | |
|---|---|---|---|---|---|
| | | Obj | Gap | Obj | Gap |
| In-task | SHIELD | 6.0136 | 2.3747% | **6.2784** | **2.7376%** |
| | SHIELD ($N_c = 10$) | **6.0100** | **2.3166%** | 6.3400 | 3.7522% |
| | SHIELD ($N_c = 20$) | 6.0124 | 2.3272% | 6.3437 | 3.8127% |
| Out-task | SHIELD | **5.7779** | **6.0810%** | **6.1570** | **6.3520%** |
| | SHIELD ($N_c = 10$) | 5.8019 | 6.9521% | 6.1740 | 7.0129% |
| | SHIELD ($N_c = 20$) | 5.9824 | 11.3453% | 6.3369 | 10.8044% |

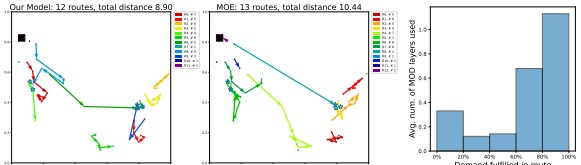

*Figure 3. Left two panels:* Plot of routes for OVRPBTW between SHIELD (left) and MVMoE (middle). Points denoted with a star are the top few points that SHIELD identified for more processing. Note that the initial routes from the depot are masked away for a better view. *Right panel:* Average number of layers used as the demand is being met for CVRP.

fect, which confuses the decoder, resulting in poor decision-making. Only attending to the k-NN nodes effectively reduces the number of interactions amongst the nodes and thus introduces sparsity into the attention mechanism, a somewhat similar approach to SHIELD. A key difference between the approaches is that INViT's reduction is based on a heuristic, the k-NN, while in SHIELD, we opt to learn which nodes to focus on based on MoD.

We adapt and train INViT on the MTMDVRP scenario; the results are shown in Table 6 in Appendix J. INViT struggles with the multi-task dynamics of the problem, likely because the sparse attention mechanism relies on selecting the k-NN nodes based on spatial distance. This is highly inflexible and poorly suited for a dynamic MTMDVRP setting. As such, essential nodes are possibly pruned away, leading to an inferior neural solver.

**Patterns of Layer Selection.** Figure 3 shows the output of SHIELD and MVMoE for OVRPBTW on VM22775. The starred points indicate that SHIELD selects these points more frequently when solving problems. Consider route R5 for SHIELD and route R8 for MVMoE. SHIELD can recognize that such points are far away from the depot and that visiting other points en route is more advantageous, whereas MVMoE only visited one node before returning. Likewise, for route R4 in SHIELD and route R6 in MVMoE, SHIELD identifies the two starred points to be better served as connecting points instead of making an entire loop, which results in back-tracking to a similar area. Since the problem

is an open problem, we can see that SHIELD favors ending routes at faraway locations, whereas MVMoE tends to loop back and forth in many occurrences.

The right panel of Figure 3 illustrates how the use of layers is distributed as the agent starts to address the demands of the problem. The $x$-axis represents the percentage of the sub-tour solved, while the $y$-axis denotes the average number of MoD layers used by the agent. The model initially uses some processing power to find a good starting node set. In the middle, fewer layers are being used, and finally, as the problem ends, more layers are activated to select effective ending points. Additional qualitative analysis in Figure 5 in Appendix L shows that for maps with similar top density and right bias, the model behaves somewhat similarly regarding its overall layer usage.

**Size Generalization.** To explore how our model behaves on problem sizes beyond what it was trained on, we generate and label an additional dataset with 200 nodes each. For the MTMDVRP200, we increased the time allowed to solve each instance to 80 seconds. Table 8 in Appendix M illustrates the zero-shot generalization performance of trained MTMDVRP100 models on the MTMDVRP200. SHIELD is still the superior model to the other baselines, showing a sizeable performance gap on problems larger than it was trained on. Additionally, note that the inference time of SHIELD is comparable to MVMoE and MVMoE-Light. This is because in the MTMDVRP200, inference on the MVMoE models requires smaller batch sizes, whereas SHIELD's sparsity allows it to process larger batches.

We also investigate the performance of all models on a zero-shot size generalization setting to the CVRPLib Set-X. Tables 9 and 10 in Appendix N show that SHIELD outperforms all models considerably in the Large setting ($101 \leq N \leq 251$) and the Extra Large setting ($502 \leq N \leq 1001$). We attribute the flexibility of dynamic processing in SHIELD to the strong zero-shot performance.

**Single-Task Multi-Distribution.** Table 12 in Appendix P showcases the performances of models trained on a single task, CVRP, across our various distributions. As SHIELD is still the top-performing model, the results suggest our architecture is not catered only to the MTMDVRP scenario – its flexibility allows for strong generalization across distributions for the single task case.

## 5. Conclusion

The push toward unified generic solvers is essential in building foundation models for neural combinatorial optimization. In this paper, we propose to extend such solvers to the Multi-Task Multi-Distribution VRP, a significantly more practical representation of industrial problems. With this problem setting, we propose SHIELD. This neural archi-

tecture, motivated by regularization via compute and parameters, is designed to handle generalization across task and distribution dimensions, making it a robust solver for practical problems. Extensive experiments and thorough analysis of the empirical results demonstrate that *sparsity* and *hierarchy*, two key techniques in SHIELD, substantially influence the model's generalization ability. This forms a stepping stone towards other foundation models, such as generalizing across various sizes.

## Acknowledgements

This work is funded by the Grab-NUS AI Lab, a joint collaboration between GrabTaxi Holdings Pte. Ltd. and National University of Singapore, and the Industrial Postgraduate Program (Grant: S18-1198-IPP-II) funded by the Economic Development Board of Singapore. This research is also supported by the Ministry of Education, Singapore, under its Academic Research Fund Tier 1 (A-8001814-00-00). This research is also supported by the National Research Foundation, Singapore under its AI Singapore Programme (AISG Award No: AISG3-RP-2022-031).

## Impact Statement

This paper presents work whose goal is to advance the field of Machine Learning. There are many potential societal consequences of our work, none which we feel must be specifically highlighted here.

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

# A. Related Work

**Generalization Study.** Joshi et al. (2021) highlighted the generalization challenge faced by neural solvers, where their performance drops significantly on out-of-distribution (OOD) instances. Numerous studies have sought to improve generalization performance in cross-size (Bdeir et al., 2022; Son et al., 2023; Huang et al., 2025), cross-distribution (Fang et al., 2024; Jiang et al., 2022; Bi et al., 2022; Zhang et al., 2022; Zhou et al., 2023), and cross-task (Lin et al., 2024; Liu et al., 2024; Zhou et al., 2024; Berto et al., 2024) settings. However, their methods are tailored to specific settings and cannot handle our MTMDVRP setup, which considers crossing tasks and realistic customer distributions. While a recent work Goh et al. (2024) explores more realistic TSPs, their approach struggles with complex cross-problem scenarios. In this paper, we take a step further by exploring generalization across both different problems and real-world distributions in VRPs.

**Multi-task VRP Solver.** Recent work in (Liu et al., 2024) explored the training of a Multi-Task VRP solver across a range of VRP variants sharing a set of common features indicating the presence or absence of specific constraints. Zhou et al. (2024) enhanced the model architecture by introducing Mixture-of-Experts within the transformer layers, allowing the model to capture representations tailored to different tasks effectively. These studies focus on zero-shot generalization, where models are trained on a subset of tasks and evaluated on unseen tasks that combine common features. Other studies (Wang & Yu, 2023; Drakulic et al., 2024) investigate this promising direction, but with different problem settings. Alternatively, Berto et al. (2024) improved convergence robustness by training on all tasks within a batch using a mixed environment.

**Single-task VRP Solver.** Most research focuses on developing single-task VRP solvers, which primarily follows two key paradigms: constructive solvers and improvement solvers. *Constructive solvers* learn policies that generate solutions from scratch in an end-to-end fashion. Early works proposed Pointer Networks (Vinyals et al., 2015) to approximate optimal solutions for TSP (Bello et al., 2017) and CVRP (Nazari et al., 2018) in an autoregressive (AR) way. A major breakthrough in AR-based methods came with the Attention Model (AM) (Kool et al., 2018), which became a foundational approach for solving VRPs. The policy optimization with multiple optima (POMO) (Kwon et al., 2020) improved upon AM by considering the symmetry property of VRP solutions. More recently, a wave of studies has focused on further boosting either the performance (Kim et al., 2022; Drakulic et al., 2023; Chalumeau et al., 2023; Grinsztajn et al., 2023; Luo et al., 2023; Hottung et al., 2024) or versatility (Kwon et al., 2021; Berto et al., 2023; Son et al., 2025) of these solvers to handle more complex and varied problem instances. Beyond AR methods, non-autoregressive (NAR) constructive approaches (Joshi et al., 2019; Fu et al., 2021; Kool et al., 2022; Qiu et al., 2022; Sun & Yang, 2023; Min et al., 2023; Ye et al., 2023; Kim et al., 2024; Xia et al., 2024) construct matrices, such as heatmaps representing the probability of each edge being part of the optimal solution, to solve VRPs through complex post-hoc search. In contrast, *improvement solvers* (Chen & Tian, 2019; Lu et al., 2020; Hottung & Tierney, 2020; Costa et al., 2020; Wu et al., 2021; Ma et al., 2021; Xin et al., 2021; Hudson et al., 2022; Ma et al., 2023) typically learn more efficient and effective search components, often within the framework of classic heuristics or meta-heuristics, to iteratively refine an initial solution. While constructive solvers can efficiently achieve desirable performance, improvement solvers have the potential to find near-optimal solutions given a longer time. There are also studies that focus on the scalability (Li et al., 2021; Hou et al., 2023; Ye et al., 2024), robustness (Geisler et al., 2022; Lu et al., 2023), and constraint handling (Bi et al., 2024) of neural VRP solvers, which are less related to our work. For those interested, we refer readers to Bogyrbayeva et al. (2024). Apart from such single-task VRP solvers, there are alternative approaches to complex routing problems, such as the PDP, where travel times change over time (Wen et al., 2022; Mao et al., 2023). These problems present additional dynamics that further increase the realism of VRPs.

# B. Generation of VRP Variants

As mentioned in Section 2, we consider four additional constraints on top of the CVRP, resulting in 16 different variants in total. Note that unlike (Liu et al., 2024; Zhou et al., 2024), we do not generate node coordinates from a uniform distribution. Instead, we sample a set of fixed points from a given map. Here, we detail the generation of the five total constraints.

**Capacity (C):** We adopt the settings from (Kool et al., 2018), whereby each node's demand $\delta_i$ is randomly sampled from a discrete distribution set, $\{1, 2, ..., 9\}$. For $N = 50$, the vehicle capacity $Q$ is set to 40, and for $N = 100$, the vehicle capacity is set to 50. All demands are first normalized to their vehicle capacities, so that $\delta_i' = \delta_i/Q$.

**Open route (O):** For open routes, we set $o_t = 1$ in the dynamic feature set received by the decoder. Apart from this, we remove the constraint that the vehicle has to return to the depot when it has completed the route or is unable to proceed further due to other constraints. Suppose the problem has both open routes (O) and duration limit (L), then we mask all nodes $v_j$ such that $l_t + d_{ij} > L$, whereby $d_{ij}$ is the distance between node $v_i$ and the potentially masked node $v_j$, and $L$ is

the duration limit constraint. For problems with both open routes (O) and time windows (TW), we mask all nodes $v_j$ such that $t_t + d_{ij} > w_j^c$, where $t_t$ is the current time after servicing the current node. Finally, suppose a route has both open routes (O) and backhauls (B), no special masking considerations are required as the vehicle does not return to the origin.

**Backhaul (B):** We adopt the approach from (Liu et al., 2024) by randomly selecting 20% of customer nodes to be backhauls, thus changing their demand to be negative instead. We also follow the same setup as (Zhou et al., 2024) whereby routes can have a mix of linehauls and backhauls without any strict precedence. To ensure feasible solutions, we ensure that all starting points are linehauls only unless all remaining nodes are backhauls.

**Duration limit (L):** The duration limit is fixed such that the maximum length of the vehicle, $L = 3$, which ensures that a feasible route can be found as all points are normalized to a unit square.

**Time window (TW):** For time windows, we follow the methodology in (Li et al., 2021). The depot node $v_0$ has a time window of $[0, 3]$ with no service time. As for other nodes, each node has a service time of $s_i = 0.2$, and the time windows are obtained as following: (1) first we sample a time window center given by $\gamma_i\, U(w_0^o + d_{0i}, w_i^c - d_{i0} - s_i)$, whereby $d_{0i} = d_{i0}$ is the distance or travel time between depot $v_0$ and node $v_i$, (2) then we sample a time window half-width $h_i$ uniformly from $[s_i/2, w_0^c/3] = [0.1, 1]$, (3) then we set the time window as $[w_i^o, w_i^c] = [\text{MAX}(w_i^o, \gamma_i - h_i), \text{MIN}(w_i^c, \gamma_i + h_i)]$.

## C. Neural Combinatorial Optimization Model Details

Neural constructive solvers are typically parameterized by a neural network, whereby a policy, $\pi_\theta$, is trained by reinforcement learning so as to construct a solution sequentially (Kool et al., 2018; Kwon et al., 2020). The attention-based mechanism (Vaswani, 2017) is popularly used, whereby attention scores govern the decision-making process in an autoregressive fashion. The overall feasibility of solution can be managed by the use of masking, whereby invalid moves are masked away during the construction process. Classically, neural constructive solvers employ an encoder-decoder architecture and are trained as sequence-to-sequence models (Sutskever, 2014). The probability of a sequence can be factorized using the chain-rule of probability, such that

$$p_\theta(\tau|\mathcal{G}) = \prod_{t=1}^{T} p_\theta(\tau_t|\mathcal{G}, \tau_{1:t-1}) \tag{9}$$

The encoder tends employ a typical transformer layer, whereby

$$\tilde{\mathbf{h}} = \text{LN}^l(\mathbf{h}_i^{l-1} + \text{MHA}_i^l(\mathbf{h}_i^{l-1}, ..., \mathbf{h}_N^{l-1})) \tag{10}$$

$$\mathbf{h}_i^l = \text{LN}^l(\tilde{\mathbf{h}}_i + \text{FF}(\tilde{\mathbf{h}}_i)) \tag{11}$$

where $h_i^l$ is the embedding of the $i$-th node at the $l$-th layer, MHA is the multi-headed attention layer, LN the layer normalization function, and FF a feed-forward multi-layer perceptron (MLP). All embeddings are passed through $L$ layers before reaching the decoder.

The decoder produces the solutions autoregressively, whereby a contextual embedding combines the embeddings from the starting and current location as follows

$$\mathbf{h}_{(c)} = \mathbf{h}_{\text{LAST}}^L + \mathbf{h}_{\text{START}}^L \tag{12}$$

Then, the attention mechanism is used to produce the attention scores. Notably, the context vectors $\mathbf{h}_{(c)}$ are denoted as query vectors, while keys and values are the set of $N$ node embeddings. This is mathematically represented as

$$a_j = \begin{cases} U \cdot \text{TANH}(\frac{\mathbf{Q}\mathbf{K}^\top}{\sqrt{\text{DIM}}}) & j \neq \tau_{t'}, \forall t' < t \\ -\infty & \text{otherwise} \end{cases} \tag{13}$$

whereby $U$ is a clipping function and DIM the dimension of the latent vector. These attention scores are then normalized using a softmax function to generate the following selection probability

$$p_i = p_\theta(\tau_t = i|s, \tau_{1:t-1}) = \frac{e^{a_j}}{\sum_j e^{a_j}} \tag{14}$$

Finally, given a baseline function $b(\cdot)$, the policy is trained with the REINFORCE algorithm (Williams, 1992) and gradient ascent, with the expected return $J$

$$\nabla_\theta J(\theta) \approx \mathbb{E}\Big[(R(\tau^i) - b^i(s))\nabla_\theta \log p_\theta(\tau^i|s)\Big] \tag{15}$$

The reward of each solution $R$ is the length of the solution tour.

## D. Soft-clustering Algorithm Details

---
**Algorithm 1** Psuedo code of soft clustering algorithm

---
1: **function** CLUSTER
**Require:**    encoder embeddings $H$, constraints vector $\gamma_k$, number of centers $N_c$, number of iterations $B$, initial embeddings $C$, embedding size $d$
2:    $\alpha_d = \mathbf{W}_\theta^\top \gamma_k$
3:    **for** $b \leftarrow 1$ to $B$ **do**
4:        $\hat{H} \leftarrow W_H(H)$
5:        $\hat{C} \leftarrow W_C([C, \alpha_d])$
6:        $\psi = \text{SOFTMAX}(\frac{\hat{H}\hat{C}^\top}{\sqrt{d}})$ {*Compute attention scores*}
7:        $C = \sum_i \psi_i h_i$ {*Update the centers with data*}
8:        $C_{\text{OUT}} = \hat{C} + C$ {*Residual connection*}
9:        $C = \text{NORM}(C_{\text{OUT}})$ {*Layer normalization*}
10:    **end for**
11:    **return** $C$
12: **end function**

---

# E. Dataset Details

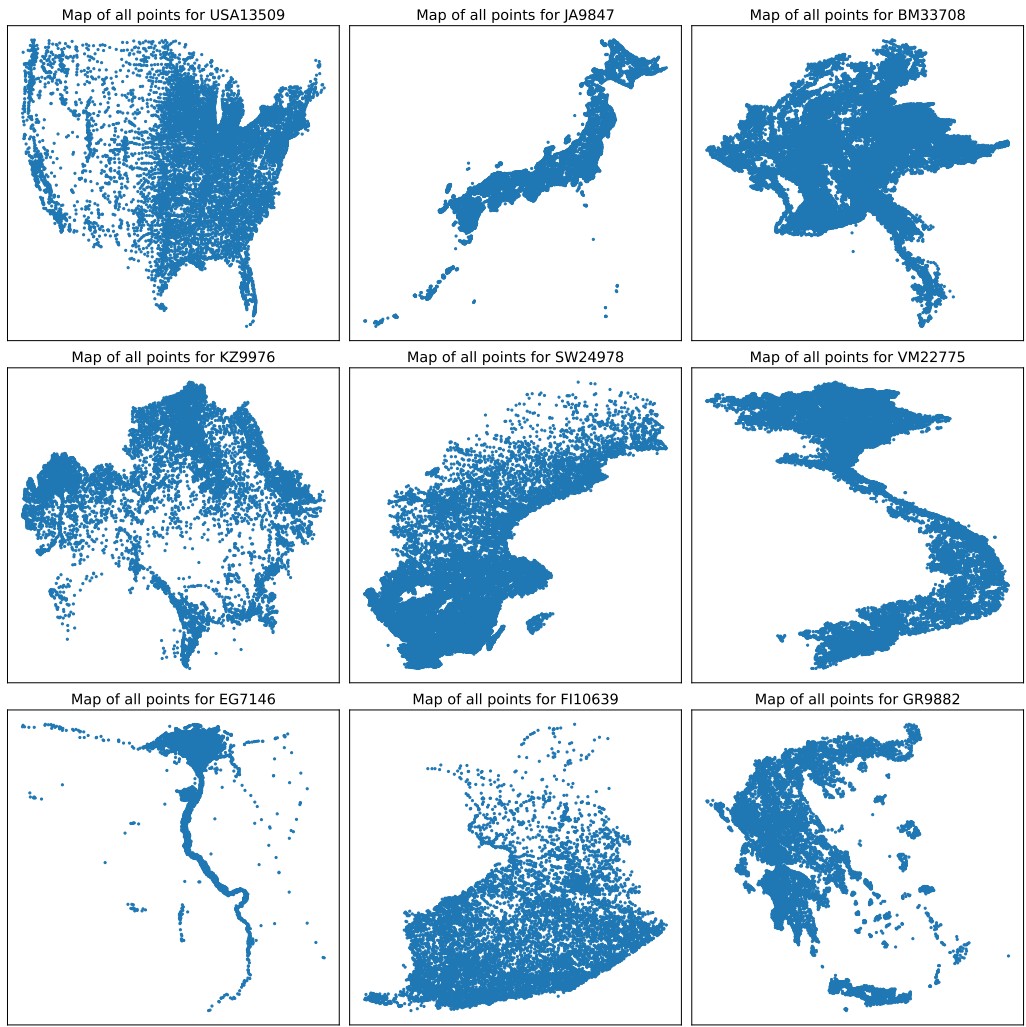

*Figure 4.* Plot of all 9 World Maps and their points

We utilize the following 9 country maps[2] shown in Figure 4: (1) USA13509: USA containing 13,509 cities; (2) JA9847: Japan containing 9,847 cities; (3) BM33708: Burma containing 33,708 cities; (4) KZ9976: Kazakhstan containing 9,976; (5) SW24978: Sweden containing 24,978 cities; (6) VM22775: Vietnam containing 22,775 cities; (7) EG7146: Egypt containing 7,146 cities; (8) FI10639: Finland containing 10,639 cities; (9) GR9882: Greece containing 9,882 cities.

---

[2]https://www.math.uwaterloo.ca/tsp/world/countries.html

## F. Model sizes and average runtimes

*Table 4.* Overall number of parameters and average runtimes for all models.

| Model | Num. Parameters | FLOPs on VRP50 | Runtime on VRP50 | Runtime on VRP100 |
|---|---|---|---|---|
| POMO-MTVRP | 1.25M | 52.88 GFLOPs | 2.74s | 8.30s |
| MVMoE | 3.68M | 84.41 GFLOPs | 3.72s | 11.21s |
| MVMoE-Light | 3.70M | 84.03 GFLOPs | 3.45s | 10.38s |
| MVMoE-Deeper | 4.46M | 114.99 GFLOPs | 9.23s | OOM |
| SHIELD-MoD | 4.37M | 95.76 GFLOPs | 5.43s | 17.70s |
| SHIELD | 4.59M | 106.72 GFLOPs | 6.16s | 20.07s |

Table 4 showcases the number of parameters per model, the number of floating operations on MTMDVRP50, and the runtimes on MTMDVRP50 and MTMDVRP100. Note that the total FLOPs is calculated based on a single pass through the encoder and one decoding step. The FLOPs is also a sum of the forward and backward passes for gradient updates. We only use one decoding step as inferior solvers will require more steps to solve the problem, and thus would also require more FLOPs.

## G. Mathematical Notations

| | |
|---|---|
| $\mathcal{S}_i$ | A problem instance $i$ |
| $\mathcal{D}_t$ | Set of dynamic features at decoding time-step $t$ |
| $t$ | Decoding time-step |
| $x_i$ | $x$-coordinate of problem instance $i$ |
| $y_i$ | $y$-coordinate of problem instance $i$ |
| $\delta_i$ | Demand of node $i$ |
| $w_i^o$ | Opening timing of time-window for node $i$ |
| $w_i^c$ | Closing timing of time-window for node $i$ |
| $z_t$ | Capacity of vehicle at decoding time-step $t$ |
| $t_t$ | Current time-step |
| $o_t$ | Presence of open route at time-step $t$ |
| $l_t$ | Current length of partial route at time-step $t$ |
| $\mathcal{K}$ | Set of all possible VRP tasks |
| $\mathcal{Q}$ | Set of all possible distributions |
| $\beta$ | The percentage of tokens allowed through a MoD layer |
| $r_i$ | Router score for node $i$ |
| $\gamma_k$ | One-hot encoded vector of constraints for task $k$ |
| $o_t$ | Presence of open route at time-step $t$ |
| $B$ | Number of iterations of clustering |
| $N_c$ | Number of cluster centers |
| $\psi_{ij}$ | Mixing coefficient between node $i$ and cluster $j$ |

# H. Solver and Metric Details

We use HGS (Vidal, 2022) for CVRP and VRPTW instances, and Google's OR-tools routing solver (Furnon & Perron) for the rest. For HGS, we use the default hyperparameters, while for OR-tools, we apply parallel cheapest insertion as the initial solution strategy and guided local search as the local search strategy. The time limit is set to 20s and 40s for solving a single instance of size $N = 50, 100$, respectively. For neural solving, we utilize 8x augmentations on the $(x, y)$-coordinates for the test set as proposed by (Kwon et al., 2020). The following table details the various transformations applied.

*Table 5.* List of augmentations suggested by (Kwon et al., 2020)

| $f(x, y)$ | |
| --- | --- |
| $(x, y)$ | $(y, x)$ |
| $(x, 1 - y)$ | $(y, 1 - x)$ |
| $(1 - x, y)$ | $(1 - y, x)$ |
| $(1 - x, 1 - y)$ | $(1 - y, 1 - x)$ |

The optimality gap is measured as the percentage gap between the neural solver's tour length and the traditional solver. This is defined as

$$O = \left( \frac{\frac{1}{N} \sum_i^N R_i}{\frac{1}{N} \sum_i^N L_i} - 1 \right) * 100 \tag{16}$$

where $L_i$ is the tour length of test instance $i$ computed by the traditional solver, HGS or OR-Tools.

# I. Detailed hyperparameter and training settings

- Number of MoE encoder layers: 6
- Total number of experts: 4
- Number of experts used per layer: 2
- Number of MoD decoder layers: 3
- Capacity of MoD layer (number of tokens allowed): 10%
- Number of single-headed attention decision-making layer: 1
- Latent dimension size: 128
- Number of heads per transformer layer: 8
- Feedforward MLP size: 512
- Logit clipping $U$: 10
- Learning rate: $1e^{-4}$
- Number of clustering layers: 1
- Number of iterations for clustering: 5
- Number of learnable cluster embeddings: 5
- Number of episodes per epoch: 20,000
- Number of epochs: 1,000
- Batch size: 128

## J. Additional Experiments – Alternative sparse approaches in Encoder

*Table 6.* Performance of INViT and SHIELD on the MTMDVRP50 and MTMDVRP100 scenarios. INViT struggles with the complexity of the MTMDVRP compared to SHIELD despite using some form of sparse attention.

| | | MTMDVRP50 | | | | | | MTMDVRP100 | | | | | |
| | | In-dist | | | Out-dist | | | In-dist | | | Out-dist | | |
| | Model | Obj | Gap | Time | Obj | Gap | Time | Obj | Gap | Time | Obj | Gap | Time |
|---|---|---|---|---|---|---|---|---|---|---|---|---|---|
| In-task | INViT | 6.4082 | 9.1437% | 66.48s | 6.7462 | 9.0992% | 66.84s | 10.6057 | 17.2425% | 66.65s | 11.4286 | 18.4235% | 68.06s |
| | SHIELD | 6.0136 | 2.3747% | 6.13s | 6.2784 | 2.7376% | 6.11s | 9.2743 | 2.4397% | 19.93s | 9.9501 | 3.1638% | 20.25s |
| Out-task | INViT | 6.2996 | 15.3570% | 69.43s | 6.6932 | 15.2064% | 70.11s | 11.1489 | 26.8217% | 68.00s | 12.1012 | 27.9947% | 69.98s |
| | SHIELD | 5.7779 | 6.0810% | 6.20s | 6.1570 | 6.3520% | 6.20s | 9.2400 | 5.6104% | 19.92s | 9.9867 | 6.2727% | 20.18s |

A similar sparse attention approach would be INViT (Fang et al., 2024). Essentially, INViT proposes to only attend to the k-Nearest Neighbors (k-NN) during solution construction, as attention to all nodes introduces an aliasing effect, which confuses the decoder, resulting in poor decision-making. Only attending to the k-NN nodes effectively reduces the number of interactions amongst the nodes and thus introduces sparsity into the attention mechanism, a somewhat similar approach to SHIELD. A key difference between the approaches is that INViT's reduction is based on a heuristic, the k-NN, while in SHIELD, we opt to learn which nodes to focus on based on MoD.

Results shown in 6 compares SHIELD and a trained INViT model. We utilize the same training and hyperparameter settings as INViT-3 on our data and environment setup. As shown, INViT struggles with the multi-task dynamics of the problem, likely because the sparse attention mechanism relies on selecting the k-NN nodes based on spatial distance. This is highly inflexible and poorly suited for a dynamic MTMDVRP setting. As such, essential nodes are possibly pruned away, leading to an inferior neural solver.

## K. Additional experiments – Effect of sparsity in Encoder

*Table 7.* Experimental study for the impacts of using MoD layers in the encoder on MTMDVRP50. Even by increasing the number of layers, the model's performance is unsatisfactory.

| | | In-dist | | Out-dist | |
| | Model | Obj | Gap | Obj | Gap |
|---|---|---|---|---|---|
| | SHIELD | 6.0136 | 2.3747% | 6.2784 | 2.7376% |
| In-task | SHIELD (MoDEnc-6) | 6.2271 | 6.2578% | 6.6213 | 7.6650% |
| | SHIELD (MoDEnc-12) | 6.1838 | 5.4944% | 6.5817 | 7.1229% |
| | SHIELD | 5.7779 | 6.0810% | 6.1570 | 6.3520% |
| Out-task | SHIELD (MoDEnc-6) | 6.0432 | 11.5021% | 6.4894 | 12.9905% |
| | SHIELD (MoDEnc-12) | 5.9846 | 10.3009% | 6.4322 | 12.0432% |

Table 7 studies the impact of sparsity in the encoder. We replace encoder layers with MoD layers of capacity of 10% and find that the model's performance degrades significantly, even after doubling the number of layers. This shows that the MoE encoder plays a crucial role in the architecture – it enables the model to leverage various experts to capture a broad range of representations for effective encoding. In contrast, the MoD introduces greater flexibility in the decoder, allowing the model to dynamically select layers for decision-making, which helps it adapt effectively to varying outputs.

## L. Additional experiments – Average layer usage per token for CVRP on various distributions

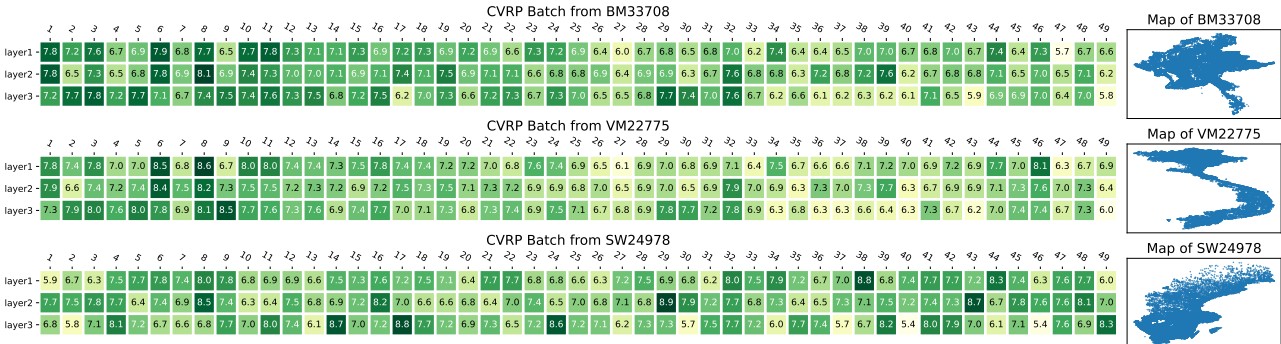

*Figure 5.* Plot of layer usage for CVRP samples across three maps, with the $x$-axis as node IDs, $y$-axis as layer numbers, and values as average usage frequency during decoding.

We conduct further analysis on the simpler CVRP to examine how the model generalizes across tasks and distributions. Figure 5 presents a heat map where we average the number of times a layer is used when the agent is positioned on a node. Note that the $x$-axis denotes the node ID, while the $y$-axis denotes the layer number, with the value indicating the average number of times that combination is called. For this analysis, we sort the nodes in anticlockwise order based on their $x$ and $y$ coordinates to impose a spatial ordering. We observe that for maps with similar top density and curved shapes, such as BM33708 and VM22775, the MoD layers tend to exhibit a similar pattern in layer usage, whereas a map like SW24978 has a much different sort of distribution.

## M. Additional experiments – Size Generalization to MTMDVRP200

*Table 8.* Performance of trained MTMDVRP100 models on MTMDVRP200. SHIELD is the superior model even when tested on problem sizes larger than those it was trained on.

|  | Model | MTMDVRP200 | | | | | |
|---|---|---|---|---|---|---|---|
|  |  | | In-dist | | | Out-dist | |
|  | | Obj | Gap | Time | Obj | Gap | Time |
| In-task | Solver | 13.7525 | - | 943.23s | 14.8228 | - | 921.81s |
|  | POMO-MTVRP | 14.5695 | 5.4613% | 19.80s | 15.9036 | 7.0430% | 20.01s |
|  | MVMoE | 14.6137 | 5.8753% | 44.25s | 15.9391 | 7.3486% | 44.40s |
|  | MVMoE-Light | 14.6420 | 6.0924% | 40.87s | 15.9581 | 7.4784% | 41.92s |
|  | SHIELD-MoD | 14.4123 | 4.7980% | 37.89s | 15.7342 | 6.1487% | 38.01s |
|  | SHIELD | 14.3648 | 3.7939% | 42.24s | 15.6536 | 5.0516% | 40.04s |
| Out-task | Solver | 14.4622 | - | 973.11s | 15.7897 | - | 959.09s |
|  | POMO-MTVRP | 15.5735 | 8.5203% | 21.31s | 17.1759 | 10.2531% | 26.78s |
|  | MVMoE | 15.6040 | 8.8840% | 45.49s | 17.2145 | 10.5085% | 45.23 |
|  | MVMoE-Light | 15.6412 | 9.1470% | 43.22s | 17.2423 | 10.7143% | 43.94s |
|  | SHIELD-MoD | 15.5373 | 7.4336% | 39.13s | 17.1948 | 8.8987% | 39.06s |
|  | SHIELD | 15.3896 | 6.4856% | 42.86s | 16.9555 | 7.8179% | 47.92s |

We generate and label an additional dataset with 200 nodes each. For the MTMDVRP200, we increased the time allowed to solve each instance to 80 seconds. Table 8 illustrates the zero-shot generalization performance of trained MTMDVRP100 models on the MTMDVRP200. SHIELD is still the superior model to the other baselines, showing a sizeable performance gap on problems larger than it was trained on. Additionally, note that the inference time of SHIELD is comparable to MVMoE and MVMoE-Light. This is because in the MTMDVRP200, inference on the MVMoE models requires smaller batch sizes, whereas SHIELD's sparsity allows it to process larger batches.

# N. Additional experiments – Generalization to CVRPLib

*Table 9.* Performance on CVRPLib data Set-X-1. Instances vary from 101 to 251 nodes.

| Set-X-1 | | POMO-MTL | | MVMoE | | MVMoE-Light | | SHIELD-MoD | | SHIELD | | SHIELD-Ep400 | |
|---|---|---|---|---|---|---|---|---|---|---|---|---|---|
| Instance | Opt. | Obj. | Gap | Obj. | Gap | Obj. | Gap | Obj. | Gap | Obj. | Gap | Obj. | Gap |
| X-n101-k25 | 27591 | 29875 | 8.2781% | 29189 | 5.7917% | 29445 | 6.7196% | 28967 | 4.9871% | 28678 | 3.9397% | 29346 | 6.3608% |
| X-n106-k14 | 26362 | 27158 | 3.0195% | 27061 | 2.6515% | 27356 | 3.7706% | 26909 | 2.0750% | 27076 | 2.7084% | 27192 | 3.1485% |
| X-n110-k13 | 14971 | 15420 | 2.9991% | 15379 | 2.7253% | 15387 | 2.7787% | 15450 | 3.1995% | 15316 | 2.3045% | 15312 | 2.2777% |
| X-n115-k10 | 12747 | 13680 | 7.3194% | 13368 | 4.8717% | 13536 | 6.1897% | 13245 | 3.9068% | 13290 | 4.2598% | 13472 | 5.6876% |
| X-n120-k6 | 13332 | 13939 | 4.5530% | 14082 | 5.6256% | 13980 | 4.8605% | 13901 | 4.2679% | 13724 | 2.9403% | 13971 | 4.7930% |
| X-n125-k30 | 55539 | 58929 | 6.1038% | 58443 | 5.2288% | 59056 | 6.3325% | 58648 | 5.5979% | 57426 | 3.3976% | 58277 | 4.9299% |
| X-n129-k18 | 28940 | 30114 | 4.0567% | 29905 | 3.3345% | 29970 | 3.5591% | 29802 | 2.9786% | 29540 | 2.0733% | 29695 | 2.6088% |
| X-n134-k13 | 10916 | 11637 | 6.6050% | 11658 | 6.7974% | 11612 | 6.3760% | 11519 | 5.5240% | 11274 | 3.2796% | 11447 | 4.8644% |
| X-n139-k10 | 13590 | 14295 | 5.1876% | 14155 | 4.1575% | 14121 | 3.9073% | 13988 | 2.9286% | 14004 | 3.0464% | 14152 | 4.1354% |
| X-n143-k7 | 15700 | 17091 | 8.8599% | 16710 | 6.4331% | 16744 | 6.6497% | 16621 | 5.8662% | 16548 | 5.4013% | 16792 | 6.9554% |
| X-n148-k46 | 43448 | 47317 | 8.9049% | 45621 | 5.0014% | 45794 | 5.3996% | 45728 | 5.2477% | 44739 | 2.9714% | 45082 | 3.7608% |
| X-n153-k22 | 21220 | 23689 | 11.6352% | 23267 | 9.6466% | 23510 | 10.7917% | 23541 | 10.9378% | 23252 | 9.5759% | 23392 | 10.2356% |
| X-n157-k13 | 16876 | 17730 | 5.0604% | 17698 | 4.8708% | 17713 | 4.9597% | 17386 | 3.0220% | 17366 | 2.9035% | 17583 | 4.1894% |
| X-n162-k11 | 14138 | 14845 | 5.0007% | 14884 | 5.2766% | 14746 | 4.3005% | 14703 | 3.9963% | 14767 | 4.4490% | 14804 | 4.7107% |
| X-n167-k10 | 20557 | 21863 | 6.3531% | 21898 | 6.5233% | 21827 | 6.1779% | 21644 | 5.2877% | 21326 | 3.7408% | 21566 | 4.9083% |
| X-n172-k51 | 45607 | 50381 | 10.4677% | 48863 | 7.1393% | 48686 | 6.7512% | 48434 | 6.1986% | 48091 | 5.4465% | 48613 | 6.5911% |
| X-n176-k26 | 47812 | 53848 | 12.6244% | 52302 | 9.3909% | 51433 | 7.5734% | 52313 | 9.4140% | 51811 | 8.3640% | 50887 | 6.4314% |
| X-n181-k23 | 25569 | 26480 | 3.5629% | 26661 | 4.2708% | 26490 | 3.6020% | 26156 | 2.2957% | 26237 | 2.6125% | 26333 | 2.9880% |
| X-n186-k15 | 24145 | 25900 | 7.2686% | 25695 | 6.4195% | 25613 | 6.0799% | 25409 | 5.2350% | 25503 | 5.6244% | 25372 | 5.0818% |
| X-n190-k8 | 16980 | 17826 | 4.9823% | 18121 | 6.7197% | 18125 | 6.7432% | 17417 | 2.5736% | 17802 | 4.8410% | 17846 | 5.1001% |
| X-n195-k51 | 44225 | 49703 | 12.3867% | 47834 | 8.1605% | 47704 | 7.8666% | 47608 | 7.6495% | 46509 | 5.1645% | 47731 | 7.9276% |
| X-n200-k36 | 58578 | 61857 | 5.5977% | 62039 | 5.9084% | 61871 | 5.6216% | 61384 | 4.7902% | 61375 | 4.7748% | 61729 | 5.3792% |
| X-n209-k16 | 30656 | 32754 | 6.8437% | 32725 | 6.7491% | 32605 | 6.3576% | 32157 | 4.8963% | 32244 | 5.1801% | 32083 | 4.6549% |
| X-n219-k73 | 117595 | 120795 | 2.7212% | 119924 | 1.9805% | 121201 | 3.0665% | 119679 | 1.7722% | 119847 | 1.9150% | 119560 | 1.6710% |
| X-n228-k23 | 25742 | 30042 | 16.7042% | 28629 | 11.2151% | 28754 | 11.7007% | 28206 | 9.5719% | 28118 | 9.2301% | 28119 | 9.2339% |
| X-n237-k14 | 27042 | 29217 | 8.0430% | 29252 | 8.1725% | 29003 | 7.2517% | 28560 | 5.6135% | 28743 | 6.2902% | 28880 | 6.7968% |
| X-n247-k50 | 37274 | 43111 | 15.6597% | 40868 | 9.6421% | 41735 | 11.9681% | 41556 | 11.4879% | 40676 | 9.1270% | 41266 | 10.7099% |
| X-n251-k28 | 38684 | 41321 | 6.8168% | 40874 | 5.6613% | 40854 | 5.6096% | 40316 | 4.2188% | 40410 | 4.4618% | 40602 | 4.9581% |
| Averages | 31280 | 33601 | 7.4148% | 33111 | 6.0845% | 33174 | 6.1773% | 32902 | 5.1979% | 32703 | 4.6437% | 32897 | 5.3961% |

*Table 10.* Performance on CVRPLib data Set-X-2. Instances vary from 502 to 1001 nodes.

| Set-X-2 | | POMO-MTL | | MVMoE | | MVMoE-Light | | SHIELD-MoD | | SHIELD | | SHIELD-Ep400 | |
|---|---|---|---|---|---|---|---|---|---|---|---|---|---|
| Instance | Opt. | Obj. | Gap | Obj. | Gap | Obj. | Gap | Obj. | Gap | Obj. | Gap | Obj. | Gap |
| X-n502-k39 | 69226 | 73599 | 6.3170% | 75113 | 8.5040% | 75679 | 9.3216% | 73184 | 5.7175% | 73062 | 5.5413% | 73445 | 6.0945% |
| X-n513-k21 | 24201 | 27955 | 15.5118% | 29444 | 21.6644% | 28483 | 17.6935% | 27478 | 13.5408% | 27217 | 12.4623% | 27373 | 13.1069% |
| X-n524-k153 | 154593 | 175923 | 13.7975% | 174409 | 12.8182% | 170334 | 10.1822% | 167380 | 8.2714% | 169715 | 9.7818% | 166660 | 7.8057% |
| X-n536-k96 | 94846 | 104866 | 10.5645% | 105896 | 11.6505% | 104408 | 10.0816% | 102157 | 7.7083% | 102237 | 7.7926% | 103042 | 8.6414% |
| X-n548-k50 | 86700 | 94290 | 8.7543% | 93623 | 7.9850% | 92798 | 7.0334% | 91483 | 5.5167% | 91726 | 5.7970% | 92055 | 6.1765% |
| X-n561-k42 | 42717 | 48781 | 14.1958% | 49953 | 16.9394% | 48678 | 13.9546% | 47328 | 10.7943% | 47639 | 11.5223% | 47485 | 11.1618% |
| X-n573-k30 | 50673 | 57151 | 12.7839% | 55796 | 10.1099% | 55870 | 10.2560% | 54664 | 7.8760% | 53936 | 6.4393% | 55204 | 8.9416% |
| X-n586-k159 | 190316 | 208217 | 9.4059% | 209038 | 9.8373% | 208510 | 9.5599% | 205408 | 7.9300% | 205487 | 7.9715% | 208175 | 9.3839% |
| X-n599-k92 | 108451 | 118994 | 9.7214% | 119879 | 10.5375% | 118864 | 9.6016% | 117615 | 8.4499% | 116950 | 7.8367% | 118514 | 9.2788% |
| X-n613-k62 | 59535 | 68882 | 15.7000% | 72992 | 22.6035% | 69091 | 16.0511% | 66657 | 11.9627% | 66715 | 12.0601% | 66419 | 11.5629% |
| X-n627-k43 | 62164 | 69756 | 12.2129% | 69197 | 11.3136% | 68302 | 9.8739% | 67125 | 7.9805% | 67494 | 8.5741% | 67059 | 7.8743% |
| X-n641-k35 | 63682 | 72638 | 14.0636% | 72348 | 13.6082% | 71041 | 11.5559% | 69425 | 9.0182% | 69156 | 8.5958% | 69617 | 9.3197% |
| X-n655-k131 | 106780 | 115083 | 7.7758% | 113186 | 5.9993% | 113610 | 6.3963% | 111711 | 4.6179% | 110508 | 3.4913% | 111542 | 4.4596% |
| X-n670-k130 | 146332 | 177344 | 21.1929% | 173046 | 18.2557% | 170328 | 16.3983% | 164820 | 12.6343% | 166737 | 13.9443% | 164140 | 12.1696% |
| X-n685-k75 | 68205 | 79362 | 16.3580% | 84485 | 23.8692% | 79502 | 16.5633% | 76224 | 11.7572% | 76676 | 12.4199% | 76195 | 11.7147% |
| X-n701-k44 | 81923 | 90163 | 10.0582% | 92522 | 12.9378% | 89812 | 9.6298% | 88608 | 8.1601% | 87959 | 7.3679% | 88603 | 8.1540% |
| X-n716-k35 | 43373 | 50636 | 16.7454% | 51003 | 17.5916% | 49429 | 13.9626% | 47821 | 10.2552% | 47996 | 10.6587% | 47586 | 9.7134% |
| X-n733-k159 | 136187 | 158694 | 16.5265% | 156545 | 14.9486% | 156747 | 15.0969% | 148203 | 8.8232% | 149217 | 9.5677% | 153664 | 12.8331% |
| X-n749-k98 | 77269 | 88333 | 14.3188% | 91569 | 18.5068% | 88438 | 14.4547% | 84651 | 9.5536% | 85367 | 10.4803% | 85824 | 11.0717% |
| X-n766-k71 | 114417 | 135772 | 18.6642% | 133725 | 16.8751% | 129996 | 13.6160% | 128128 | 11.9834% | 128052 | 11.9169% | 127179 | 11.1539% |
| X-n783-k48 | 72386 | 84162 | 16.2683% | 85094 | 17.5559% | 82690 | 14.2348% | 80855 | 11.6998% | 80521 | 11.2384% | 80358 | 11.0132% |
| X-n801-k40 | 73305 | 85008 | 15.9648% | 84025 | 14.6238% | 83210 | 13.5120% | 81070 | 10.5927% | 80637 | 10.0020% | 81015 | 10.5177% |
| X-n819-k171 | 158121 | 177282 | 12.1179% | 178589 | 12.9445% | 175340 | 10.8898% | 171630 | 8.5435% | 172020 | 8.7901% | 175820 | 11.1933% |
| X-n837-k142 | 193737 | 213908 | 10.4115% | 214165 | 10.5442% | 211521 | 9.1795% | 208552 | 7.6470% | 209350 | 8.0589% | 210464 | 8.6339% |
| X-n856-k95 | 88965 | 99911 | 12.3037% | 102485 | 15.1970% | 98990 | 11.2685% | 99014 | 11.2955% | 96889 | 8.9069% | 97602 | 9.7083% |
| X-n876-k59 | 99299 | 110191 | 10.9689% | 111857 | 12.6467% | 111044 | 11.8279% | 106826 | 7.5801% | 106180 | 6.9296% | 107710 | 8.4704% |
| X-n895-k37 | 53860 | 65277 | 21.1975% | 66353 | 23.1953% | 64716 | 20.1560% | 62114 | 15.3249% | 62101 | 15.3008% | 61552 | 14.2815% |
| X-n916-k207 | 329179 | 360052 | 9.3788% | 362596 | 10.1516% | 359444 | 9.1941% | 354793 | 7.7812% | 353567 | 7.4087% | 355423 | 7.9726% |
| X-n936-k151 | 132715 | 173297 | 30.5783% | 167723 | 26.3783% | 163193 | 22.9650% | 158308 | 19.2842% | 159965 | 20.5327% | 156897 | 18.2210% |
| X-n957-k87 | 85465 | 98132 | 14.8213% | 99442 | 16.3541% | 97109 | 13.6243% | 94209 | 10.2311% | 93672 | 9.6028% | 94118 | 10.1246% |
| X-n979-k58 | 118976 | 132128 | 11.0543% | 132449 | 11.3241% | 131752 | 10.7383% | 128765 | 8.2277% | 129968 | 9.2388% | 127952 | 7.5444% |
| X-n1001-k43 | 72355 | 87428 | 20.8320% | 87802 | 21.3489% | 86285 | 19.2523% | 82866 | 14.5270% | 82407 | 13.8926% | 82253 | 13.6798% |
| Averages | 101874 | 115725 | 14.0802% | 116136 | 14.9631% | 114225 | 12.7539% | 111534 | 9.8527% | 111598 | 9.8164% | 111905 | 10.0618% |

Tables 9 and 10 showcase various models trained on MTMDVRP100 applied to data from the CVRPLib Set-X-1 (Large) and Set-X-2 (Extra Large). These instances have varying sizes from 101 to 1001 nodes. Additionally, we include SHIELD-Ep400, the 400th epoch of training SHIELD, which has similar in-task in-dist performance compared to MVMoE. SHIELD is a significantly superior model in terms of zero-shot size generalization.

## O. Additional experiments – Importance of Varied Distributions

*Table 11.* Performance of all models when trained on only Uniform data. We retain a similar layout to Table 1 but all distributions are considered out-of-distribution in this case.

| | Model | MTMDVRP50 | | | | MTMDVRP100 | | | |
|---|---|---|---|---|---|---|---|---|---|
| | | In-dist | | Out-dist | | In-dist | | Out-dist | |
| | | Obj | Gap | Obj | Gap | Obj | Gap | Obj | Gap |
| In-task | POMO-MTVRP (Uniform) | 6.0932 | 3.8834% | 6.4104 | 4.0007% | 9.5517 | 5.7774% | 10.1878 | 6.1687% |
| | MVMoE (Uniform) | 6.0779 | 3.6000% | 6.3930 | 3.6710% | 9.5065 | 5.2291% | 10.1454 | 5.7632% |
| | MVMoE-Light (Uniform) | 6.0926 | 3.8418% | 6.4061 | 3.8254% | 9.5116 | 5.3037% | 10.1407 | 5.7016% |
| | MVMoE-Deeper (Uniform) | 6.0580 | 3.1964% | 6.3822 | 3.5062% | OOM | OOM | OOM | OOM |
| | SHIELD-MoD (Uniform) | 6.0482 | 3.0379% | 6.3666 | 3.2037% | 9.4120 | 4.1218% | 10.0525 | 4.7131% |
| | SHIELD (Uniform) | **6.0414** | **2.9223%** | **6.3596** | **3.0832%** | **9.3956** | **3.9280%** | **10.0373** | **4.6271%** |
| Out-task | POMO-MTVRP (Uniform) | 5.8762 | 8.1526% | 6.2457 | 8.3681% | 9.5947 | 10.1253% | 10.3081 | 10.6234% |
| | MVMoE (Uniform) | 5.8602 | 7.7505% | 6.2251 | 7.8788% | 9.5514 | 9.4994% | 10.2716 | 10.2298% |
| | MVMoE-Light (Uniform) | 5.8802 | 8.1328% | 6.2414 | 8.0983% | 9.5490 | 9.5566% | 10.2555 | 10.1128% |
| | MVMoE-Deeper (Uniform) | 5.8292 | 7.0524% | 6.2034 | 7.4642% | OOM | OOM | OOM | OOM |
| | SHIELD-MoD (Uniform) | 5.8103 | 6.7257% | 6.1769 | 6.9455% | 9.3977 | 7.6183% | 10.1111 | 8.3284% |
| | SHIELD (Uniform) | **5.8035** | **6.6394%** | **6.1712** | **6.8616%** | **9.3721** | **7.2676%** | **10.0889** | **8.1911%** |

Table 11 displays the performance of all models when trained purely on uniform data. Note that while we retain the same table layout as Table 1, all distributions are considered as out-of-distribution in such a case as the model does not see them at all. Evidently, all models degrade in their predictive performance, even though SHIELD still retains its overall superior performance.

## P. Additional Experiments – Single-task Multi-distribution

*Table 12.* Performance of various models trained on the CVRP task with multiple distributions.

| Model | CVRP50 | | | | CVRP100 | | | |
|---|---|---|---|---|---|---|---|---|
| | In-dist | | Out-dist | | In-dist | | Out-dist | |
| | Obj | Gap | Obj | Gap | Obj | Gap | Obj | Gap |
| POMO-MTVRP | 6.6511 | 1.2260% | 6.9763 | 1.4689% | 9.9795 | 2.3587% | 10.6194 | 3.3445% |
| MVMoE | 6.6454 | 1.1401% | 6.9709 | 1.3858% | 9.9733 | 2.2932% | 10.6189 | 3.2974% |
| MVMoE-Light | 6.6482 | 1.1814% | 6.9723 | 1.4112% | 9.9681 | 2.2398% | 10.6237 | 3.4012% |
| MVMoE-Deeper | 6.6313 | 0.9207% | 6.9628 | 1.2731% | OOM | OOM | OOM | OOM |
| SHIELD-MoD | 6.6284 | 0.8798% | 6.9552 | 1.1623% | 9.9346 | 1.8948% | 10.5545 | 2.6917% |
| SHIELD | **6.6269** | **0.8570%** | **6.9474** | **1.0338%** | **9.9278** | **1.8203%** | **10.5579** | **2.6541%** |

Table 12 displays the performance of various models when trained in a single-task multi-distribution setting. Here, we choose CVRP to be the task at hand. SHIELD remains the best-performing model in such a scenario, suggesting that its architecture is not catered purely to a multi-task multi-distribution problem only.

## Q. Additional Experiments – Multi-Task VRP

*Table 13.* Performance of all models on the MTVRP scenario where all models are trained on the Uniform distribution.

|  | Model | MTVRP50 | | MTVRP100 | |
|---|---|---|---|---|---|
|  |  | Obj | Gap | Obj | Gap |
| In-task | POMO-MTVRP | 10.0470 | 2.9086% | 15.9662 | 4.2795% |
|  | MVMoE | 10.0213 | 2.6279% | 15.8868 | 3.7400% |
|  | MVMoE-Light | 10.0436 | 2.8539% | 15.9182 | 3.9825% |
|  | MVMoE-Deeper | 10.0020 | 2.4281% | OOM | OOM |
|  | SHIELD-MoD | 9.9865 | 2.2522% | 15.8134 | 3.2617% |
|  | SHIELD | **9.9732** | **2.1252%** | **15.7754** | **3.0124%** |
| Out-task | POMO-MTVRP | 10.3023 | 7.1085% | 16.9683 | 8.2123% |
|  | MVMoE | 10.2705 | 6.7095% | 16.8697 | 7.4778% |
|  | MVMoE-Light | 10.3004 | 7.0367% | 16.9036 | 7.8180% |
|  | MVMoE-Deeper | 10.2342 | 6.3488% | OOM | OOM |
|  | SHIELD-MoD | 10.2135 | 6.0721% | 16.7268 | 6.5004% |
|  | SHIELD | **10.1985** | **5.9522%** | **16.6817** | **6.2304%** |

To verify that our architecture improves overall, we trained all models on the MTVRP setting using the uniform distribution. Table 13 showcases the performance of all models. Here, we see that SHIELD is still clearly the better-performing model. Additionally, the gaps between the models are less significant once we remove the varied distributions. This indicates the difficulty of a multi-distribution scenario – having varied structures with multiple tasks is more complex. Since our architecture is more flexible, it generalizes better in the MTMDVRP scenario.

# R. Additional Experiments – Behavior of Scaling During Inference

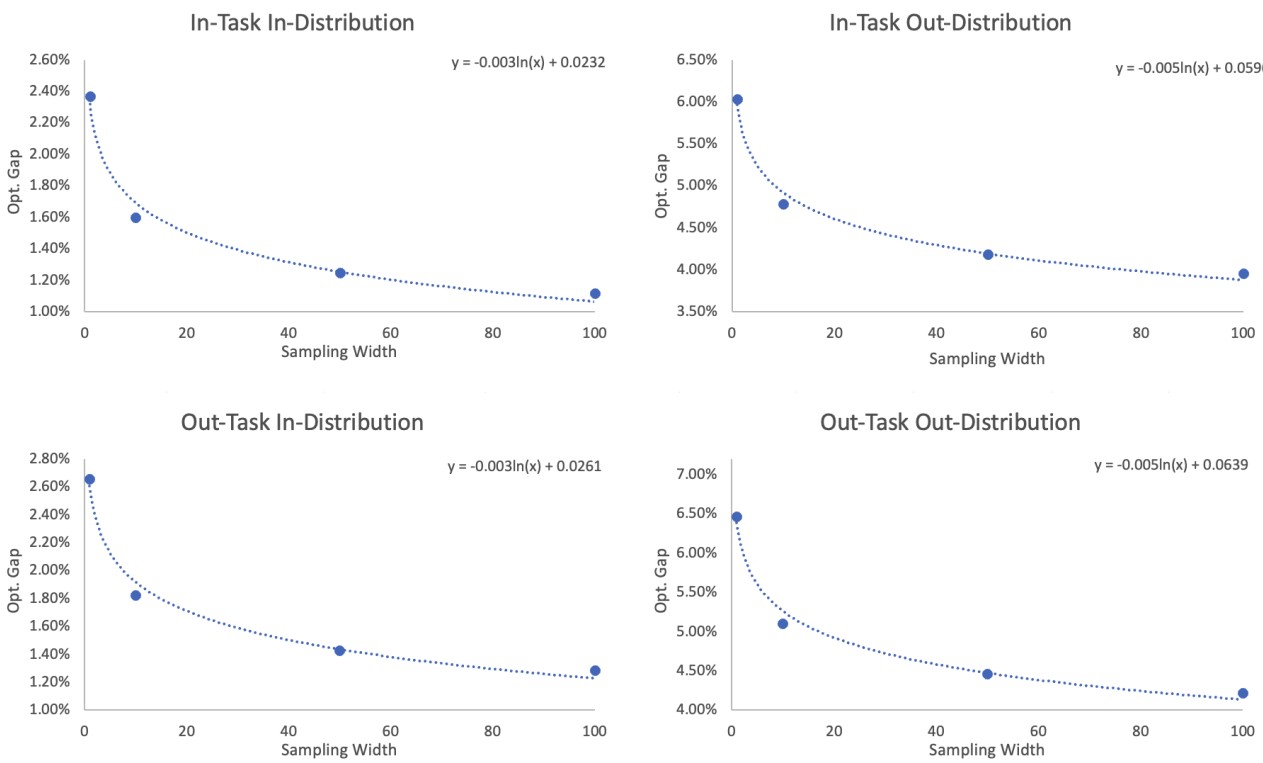

*Figure 6.* Overall performance of SHIELD with varying sampling widths.

For NCO solvers, we can allocate more test time to perform sampling and find better solutions during inference. In this experiment, we reduced the number of test instances to 100 instances per problem and performed inference with sampling widths 1x, 10x, 50x, and 100x. We plot the performance of the various widths are shown in Figure 6. As shown, as we increase the sampling width, the general performance of the model increases (lower gap is better) in a logarithmic fashion. This suggests that while we can allocate more test time for inference, its effectiveness eventually saturates.

# S. Detailed experimental results

*Table 14.* Performance of models on USA13509

**USA13509 — Part 1**

| | Problem | Solver | MTMDVRP50 Obj | Gap | Time | MTMDVRP100 Obj | Gap | Time |
|---|---|---|---|---|---|---|---|---|
| In-task | CVRP | HGS | 7.4382 | - | 1m34s | 11.0281 | - | 2m30s |
| | | POMO-MTVRP | 7.5879 | 2.0132% | 3.22s | 11.3655 | 3.0940% | 8.71s |
| | | MVMoE | 7.5507 | 1.5086% | 4.16s | 11.3352 | 2.8201% | 11.32s |
| | | MVMoE-Light | 7.5570 | 1.5887% | 3.88s | 11.3493 | 2.9397% | 10.65s |
| | | MVMoE-Deeper | 7.5411 | 1.3839% | 9.93s | - | - | 17.93s |
| | | SHIELD-MoD | 7.5295 | 1.2313% | 6.09s | 11.2797 | 2.3098% | 20.33s |
| | | SHIELD | 7.5221 | 1.1229% | 6.69s | 11.2701 | 2.2196% | |
| | OVRP | OR-tools | 4.5943 | - | 1m10s | 6.8727 | - | 2m38s |
| | | POMO-MTVRP | 4.7904 | 4.2675% | 2.31s | 7.2755 | 5.9343% | 7.46s |
| | | MVMoE | 4.7759 | 3.9312% | 3.35s | 7.2222 | 5.1669% | 10.24s |
| | | MVMoE-Light | 4.7868 | 4.2013% | 3.16s | 7.2498 | 5.5529% | 9.38s |
| | | MVMoE-Deeper | 4.7453 | 3.2923% | 8.87s | - | - | 17.00s |
| | | SHIELD-MoD | 4.7290 | 2.9359% | 5.21s | 7.1346 | 3.8933% | 19.46s |
| | | SHIELD | 4.7168 | 2.6767% | 6.15s | 7.0954 | 3.3129% | |
| | VRPB | OR-tools | 5.8325 | - | 1m8s | 8.5742 | - | 2m27s |
| | | POMO-MTVRP | 5.9595 | 2.1771% | 2.16s | 8.7183 | 1.7366% | 6.71s |
| | | MVMoE | 5.9322 | 1.7267% | 3.03s | 8.6799 | 1.2943% | 8.96s |
| | | MVMoE-Light | 5.9474 | 1.9976% | 2.90s | 8.6976 | 1.4965% | 8.28s |
| | | MVMoE-Deeper | 5.9275 | 1.6446% | 7.37s | - | - | 15.12s |
| | | SHIELD-MoD | 5.9091 | 1.3368% | 4.69s | 8.6145 | 0.5298% | 15.75s |
| | | SHIELD | 5.9019 | 1.2105% | 5.26s | 8.5929 | 0.2755% | 16.95s |
| | OVRPB | OR-tools | 4.0952 | - | 1m7s | 5.9434 | - | 2m25s |
| | | POMO-MTVRP | 4.4200 | 7.9311% | 2.28s | 6.5687 | 10.5536% | 6.99s |
| | | MVMoE | 4.4023 | 7.4408% | 3.29s | 6.4989 | 9.3525% | 9.46s |
| | | MVMoE-Light | 4.4276 | 8.0644% | 3.03s | 6.5524 | 10.2667% | 8.69s |
| | | MVMoE-Deeper | 4.3726 | 6.7749% | 8.31s | - | - | 15.75s |
| | | SHIELD-MoD | 4.3544 | 6.3291% | 4.97s | 6.3933 | 7.5796% | 17.57s |
| | | SHIELD | 4.3542 | 6.2802% | 5.69s | 6.3535 | 6.9218% | 17.70s |
| | OVRPL | OR-tools | 4.5923 | - | 1m12s | 6.9599 | - | 2m26s |
| | | POMO-MTVRP | 4.8005 | 4.5344% | 2.37s | 7.3699 | 5.9831% | 7.91s |
| | | MVMoE | 4.7809 | 4.0828% | 3.44s | 7.3170 | 5.2231% | 10.73s |
| | | MVMoE-Light | 4.7959 | 4.4275% | 3.17s | 7.3444 | 5.6140% | 9.79s |
| | | MVMoE-Deeper | 4.7667 | 3.7970% | 9.13s | - | - | 15.81s |
| | | SHIELD-MoD | 4.7383 | 3.1656% | 5.27s | 7.2257 | 3.9050% | 17.57s |
| | | SHIELD | 4.7267 | 2.9169% | 6.13s | 7.1917 | 3.4300% | 17.60s |
| Out-task | VRPBL | OR-tools | 5.8225 | - | 1m11s | 8.5809 | - | 2m22s |
| | | POMO-MTVRP | 5.9697 | 2.5288% | 2.31s | 8.7249 | 1.7249% | 7.40s |
| | | MVMoE | 5.9362 | 1.9635% | 3.24s | 8.6827 | 1.2302% | 9.64s |
| | | MVMoE-Light | 5.9479 | 2.1643% | 3.06s | 8.7017 | 1.4516% | 8.91s |
| | | MVMoE-Deeper | 5.9425 | 2.0614% | 7.78s | - | - | 15.81s |
| | | SHIELD-MoD | 5.9058 | 1.4473% | 4.77s | 8.6238 | 0.5440% | 17.60s |
| | | SHIELD | 5.8952 | 1.2590% | 5.37s | 8.5988 | 0.2555% | 20.10s |
| | VRPBTW | OR-tools | 9.2271 | - | 1m14s | 15.4369 | - | 2m40s |
| | | POMO-MTVRP | 10.1169 | 9.6432% | 2.75s | 16.6446 | 8.0943% | 8.58s |
| | | MVMoE | 10.0372 | 8.8942% | 3.88s | 16.6193 | 7.9169% | 11.39s |
| | | MVMoE-Light | 10.0612 | 9.1353% | 3.61s | 16.6460 | 8.0804% | 10.69s |
| | | MVMoE-Deeper | 10.0460 | 8.8750% | 9.58s | - | - | 15.81s |
| | | SHIELD-MoD | 10.0014 | 8.4870% | 5.62s | 16.4699 | 6.9561% | 17.99s |
| | | SHIELD | 9.9621 | 8.0901% | 6.24s | 16.3999 | 6.4802% | 20.10s |
| | VRPLTW | OR-tools | 9.2840 | - | 1m14s | 15.8230 | - | 2m36s |
| | | POMO-MTVRP | 9.6555 | 4.0016% | 2.81s | 16.2667 | 3.0087% | 9.90s |
| | | MVMoE | 9.6052 | 3.5599% | 3.92s | 16.2093 | 2.6499% | 12.93s |
| | | MVMoE-Light | 9.6202 | 3.6970% | 3.70s | 16.2532 | 2.9117% | 12.05s |
| | | MVMoE-Deeper | 9.6122 | 3.5352% | 10.2s | - | - | 20.01s |
| | | SHIELD-MoD | 9.5605 | 3.0615% | 5.74s | 16.0760 | 1.8171% | 20.01s |
| | | SHIELD | 9.5422 | 2.8639% | 6.52s | 16.0317 | 1.5234% | 22.48s |

**USA13509 — Part 2**

| Problem | Solver | MTMDVRP50 Obj | Gap | Time | MTMDVRP100 Obj | Gap | Time |
|---|---|---|---|---|---|---|---|
| VRPL | OR-tools | 7.5719 | - | 1m10s | 11.5478 | - | 2m40s |
| | POMO-MTVRP | 7.6238 | 0.6848% | 2.38s | 11.4109 | -1.1490% | 7.96s |
| | MVMoE | 7.5835 | 0.1899% | 3.28s | 11.3828 | -1.3885% | 10.71s |
| | MVMoE-Light | 7.5922 | 0.2923% | 3.07s | 11.3988 | -1.2536% | 9.82s |
| | MVMoE-Deeper | 7.5709 | 0.0221% | 8.56s | - | - | 17.29s |
| | SHIELD-MoD | 7.5612 | -0.1102% | 5.02s | 11.3256 | -1.8875% | 19.70s |
| | SHIELD | 7.5547 | -0.1943% | 5.79s | 11.3108 | -2.0178% | |
| VRPTW | OR-tools | 9.2000 | - | 1m19s | 14.9649 | - | 6m36s |
| | POMO-MTVRP | 9.7027 | 5.4640% | 2.82s | 16.1244 | 7.7828% | 9.14s |
| | MVMoE | 9.6755 | 5.1134% | 3.77s | 16.0672 | 7.3912% | 12.07s |
| | MVMoE-Light | 9.6941 | 5.3042% | 3.53s | 16.1132 | 7.6911% | 11.37s |
| | MVMoE-Deeper | 9.6401 | 4.7493% | 9.97s | - | - | 19.06s |
| | SHIELD-MoD | 9.6332 | 4.6368% | 5.71s | 15.9441 | 6.5794% | 19.06s |
| | SHIELD | 9.6035 | 4.3388% | 6.50s | 15.8862 | 6.1796% | 21.62s |
| OVRPTW | OR-tools | 5.9178 | - | 1m15s | 9.4305 | - | 2m43s |
| | POMO-MTVRP | 6.2256 | 5.2015% | 2.75s | 10.0576 | 6.7325% | 8.82s |
| | MVMoE | 6.2274 | 5.1953% | 3.91s | 10.0188 | 6.3105% | 11.80s |
| | MVMoE-Light | 6.2421 | 5.4416% | 3.56s | 10.0490 | 6.6435% | 10.85s |
| | MVMoE-Deeper | 6.1870 | 4.5388% | 10.32s | - | - | 19.04s |
| | SHIELD-MoD | 6.1663 | 4.1985% | 5.86s | 9.8820 | 4.8706% | 19.04s |
| | SHIELD | 6.1656 | 4.1838% | 6.74s | 9.8557 | 4.5972% | 21.67s |
| OVRPBL | OR-tools | 4.0893 | - | 1m9s | 5.9119 | - | 2m39s |
| | POMO-MTVRP | 4.4099 | 7.8395% | 2.35s | 6.5330 | 10.5271% | 7.41s |
| | MVMoE | 4.3940 | 7.4030% | 3.39s | 6.4591 | 9.2685% | 9.86s |
| | MVMoE-Light | 4.4193 | 8.0245% | 3.12s | 6.5207 | 10.3082% | 9.05s |
| | MVMoE-Deeper | 4.3668 | 6.7854% | 8.32s | - | - | 16.12s |
| | SHIELD-MoD | 4.3469 | 6.2984% | 5.05s | 6.3605 | 7.6109% | 18.05s |
| | SHIELD | 4.3464 | 6.2536% | 5.77s | 6.3191 | 6.9081% | 18.05s |
| OVRPBTW | OR-tools | 5.8937 | - | 1m12s | 9.3848 | - | 2m45s |
| | POMO-MTVRP | 6.5213 | 10.6484% | 2.83s | 10.5023 | 12.0111% | 8.58s |
| | MVMoE | 6.5206 | 10.5319% | 3.86s | 10.4519 | 11.4622% | 11.21s |
| | MVMoE-Light | 6.5279 | 10.6596% | 3.75s | 10.4753 | 11.7195% | 10.41s |
| | MVMoE-Deeper | 6.4880 | 10.0836% | 10.09s | - | - | 16.12s |
| | SHIELD-MoD | 6.4706 | 9.6962% | 5.89s | 10.3176 | 10.0337% | 18.34s |
| | SHIELD | 6.4546 | 9.4721% | 6.72s | 10.2672 | 9.5015% | 20.58s |
| OVRPLTW | OR-tools | 5.8319 | - | 1m15s | 9.4364 | - | 2m55s |
| | POMO-MTVRP | 6.1337 | 5.1747% | 3.15s | 10.0586 | 6.6666% | 9.24s |
| | MVMoE | 6.1313 | 5.0931% | 4.02s | 10.0215 | 6.2699% | 12.22s |
| | MVMoE-Light | 6.1423 | 5.2860% | 3.74s | 10.0512 | 6.5819% | 11.34s |
| | MVMoE-Deeper | 6.1102 | 4.7727% | 10.61s | - | - | 19.54s |
| | SHIELD-MoD | 6.0811 | 4.2453% | 6.02s | 9.8844 | 4.8210% | 19.54s |
| | SHIELD | 6.0750 | 4.1655% | 6.93s | 9.8580 | 4.5416% | 22.05s |
| VRPBLTW | OR-tools | 9.0613 | - | 1m22s | 15.4038 | - | 2m49s |
| | POMO-MTVRP | 9.9196 | 9.4717% | 3.28s | 16.6092 | 8.0791% | 9.20s |
| | MVMoE | 9.8671 | 8.9353% | 3.92s | 16.5711 | 7.8168% | 11.82s |
| | MVMoE-Light | 9.8688 | 8.9561% | 3.64s | 16.6005 | 8.0151% | 11.26s |
| | MVMoE-Deeper | 9.8627 | 8.8446% | 9.85s | - | - | 18.61s |
| | SHIELD-MoD | 9.8222 | 8.4543% | 5.73s | 16.4395 | 6.9872% | 18.61s |
| | SHIELD | 9.7819 | 7.9903% | 6.34s | 16.3696 | 6.4873% | 20.74s |
| OVRPBLTW | OR-tools | 5.8173 | - | 1m18s | 9.4629 | - | 2m38s |
| | POMO-MTVRP | 6.4355 | 10.6271% | 3.18s | 10.5841 | 11.9099% | 8.92s |
| | MVMoE | 6.4230 | 10.3227% | 3.97s | 10.5359 | 11.3903% | 11.57s |
| | MVMoE-Light | 6.4289 | 10.4245% | 3.84s | 10.5717 | 11.7672% | 10.84s |
| | MVMoE-Deeper | 6.3975 | 9.9740% | 10.25s | - | - | 18.64s |
| | SHIELD-MoD | 6.3813 | 9.6120% | 5.99s | 10.4016 | 9.9961% | 18.64s |
| | SHIELD | 6.3687 | 9.4311% | 6.82s | 10.3608 | 9.5539% | 20.80s |

*Table 15.* Performance of models on JA9847

**JA9847 — Left half**

| | Problem | Solver | MTMDVRP50 Obj | MTMDVRP50 Gap | MTMDVRP50 Time | MTMDVRP100 Obj | MTMDVRP100 Gap | MTMDVRP100 Time |
|---|---|---|---|---|---|---|---|---|
| In-task | CVRP | HGS | 5.9347 | - | 1m21s | 8.7045 | - | 2m12s |
| | | POMO-MTVRP | 5.9686 | 1.8080% | 3.15s | 8.9352 | 2.7145% | 8.67s |
| | | MVMoE | 5.9429 | 1.3723% | 4.28s | 8.9055 | 2.3673% | 11.60s |
| | | MVMoE-Light | 5.9479 | 1.4661% | 4.44s | 8.9223 | 2.5692% | 10.59s |
| | | MVMoE-Deeper | 5.9328 | 1.2084% | 9.66s | - | - | - |
| | | SHIELD-MoD | 5.9249 | 1.0679% | 5.96s | 8.8645 | 1.8902% | 18.02s |
| | | SHIELD | 5.9207 | 0.9989% | 6.63s | 8.8611 | 1.8524% | 20.42s |
| | OVRP | OR-tools | 3.3709 | - | 1m8s | 5.1676 | - | 2m40s |
| | | POMO-MTVRP | 3.5699 | 5.9032% | 2.33s | 5.5171 | 6.9041% | 7.55s |
| | | MVMoE | 3.5610 | 5.6759% | 3.63s | 5.4689 | 5.9189% | 10.99s |
| | | MVMoE-Light | 3.5860 | 6.4499% | 3.22s | 5.4955 | 6.4637% | 9.31s |
| | | MVMoE-Deeper | 3.5076 | 4.0898% | 8.54s | - | - | - |
| | | SHIELD-MoD | 3.4963 | 3.7566% | 5.32s | 5.3515 | 3.6637% | 17.31s |
| | | SHIELD | 3.4910 | 3.6111% | 6.08s | 5.3300 | 3.2637% | 19.51s |
| | VRPB | OR-tools | 4.4164 | - | 1m3s | 6.4448 | - | 2m36s |
| | | POMO-MTVRP | 4.5219 | 2.3878% | 2.13s | 6.5417 | 1.6225% | 6.69s |
| | | MVMoE | 4.4959 | 1.8598% | 3.23s | 6.5100 | 1.1357% | 8.99s |
| | | MVMoE-Light | 4.5026 | 2.0176% | 2.91s | 6.5309 | 1.4577% | 8.25s |
| | | MVMoE-Deeper | 4.4856 | 1.6420% | 7.02s | - | - | - |
| | | SHIELD-MoD | 4.4747 | 1.3822% | 4.69s | 6.4567 | 0.3003% | 15.08s |
| | | SHIELD | 4.4667 | 1.2035% | 5.28s | 6.4494 | 0.1965% | 16.93s |
| | OVRPB | OR-tools | 2.6854 | - | 1m11s | 3.9796 | - | 2m39s |
| | | POMO-MTVRP | 2.9943 | 11.5013% | 2.25s | 4.5688 | 15.0151% | 7.02s |
| | | MVMoE | 2.9814 | 10.9755% | 3.63s | 4.5028 | 13.2704% | 9.56s |
| | | MVMoE-Light | 3.0220 | 12.5473% | 3.01s | 4.5577 | 14.6921% | 8.61s |
| | | MVMoE-Deeper | 2.9348 | 9.2875% | 8.1s | - | - | - |
| | | SHIELD-MoD | 2.9243 | 8.8657% | 4.91s | 4.3780 | 10.1884% | 15.83s |
| | | SHIELD | 2.9195 | 8.7000% | 5.65s | 4.3530 | 9.5759% | 17.64s |
| Out-task | OVRPL | OR-tools | 3.3761 | - | 1m14s | 5.1001 | - | 2m55s |
| | | POMO-MTVRP | 3.5789 | 6.0070% | 2.36s | 5.4586 | 7.1472% | 8.00s |
| | | MVMoE | 3.5694 | 5.6644% | 3.78s | 5.4084 | 6.1290% | 11.59s |
| | | MVMoE-Light | 3.5895 | 6.4082% | 3.02s | 5.4262 | 6.4924% | 9.77s |
| | | MVMoE-Deeper | 3.5249 | 4.4067% | 9.01s | - | - | - |
| | | SHIELD-MoD | 3.5010 | 3.7652% | 5.25s | 5.2866 | 3.7530% | 15.77s |
| | | SHIELD | 3.4964 | 3.6483% | 6.12s | 5.2775 | 3.5944% | 17.64s |
| | VRPBL | OR-tools | 4.3894 | - | 1m13s | 6.4010 | - | 2m49s |
| | | POMO-MTVRP | 4.4933 | 2.3667% | 2.25s | 6.4699 | 1.1785% | 9.64s |
| | | MVMoE | 4.4657 | 1.7842% | 3.63s | 6.4997 | 1.6473% | 7.38s |
| | | MVMoE-Light | 4.4737 | 1.9700% | 3.59s | 6.4901 | 1.4975% | 8.87s |
| | | MVMoE-Deeper | 4.4728 | 1.9007% | 8.69s | - | - | - |
| | | SHIELD-MoD | 4.4469 | 1.3514% | 4.75s | 6.4182 | 0.3668% | 15.77s |
| | | SHIELD | 4.4357 | 1.1004% | 5.35s | 6.4115 | 0.002659 | 17.64s |
| | VRPBTW | OR-tools | 6.7862 | - | 1m20s | 11.8462 | - | 2m42s |
| | | POMO-MTVRP | 7.3740 | 8.6621% | 2.73s | 12.6045 | 6.8206% | 8.52s |
| | | MVMoE | 7.3203 | 8.1467% | 4.28s | 12.5412 | 6.2745% | 11.42s |
| | | MVMoE-Light | 7.3267 | 8.2462% | 3.59s | 12.5954 | 6.7859% | 10.59s |
| | | MVMoE-Deeper | 7.3205 | 7.8732% | 9.44s | - | - | - |
| | | SHIELD-MoD | 7.2765 | 7.4651% | 5.57s | 12.4460 | 5.4695% | 17.90s |
| | | SHIELD | 7.2522 | 7.1540% | 6.12s | 12.3966 | 5.0097% | 20.18s |
| | VRPLTW | OR-tools | 7.0420 | - | 1m24s | 12.0881 | - | 2m50s |
| | | POMO-MTVRP | 7.3305 | 4.0966% | 2.84s | 12.3506 | 2.5592% | 9.92s |
| | | MVMoE | 7.2767 | 3.5074% | 4.22s | 12.3045 | 2.1596% | 12.99s |
| | | MVMoE-Light | 7.2805 | 3.5980% | 3.67s | 12.3385 | 2.4430% | 12.01s |
| | | MVMoE-Deeper | 7.2765 | 3.3301% | 10.02s | - | - | - |
| | | SHIELD-MoD | 7.2300 | 2.8305% | 5.71s | 12.1935 | 1.2312% | 19.96s |
| | | SHIELD | 7.2230 | 2.7590% | 6.50s | 12.1775 | 1.0937% | 22.52s |

**JA9847 — Right half**

| Problem | Solver | MTMDVRP50 Obj | MTMDVRP50 Gap | MTMDVRP50 Time | MTMDVRP100 Obj | MTMDVRP100 Gap | MTMDVRP100 Time |
|---|---|---|---|---|---|---|---|
| VRPL | OR-tools | 5.9291 | - | 1m9s | 8.9750 | - | 2m39s |
| | POMO-MTVRP | 5.9665 | 0.6312% | 2.35s | 8.8637 | -1.2338% | 8.00s |
| | MVMoE | 5.9350 | 0.1612% | 3.58s | 8.8337 | -1.5670% | 11.07s |
| | MVMoE-Light | 5.9393 | 0.2371% | 3.07s | 8.8521 | -1.3629% | 9.86s |
| | MVMoE-Deeper | 5.9257 | 0.0021% | 8.34s | - | - | - |
| | SHIELD-MoD | 5.9177 | -0.1350% | 5.02s | 8.7924 | -2.0387% | 17.32s |
| | SHIELD | 5.9140 | -0.2034% | 5.79s | 8.7904 | -2.0540% | 19.69s |
| VRPTW | OR-tools | 6.9905 | - | 1m18s | 11.3101 | - | 6m33s |
| | POMO-MTVRP | 7.2449 | 5.9169% | 2.80s | 12.1846 | 7.9817% | 9.14s |
| | MVMoE | 7.2083 | 5.4147% | 4.21s | 12.1293 | 7.4828% | 12.19s |
| | MVMoE-Light | 7.2174 | 5.5424% | 3.53s | 12.1740 | 7.8991% | 11.34s |
| | MVMoE-Deeper | 7.1708 | 4.9030% | 9.88s | - | - | - |
| | SHIELD-MoD | 7.1688 | 4.7880% | 5.66s | 12.0365 | 6.6742% | 19.06s |
| | SHIELD | 7.1579 | 4.7112% | 6.43s | 12.0109 | 6.4377% | 21.62s |
| OVRPTW | OR-tools | 4.1882 | - | 1m12s | 6.7764 | - | 2m44s |
| | POMO-MTVRP | 4.4770 | 6.8958% | 2.69s | 7.2570 | 7.3553% | 8.79s |
| | MVMoE | 4.4652 | 6.6707% | 4.25s | 7.2159 | 6.7570% | 12.29s |
| | MVMoE-Light | 4.4809 | 7.0366% | 3.51s | 7.2523 | 7.2667% | 10.92s |
| | MVMoE-Deeper | 4.4289 | 5.8712% | 10.29s | - | - | - |
| | SHIELD-MoD | 4.4245 | 5.6412% | 5.71s | 7.0928 | 4.9476% | 18.80s |
| | SHIELD | 4.4182 | 5.5804% | 6.58s | 7.0919 | 4.9260% | 21.33s |
| OVRPBL | OR-tools | 2.7264 | - | 1m7s | 3.9870 | - | 2m35s |
| | POMO-MTVRP | 3.0326 | 11.2299% | 2.37s | 4.5780 | 15.0175% | 7.40s |
| | MVMoE | 3.0226 | 10.8299% | 3.69s | 4.5206 | 13.5069% | 10.00s |
| | MVMoE-Light | 3.0648 | 12.4295% | 3.11s | 4.5720 | 14.8258% | 8.94s |
| | MVMoE-Deeper | 2.9763 | 9.1647% | 8.22s | - | - | - |
| | SHIELD-MoD | 2.9640 | 8.7163% | 5.04s | 4.3889 | 10.2606% | 16.27s |
| | SHIELD | 2.9638 | 8.6944% | 5.70s | 4.3578 | 9.4704% | 18.05s |
| OVRPBTW | OR-tools | 4.1148 | - | 1m15s | 6.8126 | - | 2m41s |
| | POMO-MTVRP | 4.5988 | 11.7626% | 2.80s | 7.5900 | 11.7303% | 8.45s |
| | MVMoE | 4.5813 | 11.3687% | 4.20s | 7.5476 | 11.1123% | 11.20s |
| | MVMoE-Light | 4.5881 | 11.5339% | 3.67s | 7.5759 | 11.5058% | 10.45s |
| | MVMoE-Deeper | 4.5493 | 10.5600% | 9.91s | - | - | - |
| | SHIELD-MoD | 4.5259 | 10.0152% | 5.74s | 7.4264 | 9.3600% | 17.94s |
| | SHIELD | 4.5198 | 9.9372% | 6.52s | 7.3886 | 8.7518% | 20.13s |
| OVRPLTW | OR-tools | 4.1520 | - | 1m17s | 6.8440 | - | 2m51s |
| | POMO-MTVRP | 4.4365 | 6.8511% | 2.80s | 7.3216 | 7.3075% | 9.15s |
| | MVMoE | 4.4265 | 6.6381% | 4.20s | 7.2802 | 6.6918% | 12.62s |
| | MVMoE-Light | 4.4423 | 7.0211% | 3.68s | 7.3130 | 7.1614% | 11.48s |
| | MVMoE-Deeper | 4.4021 | 6.0241% | 10.39s | - | - | - |
| | SHIELD-MoD | 4.3809 | 5.5133% | 5.88s | 7.1516 | 4.8308% | 19.20s |
| | SHIELD | 4.3734 | 5.4095% | 6.75s | 7.1484 | 4.7587% | 21.69s |
| VRPBLTW | OR-tools | 6.8945 | - | 1m22s | 12.1613 | - | 2m46s |
| | POMO-MTVRP | 7.4997 | 8.7784% | 2.95s | 12.9318 | 6.8330% | 9.23s |
| | MVMoE | 7.4382 | 8.1638% | 4.18s | 12.8821 | 6.3975% | 12.08s |
| | MVMoE-Light | 7.4476 | 8.2867% | 3.59s | 12.9371 | 6.8322% | 11.29s |
| | MVMoE-Deeper | 7.4418 | 7.9377% | 9.76s | - | - | - |
| | SHIELD-MoD | 7.3976 | 7.5407% | 5.65s | 12.7737 | 5.5222% | 18.69s |
| | SHIELD | 7.3750 | 7.2483% | 6.26s | 12.7383 | 5.1504% | 20.94s |
| OVRPBLTW | OR-tools | 4.0716 | - | 1m19s | 6.8237 | - | 2m33s |
| | POMO-MTVRP | 4.5587 | 11.9622% | 2.85s | 7.6099 | 11.8026% | 8.82s |
| | MVMoE | 4.5427 | 11.5344% | 4.33s | 7.5505 | 10.9459% | 11.62s |
| | MVMoE-Light | 4.5505 | 11.7197% | 3.72s | 7.5902 | 11.5239% | 10.79s |
| | MVMoE-Deeper | 4.5088 | 10.7374% | 10.03s | - | - | - |
| | SHIELD-MoD | 4.4880 | 10.1687% | 5.82s | 7.4267 | 9.1647% | 18.30s |
| | SHIELD | 4.4808 | 10.0961% | 6.56s | 7.4043 | 8.7984% | 20.45s |

*Table 16.* Performance of models on BM33708

| Group | Problem | Solver | MTMDVRP50 Obj | MTMDVRP50 Gap | MTMDVRP50 Time | MTMDVRP100 Obj | MTMDVRP100 Gap | MTMDVRP100 Time |
|---|---|---|---|---|---|---|---|---|
| In-task | CVRP | HGS | 6.5032 | - | 1m25s | 9.5205 | - | 2m11s |
| | | POMO-MTVRP | 6.5373 | 1.9983% | 3.28s | 9.8019 | 2.9725% | 8.71s |
| | | MVMoE | 6.5072 | 1.5219% | 4.46s | 9.7728 | 2.6616% | 11.44s |
| | | MVMoE-Light | 6.5137 | 1.6263% | 3.87s | 9.7950 | 2.8929% | 10.58s |
| | | MVMoE-Deeper | 6.4984 | 1.3845% | 9.58s | - | - | - |
| | | SHIELD-MoD | 6.4897 | 1.2503% | 5.76s | 9.7312 | 2.2300% | 18.39s |
| | | SHIELD | 6.4843 | 1.1648% | 6.47s | 9.7160 | 2.0607% | 20.45s |
| | OVRP | OR-tools | 3.9920 | - | 1m12s | 5.9998 | - | 2m39s |
| | | POMO-MTVRP | 4.1641 | 4.3105% | 2.59s | 6.3549 | 5.9698% | 7.47s |
| | | MVMoE | 4.1523 | 3.9851% | 3.67s | 6.3028 | 5.0992% | 10.18s |
| | | MVMoE-Light | 4.1634 | 4.2685% | 3.08s | 6.3316 | 5.5745% | 9.39s |
| | | MVMoE-Deeper | 4.1270 | 3.3616% | 8.62s | - | - | - |
| | | SHIELD-MoD | 4.1111 | 2.9609% | 5.21s | 6.2323 | 3.9265% | 17.05s |
| | | SHIELD | 4.1012 | 2.7202% | 5.94s | 6.1989 | 3.3706% | 19.51s |
| | VRPB | OR-tools | 5.1214 | - | 1m8s | 7.5311 | - | 2m37s |
| | | POMO-MTVRP | 5.2323 | 2.1659% | 2.30s | 7.6426 | 1.5415% | 6.72s |
| | | MVMoE | 5.2088 | 1.7189% | 3.08s | 7.6019 | 1.0042% | 8.96s |
| | | MVMoE-Light | 5.2203 | 1.9473% | 3.04s | 7.6283 | 1.3498% | 8.27s |
| | | MVMoE-Deeper | 5.1985 | 1.5129% | 7.03s | - | - | - |
| | | SHIELD-MoD | 5.1872 | 1.2899% | 4.70s | 7.5605 | 0.4521% | 15.07s |
| | | SHIELD | 5.1774 | 1.1052% | 5.24s | 7.5432 | 0.2217% | 16.96s |
| | OVRPB | OR-tools | 3.5304 | - | 1m14s | 5.1150 | - | 2m40s |
| | | POMO-MTVRP | 3.8075 | 7.8483% | 2.45s | 5.6545 | 10.5678% | 6.98s |
| | | MVMoE | 3.7854 | 7.1879% | 3.30s | 5.5802 | 9.1051% | 9.62s |
| | | MVMoE-Light | 3.8044 | 7.7261% | 3.23s | 5.6390 | 10.2535% | 8.66s |
| | | MVMoE-Deeper | 3.7667 | 6.6935% | 7.98s | - | - | - |
| | | SHIELD-MoD | 3.7513 | 6.2563% | 4.96s | 5.4924 | 7.4002% | 15.71s |
| | | SHIELD | 3.7476 | 6.1222% | 5.59s | 5.4591 | 6.7559% | 17.68s |
| | OVRPL | OR-tools | 3.9981 | - | 1m18s | 5.9357 | - | 2m53s |
| | | POMO-MTVRP | 4.1679 | 4.5092% | 2.42s | 6.2854 | 5.9394% | 7.87s |
| | | MVMoE | 4.1430 | 3.8587% | 3.26s | 6.2359 | 5.1059% | 10.72s |
| | | MVMoE-Light | 4.1571 | 4.2170% | 3.06s | 6.2623 | 5.5449% | 9.83s |
| | | MVMoE-Deeper | 4.1379 | 3.7570% | 9.04s | - | - | - |
| | | SHIELD-MoD | 4.1054 | 2.9310% | 5.21s | 6.1603 | 3.8383% | 17.47s |
| | | SHIELD | 4.0957 | 2.6807% | 6.02s | 6.1346 | 3.4063% | 19.90s |
| | VRPL | OR-tools | 6.5389 | - | 1m15s | 9.9236 | - | 2m37s |
| | | POMO-MTVRP | 6.5722 | 0.5091% | 2.51s | 9.8052 | -1.1388% | 7.97s |
| | | MVMoE | 6.5382 | 0.0282% | 3.35s | 9.7744 | -1.4515% | 10.71s |
| | | MVMoE-Light | 6.5459 | 0.1420% | 3.39s | 9.7952 | -1.2381% | 9.83s |
| | | MVMoE-Deeper | 6.5289 | -0.1156% | 8.41s | - | - | - |
| | | SHIELD-MoD | 6.5195 | -0.2635% | 5.02s | 9.7327 | -1.8684% | 17.31s |
| | | SHIELD | 6.5148 | -0.3338% | 5.76s | 9.7200 | -2.0014% | 19.70s |
| | VRPTW | OR-tools | 7.6658 | - | 1m22s | 12.0249 | - | 6m25s |
| | | POMO-MTVRP | 7.9788 | 5.7045% | 2.96s | 12.9875 | 8.0100% | 9.00s |
| | | MVMoE | 7.9591 | 5.3968% | 3.86s | 12.9415 | 7.6221% | 12.20s |
| | | MVMoE-Light | 7.9703 | 5.5481% | 3.76s | 12.9698 | 7.8572% | 11.18s |
| | | MVMoE-Deeper | 7.9357 | 5.0955% | 9.86s | - | - | - |
| | | SHIELD-MoD | 7.9158 | 4.8116% | 5.66s | 12.8390 | 6.7800% | 18.72s |
| | | SHIELD | 7.9004 | 4.6212% | 6.39s | 12.7998 | 6.4508% | 21.21s |
| | OVRPTW | OR-tools | 5.0201 | - | 1m16s | 8.0463 | - | 2m45s |
| | | POMO-MTVRP | 5.2763 | 5.1035% | 2.90s | 8.5785 | 6.6527% | 8.82s |
| | | MVMoE | 5.2834 | 5.1907% | 3.89s | 8.5565 | 6.1197% | 11.72s |
| | | MVMoE-Light | 5.2918 | 5.3632% | 3.73s | 8.5722 | 6.5587% | 10.85s |
| | | MVMoE-Deeper | 5.2542 | 4.6315% | 10.23s | - | - | - |
| | | SHIELD-MoD | 5.2383 | 4.2937% | 5.83s | 8.4346 | 4.8748% | 19.04s |
| | | SHIELD | 5.2333 | 4.2265% | 6.68s | 8.4155 | 4.6225% | 21.62s |
| | OVRPBL | OR-tools | 3.5357 | - | 1m20s | 5.1156 | - | 2m34s |
| | | POMO-MTVRP | 3.8204 | 8.0509% | 3.10s | 5.6544 | 10.5263% | 7.38s |
| | | MVMoE | 3.7958 | 7.2984% | 3.41s | 5.5812 | 9.1010% | 9.81s |
| | | MVMoE-Light | 3.8204 | 7.9910% | 3.11s | 5.6376 | 10.2014% | 9.02s |
| | | MVMoE-Deeper | 3.7805 | 6.9246% | 8.33s | - | - | - |
| | | SHIELD-MoD | 3.7656 | 6.5012% | 5.04s | 5.4930 | 7.3890% | 16.09s |
| | | SHIELD | 3.7616 | 6.3349% | 5.77s | 5.4609 | 6.7607% | 18.04s |
| | OVRPBTW | OR-tools | 4.9702 | - | 1m20s | 7.9711 | - | 2m44s |
| | | POMO-MTVRP | 5.4885 | 10.4286% | 3.47s | 8.9031 | 11.7278% | 8.53s |
| | | MVMoE | 5.4754 | 10.1045% | 3.94s | 8.8646 | 11.2500% | 11.16s |
| | | MVMoE-Light | 5.4791 | 10.2090% | 3.75s | 8.8888 | 11.5506% | 10.34s |
| | | MVMoE-Deeper | 5.4665 | 9.9856% | 10.01s | - | - | - |
| | | SHIELD-MoD | 5.4432 | 9.4653% | 5.84s | 8.7594 | 9.9495% | 18.19s |
| | | SHIELD | 5.4334 | 9.2902% | 6.64s | 8.7285 | 9.5463% | 20.42s |
| Out-task | VRPBL | OR-tools | 5.1312 | - | 1m16s | 7.5768 | - | 2m35s |
| | | POMO-MTVRP | 5.2568 | 2.4473% | 2.53s | 7.6932 | 1.5959% | 7.38s |
| | | MVMoE | 5.2191 | 1.7541% | 3.46s | 7.6563 | 1.1044% | 9.59s |
| | | MVMoE-Light | 5.2303 | 1.9696% | 3.06s | 7.6777 | 1.3801% | 8.90s |
| | | MVMoE-Deeper | 5.2357 | 2.0373% | 8.81s | - | - | - |
| | | SHIELD-MoD | 5.1972 | 1.3226% | 4.74s | 7.6094 | 0.4846% | 15.75s |
| | | SHIELD | 5.1873 | 1.1250% | 5.34s | 7.5898 | 0.2250% | 17.67s |
| | VRPBTW | OR-tools | 7.4449 | - | 1m21s | 12.4088 | - | 2m35s |
| | | POMO-MTVRP | 8.1811 | 9.8882% | 3.17s | 13.4097 | 8.2633% | 8.46s |
| | | MVMoE | 8.1267 | 9.2211% | 3.89s | 13.3707 | 7.9318% | 11.23s |
| | | MVMoE-Light | 8.1238 | 9.1848% | 3.59s | 13.4078 | 8.2289% | 10.47s |
| | | MVMoE-Deeper | 8.1340 | 9.2566% | 9.53s | - | - | - |
| | | SHIELD-MoD | 8.0782 | 8.5745% | 5.57s | 13.2689 | 7.1352% | 17.61s |
| | | SHIELD | 8.0547 | 8.2630% | 6.17s | 13.2026 | 6.5746% | 19.68s |
| | VRPLTW | OR-tools | 7.6281 | - | 1m33s | 12.4766 | - | 2m47s |
| | | POMO-MTVRP | 7.9617 | 4.3739% | 3.50s | 12.8902 | 3.4968% | 9.72s |
| | | MVMoE | 7.9210 | 3.8770% | 3.91s | 12.8550 | 3.1971% | 12.71s |
| | | MVMoE-Light | 7.9339 | 4.0594% | 3.71s | 12.8826 | 3.4163% | 11.81s |
| | | MVMoE-Deeper | 7.9339 | 4.0089% | 10.1s | - | - | - |
| | | SHIELD-MoD | 7.8808 | 3.3677% | 5.68s | 12.7513 | 2.3939% | 19.45s |
| | | SHIELD | 7.8676 | 3.1980% | 6.43s | 12.7150 | 2.0844% | 21.88s |
| | OVRPLTW | OR-tools | 4.9822 | - | 1m30s | 8.0416 | - | 2m48s |
| | | POMO-MTVRP | 5.2483 | 5.3405% | 3.51s | 8.5824 | 6.7651% | 9.23s |
| | | MVMoE | 5.2444 | 5.2144% | 4.04s | 8.5412 | 6.2512% | 12.14s |
| | | MVMoE-Light | 5.2563 | 5.4492% | 3.73s | 8.5730 | 6.6407% | 11.32s |
| | | MVMoE-Deeper | 5.2335 | 5.0438% | 10.5s | - | - | - |
| | | SHIELD-MoD | 5.2019 | 4.3630% | 6.01s | 8.4339 | 4.9271% | 19.43s |
| | | SHIELD | 5.1979 | 4.3038% | 6.88s | 8.4186 | 4.7310% | 22.07s |
| | VRPBLTW | OR-tools | 7.4143 | - | 1m40s | 12.4970 | - | 2m41s |
| | | POMO-MTVRP | 8.1315 | 9.6728% | 3.73s | 13.5021 | 8.2643% | 9.07s |
| | | MVMoE | 8.0779 | 8.9961% | 3.90s | 13.4496 | 7.8306% | 11.85s |
| | | MVMoE-Light | 8.0827 | 9.0685% | 3.62s | 13.4871 | 8.1257% | 11.09s |
| | | MVMoE-Deeper | 8.0907 | 9.1229% | 9.82s | - | - | - |
| | | SHIELD-MoD | 8.0423 | 8.5104% | 5.67s | 13.3439 | 7.0136% | 18.27s |
| | | SHIELD | 8.0099 | 8.0833% | 6.29s | 13.2892 | 6.5421% | 20.34s |
| | OVRPBLTW | OR-tools | 4.9601 | - | 1m32s | 8.0296 | - | 2m34s |
| | | POMO-MTVRP | 5.4895 | 10.6725% | 3.57s | 8.9600 | 11.6180% | 8.91s |
| | | MVMoE | 5.4732 | 10.3069% | 4.02s | 8.9232 | 11.1516% | 11.44s |
| | | MVMoE-L | 5.4830 | 10.5132% | 3.83s | 8.9511 | 11.5117% | 10.73s |
| | | MVMoE-Deeper | 5.4668 | 10.2162% | 10.21s | - | - | - |
| | | MVMoD | 5.4426 | 9.6962% | 5.95s | 8.8254 | 9.9594% | 18.62s |
| | | Ours | 5.4346 | 9.5587% | 6.75s | 8.7979 | 9.6128% | 20.86s |

*Table 17.* Performance of models on KZ9976

**KZ9976 (part 1)**

| | Problem | Solver | MTMDVRP50 Obj | Gap | Time | MTMDVRP100 Obj | Gap | Time |
|---|---|---|---|---|---|---|---|---|
| In-task | CVRP | HGS | 8.4217 | - | 1m17s | 12.4181 | - | 2m14s |
| | | POMO-MTVRP | 8.4796 | 2.1707% | 2.98s | 12.8288 | 3.3197% | 8.66s |
| | | MVMoE | 8.4334 | 1.6093% | 4.36s | 12.7846 | 2.9640% | 11.31s |
| | | MVMoE-Light | 8.441 | 1.7004% | 4.13s | 12.8041 | 3.1223% | 10.98s |
| | | MVMoE-Deeper | 8.4149 | 1.3910% | 9.56s | - | - | - |
| | | SHIELD-MoD | 8.4057 | 1.2745% | 5.79s | 12.7248 | 2.4846% | 18.03s |
| | | SHIELD | 8.3991 | 1.1865% | 6.46s | 12.7058 | 2.3312% | 20.42s |
| | OVRP | OR-tools | 5.0798 | - | 1m5s | 7.6637 | - | 2m37s |
| | | POMO-MTVRP | 5.314 | 4.6202% | 2.37s | 8.1047 | 5.8375% | 7.44s |
| | | MVMoE | 5.2892 | 4.1392% | 3.42s | 8.0450 | 5.0610% | 10.16s |
| | | MVMoE-Light | 5.2966 | 4.2863% | 3.04s | 8.0662 | 5.3313% | 9.33s |
| | | MVMoE-Deeper | 5.2511 | 3.3950% | 8.63s | - | - | - |
| | | SHIELD-MoD | 5.239 | 3.1536% | 5.13s | 7.9500 | 3.8202% | 17.04s |
| | | SHIELD | 5.2274 | 2.9232% | 5.92s | 7.8905 | 3.0341% | 19.49s |
| | VRPB | OR-tools | 6.332 | - | 1m4s | 9.3879 | - | 2m38s |
| | | POMO-MTVRP | 6.4841 | 2.4023% | 2.20s | 9.5585 | 1.2055% | 6.74s |
| | | MVMoE | 6.4416 | 1.7613% | 3.02s | 9.4946 | 1.5526% | 8.96s |
| | | MVMoE-Light | 6.459 | 2.0400% | 2.83s | 9.5275 | | 8.30s |
| | | MVMoE-Deeper | 6.4264 | 1.5232% | 7.04s | - | - | - |
| | | SHIELD-MoD | 6.4131 | 1.3130% | 4.71s | 9.4364 | 0.5879% | 15.04s |
| | | SHIELD | 6.4203 | 1.1505% | 5.25s | 9.4097 | 0.2961% | 16.92s |
| | OVRPB | OR-tools | 4.2834 | - | 1m10s | 6.2087 | - | 2m54s |
| | | POMO-MTVRP | 4.6591 | 8.7721% | 2.25s | 6.9177 | 11.4873% | 6.98s |
| | | MVMoE | 4.6161 | 7.7478% | 3.36s | 6.7990 | 9.5636% | 9.40s |
| | | MVMoE-Light | 4.6559 | 8.6910% | 2.98s | 6.8691 | 10.7103% | 8.66s |
| | | MVMoE-Deeper | 4.5958 | 7.2939% | 7.99s | - | - | - |
| | | SHIELD-MoD | 4.5812 | 6.9517% | 4.91s | 6.6961 | 7.9126% | 15.67s |
| | | SHIELD | 4.5743 | 6.7822% | 5.57s | 6.6557 | 7.2445% | 17.61s |
| | OVRPL | OR-tools | 5.0382 | - | 1m14s | 7.6885 | - | 2m54s |
| | | POMO-MTVRP | 5.2716 | 4.6326% | 2.36s | 8.1227 | 5.7422% | 7.88s |
| | | MVMoE | 5.2428 | 4.0808% | 3.48s | 8.0570 | 4.8769% | 10.76s |
| | | MVMoE-Light | 5.2548 | 4.3204% | 3.16s | 8.0883 | 5.2854% | 9.78s |
| | | MVMoE-Deeper | 5.2318 | 3.8432% | 9.03s | - | - | - |
| | | SHIELD-MoD | 5.1909 | 3.0462% | 5.23s | 7.9654 | 3.6885% | 17.51s |
| | | SHIELD | 5.1812 | 2.8560% | 6.04s | 7.9175 | 3.0511% | 19.83s |
| Out-task | VRPBL | OR-tools | 6.3024 | - | 1m13s | 9.4149 | - | 2m33s |
| | | POMO-MTVRP | 6.4771 | 2.7726% | 2.26s | 9.6073 | 2.1055% | 7.43s |
| | | MVMoE | 6.4204 | 1.8865% | 3.28s | 9.5380 | 1.3777% | 9.59s |
| | | MVMoE-Light | 6.4364 | 2.1453% | 3.02s | 9.5682 | 1.6922% | 8.91s |
| | | MVMoE-Deeper | 6.4305 | 2.0327% | 8.8s | - | - | - |
| | | SHIELD-MoD | 6.3904 | 1.4119% | 4.76s | 9.4791 | 0.7505% | 15.72s |
| | | SHIELD | 6.3788 | 1.2310% | 5.32s | 9.4541 | 0.4835% | 17.63s |
| | VRPBTW | OR-tools | 10.6457 | - | 1m20s | 18.3619 | - | 2m44s |
| | | POMO-MTVRP | 11.7415 | 10.2929% | 2.77s | 19.8107 | 8.0818% | 8.62s |
| | | MVMoE | 11.6367 | 9.4073% | 4.00s | 19.7718 | 7.8616% | 11.26s |
| | | MVMoE-Light | 11.6477 | 9.6440% | 3.61s | 19.7959 | 7.9818% | 10.65s |
| | | MVMoE-Deeper | 11.6333 | 9.2771% | 9.52s | - | - | - |
| | | SHIELD-MoD | 11.5870 | 8.9121% | 5.62s | 19.5695 | 6.7684% | 17.94s |
| | | SHIELD | 11.5423 | 8.5047% | 6.22s | 19.4954 | 6.3436% | 20.06s |
| | VRPLTW | OR-tools | 10.6950 | - | 1m23s | 18.2887 | - | 2m49s |
| | | POMO-MTVRP | 11.1707 | 4.4476% | 2.82s | 18.8087 | 3.0163% | 9.86s |
| | | MVMoE | 11.0690 | 3.5796% | 3.95s | 18.7728 | 2.8171% | 12.71s |
| | | MVMoE-Light | 11.1070 | 3.9295% | 3.68s | 18.8008 | 2.9582% | 11.99s |
| | | MVMoE-Deeper | 11.0888 | 3.6817% | 10s | - | - | - |
| | | SHIELD-MoD | 11.0282 | 3.1870% | 5.78s | 18.5787 | 1.7581% | 19.89s |
| | | SHIELD | 10.9948 | 2.8804% | 6.50s | 18.5216 | 1.4410% | 22.30s |

**KZ9976 (part 2)**

| Problem | Solver | MTMDVRP50 Obj | Gap | Time | MTMDVRP100 Obj | Gap | Time |
|---|---|---|---|---|---|---|---|
| VRPL | OR-tools | 8.4633 | - | 1m10s | 12.8865 | - | 2m39s |
| | POMO-MTVRP | 8.5304 | 0.7927% | 2.40s | 12.7791 | -0.7886% | 8.01s |
| | MVMoE | 8.4747 | 0.1676% | 3.35s | 12.7344 | -1.1332% | 10.71s |
| | MVMoE-Light | 8.4820 | 0.2478% | 3.05s | 12.7580 | -0.9518% | 9.84s |
| | MVMoE-Deeper | 8.4577 | -0.03667% | 8.44s | - | - | - |
| | SHIELD-MoD | 8.4511 | -0.1117% | 5.01s | 12.6752 | -1.5891% | 17.34s |
| | SHIELD | 8.4423 | -0.2183% | 5.71s | 12.6599 | -1.7116% | 19.68s |
| VRPTW | OR-tools | 10.6491 | - | 1m19s | 17.3625 | - | 6m32s |
| | POMO-MTVRP | 11.1016 | 6.1918% | 2.82s | 18.8165 | 8.4175% | 9.10s |
| | MVMoE | 11.0366 | 5.5490% | 3.90s | 18.7844 | 8.2241% | 11.95s |
| | MVMoE-Light | 11.0857 | 5.9973% | 3.50s | 18.8030 | 8.3233% | 11.31s |
| | MVMoE-Deeper | 10.9993 | 5.1885% | 9.9s | - | - | - |
| | SHIELD-MoD | 10.9963 | 5.1533% | 5.74s | 18.6051 | 7.1926% | 19.07s |
| | SHIELD | 10.9675 | 4.8838% | 6.46s | 18.5330 | 6.7691% | 21.51s |
| OVRPTW | OR-tools | 6.4917 | - | 1m13s | 10.6668 | - | 2m42s |
| | POMO-MTVRP | 6.8737 | 5.8847% | 2.73s | 11.3865 | 6.8257% | 8.79s |
| | MVMoE | 6.8558 | 5.6048% | 3.91s | 11.3429 | 6.4039% | 11.67s |
| | MVMoE-Light | 6.8743 | 5.8860% | 3.52s | 11.3584 | 6.5555% | 10.71s |
| | MVMoE-Deeper | 6.8098 | 4.9103% | 10.22s | - | - | - |
| | SHIELD-MoD | 6.8059 | 4.8336% | 5.83s | 11.1768 | 4.8610% | 19.00s |
| | SHIELD | 6.7798 | 4.4422% | 6.63s | 11.1398 | 4.5167% | 21.61s |
| OVRPBL | OR-tools | 4.2813 | - | 1m6s | 6.1967 | - | 2m31s |
| | POMO-MTVRP | 4.6503 | 8.6179% | 2.34s | 6.9034 | 11.4733% | 7.35s |
| | MVMoE | 4.6112 | 7.6726% | 3.36s | 6.7926 | 9.6704% | 9.76s |
| | MVMoE-Light | 4.6486 | 8.5437% | 3.07s | 6.8705 | 10.9271% | 9.04s |
| | MVMoE-Deeper | 4.5877 | 7.1562% | 8.29s | - | - | - |
| | SHIELD-MoD | 4.5717 | 6.7819% | 5.03s | 6.6910 | 8.0347% | 16.11s |
| | SHIELD | 4.5686 | 6.6880% | 5.65s | 6.6351 | 7.1246% | 17.96s |
| OVRPBTW | OR-tools | 6.4426 | - | 1m14s | 10.6121 | - | 2m42s |
| | POMO-MTVRP | 7.2019 | 11.7856% | 2.77s | 11.9287 | 12.4815% | 8.50s |
| | MVMoE | 7.1797 | 11.4104% | 3.92s | 11.8841 | 12.0447% | 11.14s |
| | MVMoE-Light | 7.1893 | 11.5716% | 3.69s | 11.8949 | 12.1569% | 10.30s |
| | MVMoE-Deeper | 7.1516 | 11.0056% | 10.04s | - | - | - |
| | SHIELD-MoD | 7.1353 | 10.7090% | 5.84s | 11.7189 | 10.4973% | 18.09s |
| | SHIELD | 7.1020 | 10.2196% | 6.59s | 11.6645 | 9.9822% | 20.31s |
| OVRPLTW | OR-tools | 6.5074 | - | 1m17s | 10.5746 | - | 2m54s |
| | POMO-MTVRP | 6.8985 | 6.0097% | 2.82s | 11.2864 | 6.7992% | 9.22s |
| | MVMoE | 6.8964 | 5.5594% | 4.11s | 11.2550 | 6.4918% | 12.12s |
| | MVMoE-Light | 6.8832 | 5.7811% | 3.70s | 11.2703 | 6.6409% | 11.17s |
| | MVMoE-Deeper | 6.8490 | 5.2494% | 10.46s | - | - | - |
| | SHIELD-MoD | 6.8213 | 4.8220% | 5.97s | 11.0853 | 4.9005% | 19.42s |
| | SHIELD | 6.7949 | 4.4390% | 6.79s | 11.0480 | 4.5534% | 21.95s |
| VRPBLTW | OR-tools | 10.5947 | - | 1m22s | 18.3014 | - | 2m47s |
| | POMO-MTVRP | 11.7074 | 10.5025% | 2.94s | 19.7894 | 8.3381% | 9.25s |
| | MVMoE | 11.5911 | 9.5324% | 4.00s | 19.7494 | 8.1026% | 11.78s |
| | MVMoE-Light | 11.6260 | 9.8357% | 3.63s | 19.7794 | 8.2802% | 11.22s |
| | MVMoE-Deeper | 11.6011 | 9.4993% | 9.79s | - | - | - |
| | SHIELD-MoD | 11.5585 | 9.2067% | 5.74s | 19.5707 | 7.1332% | 18.54s |
| | SHIELD | 11.5051 | 8.6889% | 6.32s | 19.4922 | 6.6791% | 20.70s |
| OVRPBLTW | OR-tools | 6.4313 | - | 1m19s | 10.6460 | - | 2m33s |
| | POMO-MTVRP | 7.1961 | 11.8922% | 2.90s | 11.9586 | 12.3982% | 8.81s |
| | MVMoE | 7.1622 | 11.3643% | 4.05s | 11.9137 | 11.9667% | 11.41s |
| | MVMoE-L | 7.1742 | 11.5651% | 3.76s | 11.9164 | 11.9921% | 10.68s |
| | MVMoE-Deeper | 7.1340 | 10.9255% | 10.24s | - | - | - |
| | MVMoD | 7.1239 | 10.7710% | 5.93s | 11.7414 | 10.3568% | 18.46s |
| | Ours | 7.0845 | 10.1846% | 6.67s | 11.6952 | 9.9203% | 20.71s |

Table 18. Performance of models on SW24978

**SW24978 — In-task / Out-task (left block)**

| Task | Problem | Solver | MTMDVRP50 Obj | MTMDVRP50 Gap | MTMDVRP50 Time | MTMDVRP100 Obj | MTMDVRP100 Gap | MTMDVRP100 Time |
|---|---|---|---|---|---|---|---|---|
| In-task | CVRP | HGS | 6.6979 | - | 1m18s | 9.8826 | - | 2m11s |
| | | POMO-MTVRP | 6.7538 | 2.2739% | 3.06s | 10.2519 | 3.78760% | 8.66s |
| | | MVMoE | 6.7181 | 1.7447% | 4.43s | 10.2290 | 3.56040% | 12.50s |
| | | MVMoE-Light | 6.7260 | 1.8586% | 3.89s | 10.2507 | 3.77250% | 10.58s |
| | | MVMoE-Deeper | 6.7072 | 1.5831% | 9.7s | - | - | - |
| | | SHIELD-MoD | 6.6937 | 1.3636% | 5.81s | 10.1538 | 2.79320% | 18.11s |
| | | SHIELD | 6.6842 | 1.2169% | 6.54s | 10.1386 | 2.62190% | 20.38s |
| | OVRP | OR-tools | 4.0521 | - | 1m9s | 6.1626 | - | 2m38s |
| | | POMO-MTVRP | 4.2564 | 5.0417% | 2.39s | 6.5952 | 7.13210% | 7.47s |
| | | MVMoE | 4.2382 | 4.6192% | 3.49s | 6.5459 | 6.33510% | 11.83s |
| | | MVMoE-Light | 4.2492 | 4.8916% | 3.09s | 6.5766 | 6.81880% | 9.36s |
| | | MVMoE-Deeper | 4.2075 | 3.8666% | 8.72s | - | - | - |
| | | SHIELD-MoD | 4.1888 | 3.3901% | 5.30s | 6.4406 | 4.62700% | 17.16s |
| | | SHIELD | 4.1733 | 3.0110% | 6.05s | 6.3878 | 3.75130% | 19.49s |
| | VRPB | OR-tools | 5.2139 | - | 1m2s | 7.6890 | - | 2m36s |
| | | POMO-MTVRP | 5.3608 | 2.8184% | 2.18s | 7.8945 | 2.77250% | 6.73s |
| | | MVMoE | 5.3331 | 2.3264% | 3.30s | 7.8535 | 2.24830% | 10.36s |
| | | MVMoE-Light | 5.3395 | 2.4519% | 3.06s | 7.8847 | 2.64580% | 8.30s |
| | | MVMoE-Deeper | 5.3178 | 2.0248% | 7.02s | - | - | - |
| | | SHIELD-MoD | 5.2987 | 1.6618% | 4.70s | 7.7733 | 1.21340% | 15.13s |
| | | SHIELD | 5.2861 | 1.4189% | 5.27s | 7.7549 | 0.95640% | 16.94s |
| | OVRPB | OR-tools | 3.5427 | - | 1m12s | 5.1907 | - | 2m40s |
| | | POMO-MTVRP | 3.8655 | 9.1129% | 2.30s | 5.8643 | 13.03590% | 7.00s |
| | | MVMoE | 3.8442 | 8.4761% | 3.28s | 5.7911 | 11.60920% | 10.34s |
| | | MVMoE-Light | 3.8671 | 9.1296% | 3.25s | 5.8531 | 12.81730% | 8.70s |
| | | MVMoE-Deeper | 3.8229 | 7.9091% | 8.02s | - | - | - |
| | | SHIELD-MoD | 3.7960 | 7.1164% | 4.95s | 5.6669 | 9.24450% | 15.71s |
| | | SHIELD | 3.7910 | 6.9758% | 5.69s | 5.6204 | 8.33150% | 17.62s |
| | OVRPL | OR-tools | 4.0512 | - | 1m13s | 5.1671 | - | 2m53s |
| | | POMO-MTVRP | 4.2591 | 5.1319% | 2.40s | 6.6124 | 7.36220% | 7.91s |
| | | MVMoE | 4.2415 | 4.7335% | 3.44s | 6.5529 | 6.37850% | 12.37s |
| | | MVMoE-Light | 4.2534 | 5.0199% | 3.44s | 6.5929 | 7.01370% | 9.79s |
| | | MVMoE-Deeper | 4.2251 | 4.2921% | 9s | - | - | - |
| | | SHIELD-MoD | 4.1890 | 3.4200% | 5.25s | 6.4510 | 4.74400% | 17.65s |
| | | SHIELD | 4.1754 | 3.0814% | 6.08s | 6.4068 | 4.00120% | 19.85s |
| | VRPBL | OR-tools | 5.1909 | - | 1m13s | 7.6594 | - | 2m33s |
| | | POMO-MTVRP | 5.3371 | 2.8163% | 2.29s | 7.8639 | 2.75730% | 7.42s |
| | | MVMoE | 5.3057 | 2.2667% | 3.22s | 7.8236 | 2.23730% | 10.92s |
| | | MVMoE-Light | 5.3141 | 2.4299% | 3.24s | 7.8552 | 2.64200% | 8.95s |
| | | MVMoE-Deeper | 5.3113 | 2.3187% | 8.85s | - | - | - |
| | | SHIELD-MoD | 5.2689 | 1.5530% | 4.80s | 7.7445 | 1.20130% | 15.81s |
| | | SHIELD | 5.2597 | 1.3699% | 5.35s | 7.7291 | 0.98340% | 17.58s |
| Out-task | VRPBTW | OR-tools | 8.0886 | - | 1m23s | 13.9665 | - | 2m52s |
| | | POMO-MTVRP | 8.8803 | 9.7879% | 2.85s | 14.4345 | 3.63460% | 10.13s |
| | | MVMoE | 8.8044 | 9.0737% | 3.95s | 14.3885 | 3.26980% | 14.01s |
| | | MVMoE-Light | 8.8173 | 9.1602% | 3.97s | 14.4490 | 3.68940% | 12.49s |
| | | MVMoE-Deeper | 8.8276 | 9.1366% | 10.08s | - | - | - |
| | | SHIELD-MoD | 8.7607 | 8.4928% | 5.69s | 14.2713 | 2.45880% | 20.38s |
| | | SHIELD | 8.7359 | 8.1860% | 6.29s | 14.2191 | 2.05060% | 23.08s |
| | VRPLTW | OR-tools | 8.1532 | - | 1m19s | 14.1676 | - | 2m49s |
| | | POMO-MTVRP | 8.5131 | 4.4146% | 2.77s | 15.3142 | 8.49130% | 8.73s |
| | | MVMoE | 8.4670 | 4.0004% | 3.92s | 15.2694 | 8.15410% | 12.06s |
| | | MVMoE-Light | 8.4753 | 4.0737% | 3.83s | 15.3335 | 8.61220% | 10.93s |
| | | MVMoE-Deeper | 8.4764 | 3.9635% | 9.49s | - | - | - |
| | | SHIELD-MoD | 8.4117 | 3.2956% | 5.69s | 15.1656 | 7.47000% | 18.25s |
| | | SHIELD | 8.3926 | 3.0792% | 6.29s | 15.0792 | 6.77220% | 20.56s |

**SW24978 — In-task / Out-task (right block)**

| Task | Problem | Solver | MTMDVRP50 Obj | MTMDVRP50 Gap | MTMDVRP50 Time | MTMDVRP100 Obj | MTMDVRP100 Gap | MTMDVRP100 Time |
|---|---|---|---|---|---|---|---|---|
| In-task | VRPL | OR-tools | 6.7721 | - | 1m10s | 10.3234 | - | 2m38s |
| | | POMO-MTVRP | 6.8296 | 0.8497% | 2.43s | 10.2774 | -0.4057% | 8.00s |
| | | MVMoE | 6.7881 | 0.2730% | 3.32s | 10.2491 | -0.6711% | 11.64s |
| | | MVMoE-Light | 6.7941 | 0.3548% | 3.60s | 10.2727 | -0.4486% | 9.87s |
| | | MVMoE-Deeper | 6.7762 | 0.0947% | 8.45s | - | - | - |
| | | SHIELD-MoD | 6.7623 | -0.1163% | 5.04s | 10.1749 | -1.4005% | 17.44s |
| | | SHIELD | 6.7533 | -0.2187% | 5.78s | 10.1589 | -1.5678% | 19.74s |
| | VRPTW | OR-tools | 8.3232 | - | 1m17s | 13.3531 | - | 6m29s |
| | | POMO-MTVRP | 8.6793 | 6.1415% | 2.88s | 14.4825 | 8.5406% | 9.26s |
| | | MVMoE | 8.6465 | 5.7162% | 3.86s | 14.4327 | 8.1655% | 13.26s |
| | | MVMoE-Light | 8.6542 | 5.8053% | 3.97s | 14.4849 | 8.5695% | 11.65s |
| | | MVMoE-Deeper | 8.6081 | 5.2484% | 9.91s | - | - | - |
| | | SHIELD-MoD | 8.5945 | 5.0838% | 5.81s | 14.3061 | 7.2326% | 19.33s |
| | | SHIELD | 8.5729 | 4.8030% | 6.55s | 14.2664 | 6.9055% | 22.03s |
| | OVRPTW | OR-tools | 5.2057 | - | 1m11s | 8.4320 | - | 2m42s |
| | | POMO-MTVRP | 5.5109 | 5.8629% | 2.77s | 9.0652 | 7.65730% | 8.81s |
| | | MVMoE | 5.5111 | 5.8568% | 3.84s | 9.0238 | 7.1596% | 12.81s |
| | | MVMoE-Light | 5.5221 | 6.0649% | 3.87s | 9.0742 | 7.7730% | 10.96s |
| | | MVMoE-Deeper | 5.4697 | 5.1083% | 10.39s | - | - | - |
| | | SHIELD-MoD | 5.4527 | 4.7333% | 5.85s | 8.9004 | 5.7226% | 18.90s |
| | | SHIELD | 5.4445 | 4.6356% | 6.67s | 8.8643 | 5.2987% | 21.56s |
| | OVRPBL | OR-tools | 3.5320 | - | 1m8s | 5.2096 | - | 2m36s |
| | | POMO-MTVRP | 3.8526 | 9.0782% | 2.36s | 5.8816 | 12.9788% | 7.40s |
| | | MVMoE | 3.8357 | 8.5557% | 3.41s | 5.8117 | 11.6128% | 10.78s |
| | | MVMoE-Light | 3.8564 | 9.1395% | 3.21s | 5.8700 | 12.7548% | 9.04s |
| | | MVMoE-Deeper | 3.8095 | 7.8556% | 8.34s | - | - | - |
| | | SHIELD-MoD | 3.7870 | 7.2199% | 5.08s | 5.6836 | 9.1828% | 16.07s |
| | | SHIELD | 3.7777 | 6.9288% | 5.68s | 5.6304 | 8.1570% | 18.05s |
| Out-task | OVRPBTW | OR-tools | 5.1779 | - | 1m8s | 8.4308 | - | 2m41s |
| | | POMO-MTVRP | 5.7623 | 11.2859% | 2.80s | 9.4775 | 12.5687% | 8.48s |
| | | MVMoE | 5.7434 | 10.9114% | 3.88s | 9.4291 | 11.9843% | 11.52s |
| | | MVMoE-Light | 5.7527 | 11.0847% | 3.83s | 9.4815 | 12.6198% | 10.50s |
| | | MVMoE-Deeper | 5.7286 | 10.6357% | 10.09s | - | - | - |
| | | SHIELD-MoD | 5.6986 | 10.0178% | 5.84s | 9.3156 | 10.6768% | 17.99s |
| | | SHIELD | 5.6828 | 9.7971% | 5.68s | 9.2617 | 10.0060% | 20.27s |
| | OVRPLTW | OR-tools | 5.1469 | - | 1m15s | 8.4292 | - | 2m49s |
| | | POMO-MTVRP | 5.4627 | 6.1352% | 2.81s | 9.0423 | 7.4062% | 9.20s |
| | | MVMoE | 5.4512 | 5.9274% | 4.01s | 9.0008 | 6.9011% | 12.92s |
| | | MVMoE-Light | 5.4605 | 6.0997% | 3.80s | 9.0501 | 7.4929% | 11.48s |
| | | MVMoE-Deeper | 5.4396 | 5.6877% | 10.51s | - | - | - |
| | | SHIELD-MoD | 5.4001 | 4.9193% | 6.01s | 8.8812 | 5.5065% | 19.38s |
| | | SHIELD | 5.3864 | 4.7154% | 6.84s | 8.8487 | 5.1093% | 22.00s |
| | VRPBLTW | OR-tools | 8.1677 | - | 1m22s | 13.6276 | - | 2m49s |
| | | POMO-MTVRP | 8.9615 | 9.7182% | 2.96s | 14.7188 | 8.3795% | 9.28s |
| | | MVMoE | 8.8913 | 9.0623% | 3.96s | 14.6809 | 8.0858% | 12.80s |
| | | MVMoE-Light | 8.8890 | 9.0142% | 3.79s | 14.7363 | 8.5020% | 11.41s |
| | | MVMoE-Deeper | 8.9035 | 9.0086% | 9.87s | - | - | - |
| | | SHIELD-MoD | 8.8410 | 8.4141% | 5.83s | 14.5760 | 7.3226% | 18.69s |
| | | SHIELD | 8.8013 | 7.9695% | 6.43s | 14.5032 | 6.7412% | 20.92s |
| | OVRPBLTW | OR-tools | 5.1245 | - | 1m18s | 8.4572 | - | 2m31s |
| | | POMO-MTVRP | 5.7114 | 11.4524% | 2.88s | 9.5106 | 12.5946% | 8.87s |
| | | MVMoE | 5.6997 | 11.2344% | 4.03s | 9.4532 | 11.8813% | 12.13s |
| | | MVMoE-L | 5.7007 | 11.2689% | 3.86s | 9.5087 | 12.5619% | 10.84s |
| | | MVMoE-Deeper | 5.6842 | 10.9230% | 10.3s | - | - | - |
| | | MVMoD | 5.6483 | 10.2214% | 5.95s | 9.3427 | 10.6239% | 18.44s |
| | | Ours | 5.6312 | 9.9603% | 6.75s | 9.3013 | 10.0938% | 20.74s |

*Table 19.* Performance of models on VM22775

**VM22775 — In-task / Out-task (left column group)**

| Problem | Solver | MTMDVRP50 Obj | MTMDVRP50 Gap | MTMDVRP50 Time | MTMDVRP100 Obj | MTMDVRP100 Gap | MTMDVRP100 Time |
|---|---|---|---|---|---|---|---|
| **CVRP** | HGS | 8.2120 | - | 1m35s | 12.1714 | - | 2m15s |
| | POMO-MTVRP | 8.2974 | 2.2454% | 4.30s | 12.5856 | 3.4193% | 8.70s |
| | MVMoE | 8.2459 | 1.6115% | 4.29s | 12.5450 | 3.0942% | 11.37s |
| | MVMoE-Light | 8.2554 | 1.7229% | 4.02s | 12.5657 | 3.2645% | 10.62s |
| | MVMoE-Deeper | 8.2352 | 1.4836% | 9.66s | | | 18.05s |
| | SHIELD-MoD | 8.2229 | 1.3205% | 5.78s | 12.4767 | 2.5224% | 18.05s |
| | SHIELD | 8.2143 | 1.2193% | 6.48s | 12.4608 | 2.3879% | 20.38s |
| **OVRP** | OR-tools | 4.8138 | - | 1m7s | 7.3689 | - | 2m39s |
| | POMO-MTVRP | 5.0672 | 5.2636% | 2.38s | 7.8238 | 6.2490% | 7.49s |
| | MVMoE | 5.0433 | 4.7859% | 3.38s | 7.7843 | 5.7257% | 10.39s |
| | MVMoE-Light | 5.0557 | 5.0703% | 3.04s | 7.7975 | 5.8929% | 9.38s |
| | MVMoE-Deeper | 4.9992 | 3.8900% | 8.65s | | | 17.11s |
| | SHIELD-MoD | 4.9870 | 3.6258% | 5.15s | 7.6502 | 3.8971% | 17.11s |
| | SHIELD | 4.9697 | 3.2797% | 5.98s | 7.6047 | 3.2896% | 19.63s |
| **VRPB** | OR-tools | 6.0429 | - | 1m1s | 9.0476 | - | 2m35s |
| | POMO-MTVRP | 6.2125 | 2.8072% | 2.23s | 9.2501 | 2.3584% | 6.74s |
| | MVMoE | 6.1694 | 2.1187% | 3.11s | 9.2009 | 1.8153% | 9.18s |
| | MVMoE-Light | 6.1849 | 2.3749% | 2.91s | 9.2190 | 2.0173% | 8.27s |
| | MVMoE-Deeper | 6.1576 | 1.9315% | 6.99s | | | 15.08s |
| | SHIELD-MoD | 6.1402 | 1.6367% | 4.69s | 9.1118 | 0.8220% | 15.08s |
| | SHIELD | 6.1326 | 1.5173% | 5.23s | 9.0876 | 0.5547% | 16.97s |
| **OVRPB** | OR-tools | 3.8870 | - | 1m8s | 5.7542 | - | 2m39s |
| | POMO-MTVRP | 4.2505 | 9.3515% | 2.42s | 6.4354 | 11.9283% | 7.05s |
| | MVMoE | 4.2141 | 8.4012% | 3.35s | 6.3647 | 10.6831% | 9.52s |
| | MVMoE-Light | 4.2512 | 9.3733% | 3.03s | 6.4183 | 11.6274% | 8.69s |
| | MVMoE-Deeper | 4.1841 | 7.6443% | 7.94s | | | 15.65s |
| | SHIELD-MoD | 4.1656 | 7.1677% | 4.95s | 6.2023 | 7.8597% | 15.65s |
| | SHIELD | 4.1613 | 7.0535% | 5.59s | 6.1568 | 7.0782% | 17.63s |
| **OVRPL** | OR-tools | 4.8097 | - | 1m19s | 7.3550 | - | 2m55s |
| | POMO-MTVRP | 5.0597 | 5.1971% | 2.54s | 7.8041 | 6.1984% | 7.94s |
| | MVMoE | 5.0372 | 4.7388% | 3.48s | 7.7679 | 5.7154% | 10.82s |
| | MVMoE-Light | 5.0494 | 5.0200% | 3.05s | 7.7777 | 5.8223% | 9.78s |
| | MVMoE-Deeper | 5.0157 | 4.2837% | 8.98s | | | 17.53s |
| | SHIELD-MoD | 4.9793 | 3.5419% | 5.26s | 7.6389 | 3.9467% | 17.53s |
| | SHIELD | 4.9624 | 3.1907% | 6.08s | 7.5889 | 3.2589% | 19.97s |
| **VRPBL** | OR-tools | 6.0258 | - | 1m16s | 8.9724 | - | 2m45s |
| | POMO-MTVRP | 6.1987 | 2.8686% | 2.45s | 9.1670 | 2.2641% | 7.42s |
| | MVMoE | 6.1500 | 2.1044% | 3.29s | 9.1213 | 1.7682% | 9.63s |
| | MVMoE-Light | 6.1670 | 2.3844% | 3.05s | 9.1414 | 1.9859% | 8.91s |
| | MVMoE-Deeper | 6.1722 | 2.4288% | 8.87s | | | 15.73s |
| | SHIELD-MoD | 6.1227 | 1.6549% | 4.78s | 9.0330 | 0.7832% | 15.73s |
| | SHIELD | 6.1111 | 1.4675% | 5.34s | 9.0043 | 0.4597% | 17.66s |
| **VRPLTW** | OR-tools | 10.7055 | - | 1m23s | 18.7523 | - | 2m45s |
| | POMO-MTVRP | 11.7038 | 9.3248% | 2.94s | 20.0852 | 7.3516% | 8.63s |
| | MVMoE | 11.6157 | 8.7550% | 3.96s | 20.0964 | 7.4046% | 11.29s |
| | MVMoE-Light | 11.6391 | 8.9638% | 3.62s | 20.0970 | 7.4093% | 10.65s |
| | MVMoE-Deeper | 11.6580 | 8.8974% | 9.48s | | | 17.96s |
| | SHIELD-MoD | 11.5819 | 8.3991% | 5.66s | 19.8598 | 6.1625% | 17.96s |
| | SHIELD | 11.5264 | 7.8871% | 6.25s | 19.7931 | 5.7595% | 20.13s |

**VM22775 — In-task / Out-task (right column group)**

| Problem | Solver | MTMDVRP50 Obj | MTMDVRP50 Gap | MTMDVRP50 Time | MTMDVRP100 Obj | MTMDVRP100 Gap | MTMDVRP100 Time |
|---|---|---|---|---|---|---|---|
| **VRPL** | OR-tools | 8.2151 | - | 1m11s | 12.5283 | - | 2m39s |
| | POMO-MTVRP | 8.3078 | 1.1279% | 2.55s | 12.4811 | -0.3508% | 8.01s |
| | MVMoE | 8.2539 | 0.5085% | 3.33s | 12.4472 | -0.6200% | 10.84s |
| | MVMoE-Light | 8.2593 | 0.5735% | 3.12s | 12.4618 | -0.5008% | 9.88s |
| | MVMoE-Deeper | 8.2412 | 0.3507% | 8.4s | | | 17.29s |
| | SHIELD-MoD | 8.2272 | 0.1836% | 5.03s | 12.3739 | -1.2047% | 17.29s |
| | SHIELD | 8.2167 | 0.0535% | 5.73s | 12.3579 | -1.3374% | 19.71s |
| **VRPTW** | OR-tools | 10.5525 | - | 1m16s | 17.7378 | - | 6m34s |
| | POMO-MTVRP | 10.9940 | 6.2437% | 3.03s | 19.2257 | 8.4620% | 9.17s |
| | MVMoE | 10.9227 | 5.5759% | 3.81s | 19.2077 | 8.3633% | 12.03s |
| | MVMoE-Light | 10.9546 | 5.8847% | 3.55s | 19.2231 | 8.4491% | 11.34s |
| | MVMoE-Deeper | 10.8784 | 5.1612% | 9.87s | | | 19.14s |
| | SHIELD-MoD | 10.8878 | 5.2251% | 5.77s | 18.9746 | 7.0508% | 19.14s |
| | SHIELD | 10.8471 | 4.8543% | 6.50s | 18.9211 | 6.7445% | 21.64s |
| **OVRPTW** | OR-tools | 6.0966 | - | 1m19s | 10.1562 | - | 2m44s |
| | POMO-MTVRP | 6.5159 | 6.8769% | 2.93s | 10.8685 | 7.1113% | 8.74s |
| | MVMoE | 6.4816 | 6.3126% | 3.84s | 10.8348 | 6.7774% | 11.72s |
| | MVMoE-Light | 6.5058 | 6.7237% | 3.55s | 10.8369 | 6.7987% | 10.73s |
| | MVMoE-Deeper | 6.3464 | 5.6127% | 10.32s | | | 18.91s |
| | SHIELD-MoD | 6.4294 | 5.4647% | 5.78s | 10.6427 | 4.8878% | 18.91s |
| | SHIELD | 6.3982 | 4.9368% | 6.60s | 10.5970 | 4.4416% | 21.43s |
| **OVRPBL** | OR-tools | 3.8906 | - | 1m5s | 5.7679 | - | 2m35s |
| | POMO-MTVRP | 4.2454 | 9.1193% | 2.51s | 6.4539 | 11.9664% | 7.39s |
| | MVMoE | 4.2179 | 8.4004% | 3.41s | 6.3792 | 10.6523% | 9.91s |
| | MVMoE-Light | 4.2518 | 9.2892% | 3.08s | 6.4421 | 11.7448% | 9.06s |
| | MVMoE-Deeper | 4.1888 | 7.6644% | 8.33s | | | 16.06s |
| | SHIELD-MoD | 4.1716 | 7.2214% | 5.02s | 6.2237 | 7.9653% | 16.06s |
| | SHIELD | 4.1646 | 7.0469% | 5.67s | 6.1758 | 7.1334% | 18.05s |
| **OVRPBTW** | OR-tools | 6.0530 | - | 1m15s | 10.1174 | - | 2m40s |
| | POMO-MTVRP | 6.8045 | 12.4156% | 2.89s | 11.3495 | 12.3227% | 8.49s |
| | MVMoE | 6.7815 | 12.0607% | 3.90s | 11.3082 | 11.9073% | 11.10s |
| | MVMoE-Light | 6.7831 | 12.0955% | 3.69s | 11.3072 | 11.8808% | 10.30s |
| | MVMoE-Deeper | 6.7538 | 11.5779% | 10.01s | | | 18.06s |
| | SHIELD-MoD | 6.7272 | 11.1773% | 5.77s | 11.1283 | 10.1297% | 18.06s |
| | SHIELD | 6.6794 | 10.3960% | 6.58s | 11.0622 | 9.4637% | 20.30s |
| **OVRPLTW** | OR-tools | 6.0521 | - | 1m18s | 10.1576 | - | 2m50s |
| | POMO-MTVRP | 6.4593 | 6.7289% | 2.92s | 10.8730 | 7.1771% | 9.17s |
| | MVMoE | 6.4319 | 6.2823% | 4.02s | 10.8447 | 6.9019% | 12.17s |
| | MVMoE-Light | 6.4508 | 6.5993% | 3.72s | 10.8427 | 6.8716% | 11.20s |
| | MVMoE-Deeper | 6.3882 | 5.5533% | 10.45s | | | 19.33s |
| | SHIELD-MoD | 6.3805 | 5.4425% | 5.95s | 10.6515 | 4.9986% | 19.33s |
| | SHIELD | 6.3456 | 4.8817% | 6.79s | 10.6088 | 4.5639% | 21.87s |
| **VRPBLTW** | OR-tools | 10.6434 | - | 1m18s | 18.5622 | - | 2m45s |
| | POMO-MTVRP | 11.6674 | 9.6210% | 3.08s | 19.8427 | 7.1626% | 9.27s |
| | MVMoE | 11.5700 | 8.9490% | 3.98s | 19.8255 | 7.0499% | 11.88s |
| | MVMoE-Light | 11.6111 | 9.2697% | 3.64s | 19.8335 | 7.0997% | 11.22s |
| | MVMoE-Deeper | 11.6039 | 9.0243% | 9.76s | | | 18.70s |
| | SHIELD-MoD | 11.5523 | 8.6835% | 5.80s | 19.5935 | 5.8187% | 18.70s |
| | SHIELD | 11.4789 | 8.0243% | 6.32s | 19.5566 | 5.5655% | 20.76s |
| **OVRPBLTW** | OR-tools | 6.0628 | - | 1m20s | 10.0760 | - | 2m33s |
| | POMO-MTVRP | 6.8095 | 12.3158% | 2.95s | 11.3098 | 12.3540% | 8.86s |
| | MVMoE | 6.7884 | 11.9893% | 4.06s | 11.2734 | 11.9732% | 11.48s |
| | MVMoE-L | 6.7993 | 12.1789% | 3.79s | 11.2539 | 11.7919% | 10.65s |
| | MVMoE-Deeper | 6.7635 | 11.5571% | 10.19s | | | 18.44s |
| | MVMoD | 6.7314 | 11.0334% | 5.90s | 11.0888 | 10.1594% | 18.44s |
| | Ours | 6.6856 | 10.3143% | 6.66s | 11.0178 | 9.4527% | 20.64s |

Table 20. Performance of models on EG7146

**EG7146 (left half)**

| Category | Problem | Solver | MTMDVRP50 Obj | MTMDVRP50 Gap | MTMDVRP50 Time | MTMDVRP100 Obj | MTMDVRP100 Gap | MTMDVRP100 Time |
|---|---|---|---|---|---|---|---|---|
| In-task | CVRP | HGS | 4.2661 | - | 1m21s | 6.3233 | - | 2m15s |
| | | POMO-MTVRP | 4.3335 | 2.6537% | 3.33s | 6.6029 | 4.7559% | 9.03s |
| | | MVMoE | 4.3018 | 2.0324% | 4.32s | 6.6075 | 4.8078% | 12.39s |
| | | MVMoE-Light | 4.3061 | 2.1268% | 4.17s | 6.6246 | 5.0781% | 11.98s |
| | | MVMoE-Deeper | 4.3061 | 2.1625% | 9.68s | - | - | - |
| | | SHIELD-MoD | 4.2876 | 1.6642% | 5.83s | 6.5535 | 3.9363% | 18.16s |
| | | SHIELD | 4.2802 | 1.4656% | 6.49s | 6.5367 | 3.6566% | 22.93s |
| | OVRP | OR-tools | 2.4397 | - | 1m20s | 3.7510 | - | 2m42s |
| | | POMO-MTVRP | 2.6045 | 6.7560% | 2.91s | 4.1674 | 11.8018% | 8.09s |
| | | MVMoE | 2.5861 | 6.2931% | 3.66s | 4.1673 | 11.8226% | 11.08s |
| | | MVMoE-Light | 2.5995 | 6.8360% | 3.26s | 4.1427 | 11.1366% | 10.74s |
| | | MVMoE-Deeper | 2.5787 | 6.0468% | 8.89s | - | - | - |
| | | SHIELD-MoD | 2.5514 | 4.7885% | 5.15s | 4.0474 | 8.4444% | 17.25s |
| | | SHIELD | 2.5357 | 4.1187% | 6.16s | 3.9849 | 6.7276% | 19.88s |
| | VRPB | OR-tools | 3.3731 | - | 1m1s | 4.9564 | - | 2m40s |
| | | POMO-MTVRP | 3.4892 | 3.4424% | 2.62s | 5.1741 | 4.6788% | 7.01s |
| | | MVMoE | 3.4641 | 2.8546% | 3.03s | 5.1634 | 4.4185% | 9.74s |
| | | MVMoE-Light | 3.4676 | 2.9329% | 3.02s | 5.1815 | 4.8007% | 9.44s |
| | | MVMoE-Deeper | 3.4652 | 2.8993% | 7.11s | - | - | - |
| | | SHIELD-MoD | 3.4455 | 2.2692% | 4.70s | 5.1077 | 3.2931% | 15.18s |
| | | SHIELD | 3.4347 | 1.9133% | 5.30s | 5.0930 | 3.0192% | 17.34s |
| | OVRPB | OR-tools | 2.0569 | - | 1m20s | 3.0546 | - | 2m41s |
| | | POMO-MTVRP | 2.2657 | 10.1491% | 2.76s | 3.5751 | 17.7022% | 7.32s |
| | | MVMoE | 2.2547 | 9.7586% | 3.32s | 3.5857 | 18.0985% | 10.33s |
| | | MVMoE-Light | 2.2747 | 10.7739% | 3.09s | 3.5722 | 17.6061% | 9.84s |
| | | MVMoE-Deeper | 2.2469 | 9.2368% | 7.97s | - | - | - |
| | | SHIELD-MoD | 2.2204 | 8.1100% | 4.90s | 3.4462 | 13.3146% | 15.74s |
| | | SHIELD | 2.2093 | 7.5305% | 5.59s | 3.3731 | 10.9531% | 17.60s |
| | OVRPL | OR-tools | 2.4504 | - | 1m16s | 3.7508 | - | 2m49s |
| | | POMO-MTVRP | 2.6115 | 6.5748% | 2.69s | 4.1693 | 11.8376% | 8.44s |
| | | MVMoE | 2.5969 | 6.2734% | 3.54s | 4.1683 | 11.8336% | 11.71s |
| | | MVMoE-Light | 2.6038 | 6.5720% | 3.33s | 4.1360 | 10.9600% | 11.30s |
| | | MVMoE-Deeper | 2.6103 | 6.5254% | 9.09s | - | - | - |
| | | SHIELD-MoD | 2.5630 | 4.7895% | 5.24s | 4.0473 | 8.4154% | 17.64s |
| | | SHIELD | 2.5450 | 4.0585% | 6.12s | 3.9838 | 6.7253% | 20.22s |
| Out-task | VRPBL | OR-tools | 3.2954 | - | 1m19s | 4.9569 | - | 2m33s |
| | | POMO-MTVRP | 3.4085 | 3.4311% | 3.03s | 5.1859 | 4.9025% | 7.72s |
| | | MVMoE | 3.3857 | 2.8847% | 4.00s | 5.1710 | 4.5591% | 10.50s |
| | | MVMoE-Light | 3.3891 | 2.9609% | 3.67s | 5.1941 | 5.0288% | 10.31s |
| | | MVMoE-Deeper | 3.4121 | 3.5403% | 9.51s | - | - | - |
| | | SHIELD-MoD | 3.3661 | 2.2564% | 5.58s | 5.1138 | 3.9337% | 15.87s |
| | | SHIELD | 3.3562 | 1.9297% | 6.15s | 5.0982 | 3.0999% | 18.12s |
| | VRPBTW | OR-tools | 4.7375 | - | 1m23s | 7.9075 | - | 2m43s |
| | | POMO-MTVRP | 5.1863 | 9.4734% | 3.03s | 8.6547 | 10.1123% | 8.67s |
| | | MVMoE | 5.1460 | 8.9711% | 4.00s | 8.6541 | 10.0303% | 12.47s |
| | | MVMoE-Light | 5.1448 | 8.9284% | 3.67s | 8.6629 | 10.1730% | 13.64s |
| | | MVMoE-Deeper | 5.1886 | 9.5212% | 9.51s | - | - | - |
| | | SHIELD-MoD | 5.1189 | 8.3411% | 5.58s | 8.5744 | 9.0841% | 18.31s |
| | | SHIELD | 5.1109 | 8.2421% | 6.15s | 8.5188 | 8.3280% | 21.80s |
| | VRPLTW | OR-tools | 4.8841 | - | 1m31s | 8.0086 | - | 2m55s |
| | | POMO-MTVRP | 5.1422 | 5.2845% | 3.11s | 8.4323 | 5.8016% | 10.10s |
| | | MVMoE | 5.0992 | 4.6517% | 4.12s | 8.4420 | 5.8510% | 14.30s |
| | | MVMoE-Light | 5.0994 | 4.6053% | 3.85s | 8.4599 | 6.1180% | 16.23s |
| | | MVMoE-Deeper | 5.1307 | 5.0499% | 10.22s | - | - | - |
| | | SHIELD-MoD | 5.0942 | 4.3019% | 5.80s | 8.3590 | 4.8426% | 20.45s |
| | | SHIELD | 5.0728 | 4.1259% | 6.94s | 8.3220 | 4.4071% | 24.94s |

**EG7146 (right half)**

| Problem | Solver | MTMDVRP50 Obj | MTMDVRP50 Gap | MTMDVRP50 Time | MTMDVRP100 Obj | MTMDVRP100 Gap | MTMDVRP100 Time |
|---|---|---|---|---|---|---|---|
| VRPL | OR-tools | 4.2562 | - | 1m2s | 6.5015 | - | 2m41s |
| | POMO-MTVRP | 4.3245 | 1.6041% | 2.92s | 6.5822 | 1.3993% | 8.39s |
| | MVMoE | 4.2965 | 1.0675% | 3.39s | 6.5868 | 1.4509% | 11.30s |
| | MVMoE-Light | 4.2979 | 1.0892% | 3.26s | 6.6053 | 1.7299% | 11.48s |
| | MVMoE-Deeper | 4.2990 | 1.1466% | 8.49s | - | - | - |
| | SHIELD-MoD | 4.2801 | 0.6453% | 5.07s | 6.5317 | 0.6028% | 17.47s |
| | SHIELD | 4.2717 | 0.4317% | 5.79s | 6.5171 | 0.3611% | 21.76s |
| VRPTW | OR-tools | 4.8840 | - | 1m28s | 7.5872 | - | 6m35s |
| | POMO-MTVRP | 5.1345 | 6.7583% | 3.36s | 8.3451 | 10.3102% | 9.29s |
| | MVMoE | 5.1021 | 6.1431% | 3.94s | 8.3413 | 10.1853% | 13.57s |
| | MVMoE-Light | 5.1049 | 6.1902% | 3.66s | 8.3665 | 10.5592% | 15.44s |
| | MVMoE-Deeper | 5.0940 | 6.0413% | 10.01s | - | - | - |
| | SHIELD-MoD | 5.1510 | 5.4674% | 5.74s | 8.2620 | 9.1747% | 19.75s |
| | SHIELD | 5.0787 | 5.6905% | 6.69s | 8.2334 | 8.7933% | 23.82s |
| OVRPTW | OR-tools | 3.0238 | - | 1m23s | 4.9353 | - | 2m50s |
| | POMO-MTVRP | 3.2700 | 8.1407% | 3.22s | 5.4417 | 10.9588% | 9.20s |
| | MVMoE | 3.2627 | 8.1766% | 3.99s | 5.4753 | 11.5536% | 14.07s |
| | MVMoE-Light | 3.2622 | 8.0924% | 3.56s | 5.4714 | 11.5304% | 13.93s |
| | MVMoE-Deeper | 3.2479 | 7.7094% | 10.52s | - | - | - |
| | SHIELD-MoD | 3.2360 | 7.0192% | 5.78s | 5.3584 | 9.1885% | 20.05s |
| | SHIELD | 3.2229 | 6.8535% | 6.77s | 5.3254 | 8.5817% | 23.67s |
| OVRPBL | OR-tools | 2.0523 | - | 1m9s | 3.0685 | - | 2m33s |
| | POMO-MTVRP | 2.2616 | 10.1999% | 2.63s | 3.5984 | 17.9380% | 7.70s |
| | MVMoE | 2.2526 | 9.8491% | 3.45s | 3.6028 | 18.0853% | 10.66s |
| | MVMoE-Light | 2.2652 | 10.5639% | 3.13s | 3.5860 | 17.5421% | 10.34s |
| | MVMoE-Deeper | 2.2397 | 9.1323% | 8.34s | - | - | - |
| | SHIELD-MoD | 2.2165 | 8.1623% | 5.01s | 3.4627 | 13.3097% | 16.21s |
| | SHIELD | 2.2037 | 7.5294% | 5.70s | 3.3874 | 10.8413% | 18.12s |
| OVRPBTW | OR-tools | 2.9200 | - | 1m16s | 4.8008 | - | 2m49s |
| | POMO-MTVRP | 3.2772 | 12.2321% | 3.09s | 5.5019 | 15.1401% | 8.85s |
| | MVMoE | 3.2305 | 12.1102% | 3.93s | 5.5239 | 15.5221% | 13.05s |
| | MVMoE-Light | 3.2664 | 12.0087% | 3.71s | 5.5144 | 15.3594% | 12.82s |
| | MVMoE-Deeper | 3.2752 | 12.1652% | 10.15s | - | - | - |
| | SHIELD-MoD | 3.2366 | 10.8410% | 5.80s | 5.4285 | 13.5184% | 18.52s |
| | SHIELD | 3.2274 | 10.7363% | 6.67s | 5.3650 | 12.2104% | 22.24s |
| OVRPLTW | OR-tools | 2.9926 | - | 1m21s | 4.8134 | - | 2m58s |
| | POMO-MTVRP | 3.2366 | 8.1519% | 3.09s | 5.3253 | 11.2289% | 9.66s |
| | MVMoE | 3.2305 | 8.2202% | 4.14s | 5.3475 | 11.6426% | 14.57s |
| | MVMoE-Light | 3.2343 | 8.2613% | 3.72s | 5.3524 | 11.7726% | 14.39s |
| | MVMoE-Deeper | 3.2433 | 8.3770% | 10.71s | - | - | - |
| | SHIELD-MoD | 3.2093 | 7.2411% | 5.95s | 5.2409 | 9.4280% | 20.35s |
| | SHIELD | 3.1930 | 6.9771% | 6.93s | 5.2088 | 8.7900% | 24.35s |
| VRPBLTW | OR-tools | 4.7699 | - | 1m25s | 7.9676 | - | 2m44s |
| | POMO-MTVRP | 5.2290 | 9.6259% | 3.23s | 8.6980 | 9.8474% | 9.28s |
| | MVMoE | 5.1827 | 8.9705% | 3.95s | 8.7020 | 9.8280% | 13.11s |
| | MVMoE-Light | 5.1899 | 9.1190% | 3.70s | 8.7208 | 10.0636% | 14.68s |
| | MVMoE-Deeper | 5.2269 | 9.5803% | 9.83s | - | - | - |
| | SHIELD-MoD | 5.1630 | 8.4985% | 5.65s | 8.6340 | 8.9862% | 18.95s |
| | SHIELD | 5.1542 | 8.3490% | 6.38s | 8.5772 | 8.2720% | 22.41s |
| OVRPBLTW | OR-tools | 2.9427 | - | 1m25s | 4.8417 | - | 2m39s |
| | POMO-MTVRP | 3.3049 | 12.3088% | 3.12s | 5.5503 | 15.2090% | 9.19s |
| | MVMoE | 3.3005 | 12.3844% | 4.02s | 5.5658 | 15.4365% | 13.38s |
| | MVMoE-L | 3.3004 | 12.3136% | 3.78s | 5.5611 | 15.4145% | 13.35s |
| | MVMoE-Deeper | 3.3061 | 12.3491% | 10.39s | - | - | - |
| | MVMoD | 3.2702 | 11.1276% | 5.86s | 5.4697 | 13.4759% | 19.02s |
| | Ours | 3.2579 | 10.9948% | 6.79s | 5.4196 | 12.4838% | 22.03s |

*Table 21.* Performance of models on FI10639

**FI10639 (Left block)**

| | Problem | Solver | MTMDVRP50 Obj | Gap | Time | MTMDVRP100 Obj | Gap | Time |
|---|---|---|---|---|---|---|---|---|
| In-task | CVRP | HGS | 7.1789 | - | 1m21s | 10.6055 | - | 2m11s |
| | | POMO-MTVRP | 7.2316 | 2.2536% | 3.18s | 10.9689 | 3.4421% | 8.69s |
| | | MVMoE | 7.1891 | 1.6553% | 4.09s | 10.9438 | 3.2108% | 11.58s |
| | | MVMoE-Light | 7.1959 | 1.7537% | 4.07s | 10.9590 | 3.3535% | 10.52s |
| | | MVMoE-Deeper | 7.1775 | 1.4944% | 9.66s | - | - | - |
| | | SHIELD-MoD | 7.1675 | 1.3514% | 5.97s | 10.8778 | 2.5840% | 18.05s |
| | | SHIELD | 7.1578 | 1.2110% | 6.45s | 10.8700 | 2.5115% | 20.48s |
| | OVRP | OR-tools | 4.3654 | - | 1m7s | 6.6709 | - | 2m37s |
| | | POMO-MTVRP | 4.5669 | 4.6148% | 2.32s | 7.0708 | 6.0585% | 7.44s |
| | | MVMoE | 4.5476 | 4.7107% | 3.30s | 7.0215 | 5.3245% | 10.18s |
| | | MVMoE-Light | 4.5643 | 4.5705% | 3.29s | 7.0384 | 5.5842% | 9.33s |
| | | MVMoE-Deeper | 4.5261 | 3.6903% | 8.81s | - | - | - |
| | | SHIELD-MoD | 4.5059 | 3.2248% | 5.29s | 6.9334 | 4.0124% | 17.02s |
| | | SHIELD | 4.4901 | 2.8598% | 5.96s | 6.8809 | 3.2150% | 19.42s |
| | VRPB | OR-tools | 5.5089 | - | 1m3s | 8.2519 | - | 2m35s |
| | | POMO-MTVRP | 5.6511 | 2.5818% | 2.19s | 8.4295 | 2.2114% | 6.73s |
| | | MVMoE | 5.6148 | 1.9523% | 3.02s | 8.3929 | 1.7773% | 9.21s |
| | | MVMoE-Light | 5.6260 | 2.1609% | 2.97s | 8.4094 | 1.9787% | 8.29s |
| | | MVMoE-Deeper | 5.6035 | 1.7363% | 7.05s | - | - | - |
| | | SHIELD-MoD | 5.5876 | 1.4523% | 4.70s | 8.3212 | 0.9074% | 15.16s |
| | | SHIELD | 5.5745 | 1.2165% | 5.24s | 8.3038 | 0.6926% | 16.92s |
| | OVRPB | OR-tools | 3.8078 | - | 1m11s | 5.6014 | - | 2m38s |
| | | POMO-MTVRP | 4.1457 | 8.8747% | 2.27s | 6.2396 | 11.4105% | 6.99s |
| | | MVMoE | 4.1234 | 8.2539% | 3.32s | 6.1759 | 10.2748% | 9.40s |
| | | MVMoE-Light | 4.1465 | 8.8749% | 3.03s | 6.2219 | 11.0947% | 8.70s |
| | | MVMoE-Deeper | 4.0969 | 7.5936% | 8.01s | - | - | - |
| | | SHIELD-MoD | 4.0743 | 6.9982% | 4.98s | 6.0640 | 8.2826% | 15.65s |
| | | SHIELD | 4.0687 | 6.8295% | 5.62s | 6.0129 | 7.3669% | 17.60s |
| | OVRPL | OR-tools | 4.3703 | - | 1m15s | 6.6913 | - | 2m50s |
| | | POMO-MTVRP | 4.5739 | 4.6592% | 2.38s | 7.0941 | 6.0842% | 7.85s |
| | | MVMoE | 4.5514 | 4.1338% | 3.45s | 7.0483 | 5.4115% | 10.77s |
| | | MVMoE-Light | 4.5653 | 4.4587% | 3.18s | 7.0665 | 5.6736% | 9.79s |
| | | MVMoE-Deeper | 4.5394 | 3.8684% | 9.01s | - | - | - |
| | | SHIELD-MoD | 4.5080 | 3.1491% | 5.24s | 6.9520 | 3.9647% | 17.40s |
| | | SHIELD | 4.4958 | 2.8730% | 5.62s | 6.9121 | 3.3698% | 19.82s |
| Out-task | VRPBL | OR-tools | 5.4775 | - | 1m13s | 8.2521 | - | 2m35s |
| | | POMO-MTVRP | 5.6231 | 2.6583% | 2.27s | 8.4291 | 2.2085% | 7.43s |
| | | MVMoE | 5.5861 | 2.0085% | 3.25s | 8.3967 | 1.8150% | 9.92s |
| | | MVMoE-Light | 5.5972 | 2.2190% | 3.06s | 8.4089 | 1.9562% | 8.92s |
| | | MVMoE-Deeper | 5.5920 | 2.0900% | 8.88s | - | - | - |
| | | SHIELD-MoD | 5.5587 | 1.5040% | 4.76s | 8.3250 | 0.9400% | 15.80s |
| | | SHIELD | 5.5482 | 1.3205% | 5.34s | 8.3091 | 0.7540% | 17.63s |
| | VRPBTW | OR-tools | 8.3979 | - | 1m21s | 14.4940 | - | 2m46s |
| | | POMO-MTVRP | 9.2380 | 10.0037% | 2.74s | 15.6453 | 8.1734% | 8.55s |
| | | MVMoE | 9.1706 | 9.3132% | 3.87s | 15.6160 | 7.9340% | 11.31s |
| | | MVMoE-Light | 9.1749 | 9.3735% | 3.64s | 15.6476 | 8.1580% | 11.25s |
| | | MVMoE-Deeper | 9.1749 | 9.2528% | 9.49s | - | - | - |
| | | SHIELD-MoD | 9.1328 | 8.8544% | 5.61s | 15.4787 | 7.0071% | 17.85s |
| | | SHIELD | 9.0873 | 8.3285% | 6.19s | 15.4095 | 6.5168% | 20.27s |
| | VRPLTW | OR-tools | 8.5328 | - | 1m23s | 14.5948 | - | 2m46s |
| | | POMO-MTVRP | 8.9175 | 4.5081% | 2.79s | 15.0812 | 3.5072% | 9.85s |
| | | MVMoE | 8.8754 | 4.0915% | 3.90s | 15.0478 | 3.2597% | 12.90s |
| | | MVMoE-Light | 8.8878 | 4.2350% | 3.76s | 15.0779 | 3.4699% | 12.45s |
| | | MVMoE-Deeper | 8.8705 | 3.9576% | 10.12s | - | - | - |
| | | SHIELD-MoD | 8.8237 | 3.4896% | 5.71s | 14.9107 | 2.3365% | 19.73s |
| | | SHIELD | 8.8021 | 3.2302% | 6.43s | 14.8641 | 2.0135% | 22.55s |

**FI10639 (Right block)**

| Problem | Solver | MTMDVRP50 Obj | Gap | Time | MTMDVRP100 Obj | Gap | Time |
|---|---|---|---|---|---|---|---|
| VRPL | OR-tools | 7.2655 | - | 1m11s | 11.0647 | - | 2m39 |
| | POMO-MTVRP | 7.3195 | 0.7427% | 2.44s | 10.9764 | -0.7391% | 8.01s |
| | MVMoE | 7.2732 | 0.1525% | 3.28s | 10.9476 | -0.9920% | 10.90s |
| | MVMoE-Light | 7.2799 | 0.2467% | 3.10s | 10.9619 | -0.8662% | 9.84s |
| | MVMoE-Deeper | 7.2567 | -0.0743% | 8.41s | - | - | - |
| | SHIELD-MoD | 7.2485 | -0.1881% | 5.03s | 10.8826 | -1.5856% | 17.38s |
| | SHIELD | 7.2411 | -0.2947% | 5.74s | 10.8762 | -1.6419% | 19.71s |
| VRPTW | OR-tools | 8.6076 | - | 1m21s | 13.8303 | - | 6m32s |
| | POMO-MTVRP | 8.9835 | 6.1814% | 2.92s | 14.9881 | 8.4109% | 9.10s |
| | MVMoE | 8.9383 | 5.6687% | 3.77s | 14.9514 | 8.1368% | 12.12s |
| | MVMoE-Light | 8.9575 | 5.8823% | 3.63s | 14.9753 | 8.3111% | 12.44s |
| | MVMoE-Deeper | 8.9071 | 5.2897% | 9.91s | - | - | - |
| | SHIELD-MoD | 8.8903 | 5.0787% | 5.70s | 14.8289 | 7.2670% | 18.93s |
| | SHIELD | 8.8706 | 4.8594% | 6.40s | 14.7707 | 6.8428% | 21.86s |
| OVRPTW | OR-tools | 5.5367 | - | 1m14s | 9.0269 | - | 2m44s |
| | POMO-MTVRP | 5.8542 | 5.7348% | 2.76s | 9.6772 | 7.2601% | 8.81s |
| | MVMoE | 5.8494 | 5.6378% | 3.81s | 9.6439 | 6.8831% | 11.85s |
| | MVMoE-Light | 5.8618 | 5.8692% | 3.57s | 9.6658 | 7.1204% | 11.19s |
| | MVMoE-Deeper | 5.8129 | 4.9966% | 10.41s | - | - | - |
| | SHIELD-MoD | 5.8005 | 4.7565% | 5.83s | 9.4988 | 5.2872% | 18.86s |
| | SHIELD | 5.7889 | 4.5616% | 6.55s | 9.4746 | 5.0130% | 21.82s |
| OVRPBL | OR-tools | 3.7943 | - | 1m6s | 5.6060 | - | 2m35s |
| | POMO-MTVRP | 4.1217 | 8.6277% | 2.37s | 6.2443 | 11.3928% | 7.36s |
| | MVMoE | 4.1042 | 8.1365% | 3.40s | 6.1755 | 10.1709% | 9.77s |
| | MVMoE-Light | 4.1345 | 8.9410% | 3.11s | 6.2269 | 11.0891% | 9.01s |
| | MVMoE-Deeper | 4.0782 | 7.4829% | 8.35s | - | - | - |
| | SHIELD-MoD | 4.6743 | 6.9563% | 5.04s | 6.0646 | 8.1956% | 16.03s |
| | SHIELD | 4.0511 | 6.7413% | 5.69s | 6.0163 | 7.3310% | 17.97s |
| OVRPBTW | OR-tools | 5.4856 | - | 1m14s | 9.0376 | - | 2m41s |
| | POMO-MTVRP | 6.0902 | 11.0224% | 2.75s | 10.1308 | 12.1891% | 8.48s |
| | MVMoE | 6.0783 | 10.7968% | 3.89s | 10.0940 | 11.7631% | 11.14s |
| | MVMoE-Light | 6.0859 | 10.9503% | 3.73s | 10.1191 | 12.0410% | 10.96s |
| | MVMoE-Deeper | 6.0537 | 10.3571% | 10.07s | - | - | - |
| | SHIELD-MoD | 6.0311 | 9.9335% | 5.84s | 9.9550 | 10.2390% | 18.01s |
| | SHIELD | 6.0170 | 9.7035% | 6.57s | 9.9070 | 9.7087% | 20.45s |
| OVRPLTW | OR-tools | 5.5178 | - | 1m16s | 9.0627 | - | 2m51s |
| | POMO-MTVRP | 5.8370 | 5.7846% | 2.81s | 9.7068 | 7.1746% | 9.21s |
| | MVMoE | 5.8256 | 5.5633% | 4.09s | 9.6680 | 6.7480% | 12.31s |
| | MVMoE-Light | 5.8413 | 5.8436% | 3.72s | 9.6930 | 7.0206% | 11.58s |
| | MVMoE-Deeper | 5.8108 | 5.3092% | 10.6s | - | - | - |
| | SHIELD-MoD | 5.7726 | 4.5990% | 5.97s | 9.5282 | 5.2071% | 19.32s |
| | SHIELD | 5.7654 | 4.4862% | 6.80s | 9.4963 | 4.8548% | 22.25s |
| VRPBLTW | OR-tools | 8.4892 | - | 1m23s | 14.3715 | - | 2m41s |
| | POMO-MTVRP | 9.3172 | 9.7539% | 2.94s | 15.5583 | 8.4426% | 9.20s |
| | MVMoE | 9.2484 | 9.0613% | 4.01s | 15.5064 | 8.0744% | 11.78s |
| | MVMoE-Light | 9.2586 | 9.1808% | 3.74s | 15.5511 | 8.3871% | 12.02s |
| | MVMoE-Deeper | 9.2484 | 8.9437% | 9.59s | - | - | - |
| | SHIELD-MoD | 9.2078 | 8.5868% | 5.72s | 15.3800 | 7.2121% | 18.51s |
| | SHIELD | 9.1680 | 8.0919% | 6.28s | 15.3116 | 6.7035% | 20.90s |
| OVRPBLTW | OR-tools | 5.4777 | - | 1m19s | 9.0148 | - | 2m33s |
| | POMO-MTVRP | 6.0851 | 11.0885% | 2.86s | 10.1094 | 12.2402% | 8.87s |
| | MVMoE | 6.0701 | 10.8055% | 4.02s | 10.0706 | 11.8008% | 11.52s |
| | MVMoE-L | 6.0786 | 10.9700% | 3.79s | 10.0989 | 12.1151% | 10.88s |
| | MVMoE-Deeper | 6.0498 | 10.4445% | 10.31s | - | - | - |
| | MVMoD | 6.0241 | 9.9658% | 5.94s | 9.9412 | 10.3847% | 18.39s |
| | Ours | 6.0146 | 9.8151% | 6.69s | 9.9001 | 9.9104% | 20.80s |

*Table 22.* Performance of models on GR9882

**GR9882 — Left half**

| Category | Problem | Solver | MTMDVRP50 Obj | Gap | Time | MTMDVRP100 Obj | Gap | Time |
|---|---|---|---|---|---|---|---|---|
| In-task | CVRP | HGS | 6.9560 | - | 1m 17s | 10.3936 | - | 2m 13s |
| | | POMO-MTVRP | 7.1084 | 2.1913% | 3.02s | 10.7649 | 3.6221% | 8.77s |
| | | MVMoE | 7.0647 | 1.5660% | 4.72s | 10.7410 | 3.3953% | 11.51s |
| | | MVMoE-Light | 7.0754 | 1.7233% | 3.89s | 10.7575 | 3.5564% | 10.66s |
| | | MVMoE-Deeper | 7.0566 | 1.4537% | 9.65s | - | - | - |
| | | SHIELD-MoD | 7.0445 | 1.2709% | 5.76s | 10.6632 | 2.6404% | 18.03s |
| | | SHIELD | 7.0360 | 1.1565% | 6.47s | 10.6622 | 2.6295% | 20.47s |
| | OVRP | OR-tools | 4.2741 | - | 1m 9s | 6.4873 | - | 2m 37s |
| | | POMO-MTVRP | 4.4856 | 4.9486% | 2.32s | 6.9236 | 6.8885% | 7.50s |
| | | MVMoE | 4.4670 | 4.5352% | 3.62s | 6.8612 | 5.9006% | 10.23s |
| | | MVMoE-Light | 4.4821 | 4.9079% | 3.06s | 6.8992 | 6.5072% | 9.43s |
| | | MVMoE-Deeper | 4.4342 | 3.7705% | 8.84s | - | - | - |
| | | SHIELD-MoD | 4.4165 | 3.3587% | 5.13s | 6.7528 | 4.2265% | 17.09s |
| | | SHIELD | 4.4039 | 3.0663% | 5.99s | 6.7109 | 3.5776% | 19.54s |
| | VRPB | OR-tools | 5.3878 | - | 1m 2s | 7.9488 | - | 2m 35s |
| | | POMO-MTVRP | 5.5305 | 2.6488% | 2.12s | 8.1316 | 2.3515% | 6.73s |
| | | MVMoE | 5.4960 | 2.0273% | 3.22s | 8.1031 | 1.9936% | 9.11s |
| | | MVMoE-Light | 5.5070 | 2.2479% | 3.05s | 8.1145 | 2.1470% | 8.28s |
| | | MVMoE-Deeper | 5.4825 | 1.7840% | 7.04s | - | - | - |
| | | SHIELD-MoD | 5.4659 | 1.4692% | 4.67s | 8.0045 | 0.7585% | 14.99s |
| | | SHIELD | 5.4560 | 1.2933% | 5.25s | 8.0003 | 0.7073% | 16.98s |
| | OVRPB | OR-tools | 3.6601 | - | 1m 13s | 5.3017 | - | 2m 40s |
| | | POMO-MTVRP | 3.9849 | 8.8728% | 2.29s | 5.9619 | 12.5598% | 7.09s |
| | | MVMoE | 3.9679 | 8.3625% | 3.31s | 5.8707 | 10.7826% | 9.42s |
| | | MVMoE-Light | 4.0022 | 9.3077% | 3.06s | 5.9508 | 12.3350% | 8.69s |
| | | MVMoE-Deeper | 3.9357 | 7.5287% | 8.04s | - | - | - |
| | | SHIELD-MoD | 3.9129 | 6.8585% | 4.94s | 5.7386 | 8.3071% | 15.66s |
| | | SHIELD | 3.9116 | 6.8401% | 5.61s | 5.7054 | 7.6825% | 17.64s |
| Out-task | OVRPL | OR-tools | 4.2759 | - | 1m 14s | 5.4665 | - | 2m 54s |
| | | POMO-MTVRP | 4.4924 | 5.0627% | 2.38s | 6.9109 | 7.0167% | 7.98s |
| | | MVMoE | 4.4725 | 4.6183% | 3.44s | 6.8517 | 6.0897% | 10.81s |
| | | MVMoE-Light | 4.4862 | 4.9660% | 3.21s | 6.8892 | 6.6897% | 9.81s |
| | | MVMoE-Deeper | 4.4528 | 4.1370% | 9.01s | - | - | - |
| | | SHIELD-MoD | 4.4226 | 3.4465% | 5.24s | 6.7470 | 4.4686% | 17.46s |
| | | SHIELD | 4.4093 | 3.1437% | 6.05s | 6.7033 | 3.7798% | 19.89s |
| | VRPBL | OR-tools | 5.4044 | - | 1m 13s | 7.9259 | - | 2m 46s |
| | | POMO-MTVRP | 5.5466 | 2.6310% | 2.27s | 8.0977 | 2.2328% | 7.44s |
| | | MVMoE | 5.5124 | 2.0251% | 3.26s | 8.0624 | 1.7844% | 9.60s |
| | | MVMoE-Light | 5.5290 | 2.3316% | 3.07s | 8.0825 | 2.0334% | 8.91s |
| | | MVMoE-Deeper | 5.5167 | 2.0785% | 9.01s | - | - | - |
| | | SHIELD-MoD | 5.4844 | 1.4986% | 5.58s | 7.9756 | 0.6815% | 15.70s |
| | | SHIELD | 5.4701 | 1.2376% | 6.16s | 7.9657 | 0.5610% | 17.65s |
| | VRPBTW | OR-tools | 8.5591 | - | 1m 23s | 14.7076 | - | 2m 42s |
| | | POMO-MTVRP | 9.3818 | 9.6117% | 2.77s | 15.8813 | 8.2210% | 8.55s |
| | | MVMoE | 9.3229 | 9.0631% | 3.88s | 15.8378 | 7.8885% | 11.38s |
| | | MVMoE-Light | 9.3409 | 9.2382% | 3.63s | 15.8744 | 8.1585% | 10.80s |
| | | MVMoE-Deeper | 9.3261 | 8.9607% | 9.47s | - | - | - |
| | | SHIELD-MoD | 9.2801 | 8.5275% | 5.58s | 15.6726 | 6.7998% | 17.77s |
| | | SHIELD | 9.2497 | 8.1686% | 6.16s | 15.6150 | 6.3862% | 20.07s |
| | VRPLTW | OR-tools | 8.7717 | - | 1m 24s | 14.6818 | - | 2m 49s |
| | | POMO-MTVRP | 9.1521 | 4.3371% | 2.81s | 15.1587 | 3.4803% | 9.87s |
| | | MVMoE | 9.1039 | 3.8812% | 3.93s | 15.1196 | 3.1988% | 13.51s |
| | | MVMoE-Light | 9.1157 | 4.0234% | 3.73s | 15.1365 | 3.3103% | 12.05s |
| | | MVMoE-Deeper | 9.0936 | 3.6702% | 10.08s | - | - | - |
| | | SHIELD-MoD | 9.0525 | 3.2890% | 5.73s | 14.9741 | 2.2352% | 19.73s |
| | | SHIELD | 9.0343 | 3.0922% | 6.40s | 14.9220 | 1.8556% | 22.28s |

**GR9882 — Right half**

| Problem | Solver | MTMDVRP50 Obj | Gap | Time | MTMDVRP100 Obj | Gap | Time |
|---|---|---|---|---|---|---|---|
| VRPL | OR-tools | 7.0566 | - | 1m 8s | 10.9621 | - | 2m 39s |
| | POMO-MTVRP | 7.1025 | 0.6507% | 2.35s | 10.8673 | -0.8707% | 8.00s |
| | MVMoE | 7.0583 | 0.0504% | 3.42s | 10.8343 | -1.1747% | 10.86s |
| | MVMoE-Light | 7.0674 | 0.1778% | 3.13s | 10.8499 | -1.0253% | 9.87s |
| | MVMoE-Deeper | 7.0458 | -0.1276% | 8.4s | - | - | - |
| | SHIELD-MoD | 7.0342 | -0.2962% | 5.00s | 10.7588 | -1.8647% | 17.28s |
| | SHIELD | 7.0267 | -0.4024% | 5.73s | 10.7533 | -1.9215% | 19.75s |
| VRPTW | OR-tools | 8.7191 | - | 1m 18s | 14.1579 | - | 6m 33s |
| | POMO-MTVRP | 9.0838 | 6.0783% | 2.81s | 15.3650 | 8.6007% | 9.14s |
| | MVMoE | 9.0412 | 5.5405% | 3.81s | 15.3199 | 8.2577% | 12.73s |
| | MVMoE-Light | 9.0580 | 5.7197% | 3.56s | 15.3400 | 8.3956% | 11.40s |
| | MVMoE-Deeper | 9.0011 | 5.0772% | 9.93s | - | - | - |
| | SHIELD-MoD | 8.9955 | 4.9831% | 5.68s | 15.1639 | 7.1856% | 18.96s |
| | SHIELD | 8.9728 | 4.7571% | 6.39s | 15.1141 | 6.8234% | 21.64s |
| OVRPTW | OR-tools | 5.3713 | - | 1m 14s | 8.7285 | - | 2m 43s |
| | POMO-MTVRP | 5.6981 | 6.0840% | 2.74s | 9.3763 | 7.5389% | 8.85s |
| | MVMoE | 5.6898 | 5.9100% | 3.86s | 9.3344 | 7.0460% | 11.74s |
| | MVMoE-Light | 5.7075 | 6.2320% | 3.58s | 9.3682 | 7.4403% | 10.87s |
| | MVMoE-Deeper | 5.6383 | 4.9754% | 10.44s | - | - | - |
| | SHIELD-MoD | 5.6333 | 4.8490% | 5.81s | 9.1843 | 5.3396% | 18.81s |
| | SHIELD | 5.6179 | 4.5997% | 6.65s | 9.1565 | 5.0259% | 21.63s |
| OVRPBL | OR-tools | 3.6489 | - | 1m 6s | 5.3628 | - | 2m 35s |
| | POMO-MTVRP | 3.9788 | 9.0419% | 2.37s | 6.0290 | 12.5224% | 7.41s |
| | MVMoE | 3.9540 | 8.3113% | 3.42s | 5.9425 | 10.8406% | 9.80s |
| | MVMoE-Light | 3.9894 | 9.3004% | 3.12s | 6.0122 | 12.1885% | 9.05s |
| | MVMoE-Deeper | 3.9219 | 7.4804% | 8.33s | - | - | - |
| | SHIELD-MoD | 3.9083 | 7.1102% | 5.02s | 5.7983 | 8.1741% | 16.03s |
| | SHIELD | 3.9036 | 6.9255% | 5.69s | 5.7719 | 7.6776% | 18.07s |
| OVRPBTW | OR-tools | 5.3443 | - | 1m 15s | 8.7357 | - | 2m 38s |
| | POMO-MTVRP | 5.9343 | 11.0402% | 2.79s | 9.8206 | 12.5485% | 8.55s |
| | MVMoE | 5.9228 | 10.7895% | 3.87s | 9.7818 | 12.0651% | 11.11s |
| | MVMoE-Light | 5.9307 | 10.9458% | 3.76s | 9.8090 | 12.3796% | 10.40s |
| | MVMoE-Deeper | 5.8931 | 10.2686% | 10.09s | - | - | - |
| | SHIELD-MoD | 5.8669 | 9.7525% | 5.86s | 9.6345 | 10.4030% | 18.04s |
| | SHIELD | 5.8538 | 9.5448% | 6.57s | 9.5922 | 9.9189% | 20.40s |
| OVRPLTW | OR-tools | 5.4180 | - | 1m 17s | 8.7467 | - | 2m 50s |
| | POMO-MTVRP | 5.7484 | 6.0986% | 2.83s | 9.3965 | 7.5557% | 9.20s |
| | MVMoE | 5.7388 | 5.9032% | 4.01s | 9.3546 | 7.0644% | 12.16s |
| | MVMoE-Light | 5.7578 | 6.2420% | 3.74s | 9.3821 | 7.3816% | 11.28s |
| | MVMoE-Deeper | 5.7069 | 5.3321% | 10.62s | - | - | - |
| | SHIELD-MoD | 5.6815 | 4.8271% | 5.97s | 9.2118 | 5.4452% | 19.31s |
| | SHIELD | 5.6673 | 4.6124% | 6.81s | 9.1826 | 5.1003% | 22.02s |
| VRPBLTW | OR-tools | 8.5652 | - | 1m 22s | 14.8707 | - | 2m 43s |
| | POMO-MTVRP | 9.4066 | 9.8231% | 2.94s | 16.0036 | 0.0008% | 9.19s |
| | MVMoE | 9.3398 | 9.1785% | 3.92s | 15.9774 | 7.7261% | 12.09s |
| | MVMoE-Light | 9.3566 | 9.3505% | 3.66s | 16.0050 | 7.9140% | 11.49s |
| | MVMoE-Deeper | 9.3414 | 9.0618% | 9.64s | - | - | - |
| | SHIELD-MoD | 9.3011 | 8.7148% | 5.69s | 15.8097 | 6.6217% | 18.45s |
| | SHIELD | 9.2614 | 8.2500% | 6.30s | 15.7598 | 6.2376% | 20.62s |
| OVRPBLTW | OR-tools | 5.3472 | - | 1m 18s | 8.7637 | - | 2m 35s |
| | POMO-MTVRP | 5.9479 | 11.2345% | 2.90s | 9.8478 | 12.4518% | 8.93s |
| | MVMoE | 5.9443 | 11.0920% | 4.00s | 9.8019 | 11.9205% | 11.47s |
| | MVMoE-L | 5.9446 | 11.1131% | 3.84s | 9.8301 | 12.2460% | 10.77s |
| | MVMoE-Deeper | 5.9096 | 10.5184% | 10.35s | - | - | - |
| | MVMoD | 5.8912 | 10.0965% | 5.97s | 9.6544 | 10.2485% | 18.44s |
| | Ours | 5.8696 | 9.7509% | 6.69s | 9.6219 | 9.8720% | 20.74s |

