# OpenReview forum: "SHIELD: Multi-task Multi-distribution Vehicle Routing Solver with Sparsity and Hierarchy"
_ICML.cc/2025/Conference — ICML 2025 poster_

### Official Review · Reviewer_SpW1 · 2025-03-11

**Overall Recommendation:** 3

**Summary:**

This paper proposes a foundation model for vehical route problem with multi-task and multi-distribution. The model contains the mixture-of-depth decoder, which dynamically selects nodes at each decoding step, thus improving the efficiency and generalization ability of the model. A context-based clustering layer is proposed for modeling the spatial hierarchy of different cities. Extensive experiments on 9 real-world maps with 16 VRP variants show the effectiveness of the proposed method.

**Claims And Evidence:**

Yes

**Essential References Not Discussed:**

In terms of related work, it would be better to discuss route planning methods such as:

Graph2route: A dynamic spatial-temporal graph neural network for pick-up and delivery route prediction
DRL4Route: A Deep Reinforcement Learning Framework for Pick-up and Delivery Route Prediction
which are both learning-based solutions for solving the routing problem

**Experimental Designs Or Analyses:**

The experiment design is sound, and the results are convincing.

**Methods And Evaluation Criteria:**

Yes

**Other Comments Or Suggestions:**

please see the strengths and weakness

**Other Strengths And Weaknesses:**

As for strengths, this paper solves a practical problem and is well-written. In terms of the methodology perspective, the idea of "using the amount of compute to potentially serve as a regularization for the model" is quite interesting and seems to help the model save the computation and generalize well at inference. The extensive experiments also show the effectiveness of the method.

However, in terms of the drawbacks,  each city in this paper is only associated with (x, y)  for its location. However, in the real world, each city can have various spatio-temporal attributes. How can we effectively deal with that by the proposed model?

**Questions For Authors:**

Is there any experiment to show how much computational resources can be saved at the inference time?

Is there any scale-law observed at inference time by the proposed decoder?

How would beta in the decoder influence the performance?

**Relation To Broader Scientific Literature:**

The  contribution of this paper is related to two research communities:  (1) the first is VRP-related research, which aims to solve routing problems with different constraints. (2) the second is foundation model-related research, which aims to build general models that can solve multiple tasks in a zero-shot way.

**Theoretical Claims:**

I have gone through most of the theoretical claims, and all of them are correct.

---

> ### Author Rebuttal · Authors · 2025-04-01
>
> We thank the reviewer for acknowledging that our work addresses a practical problem that interests the community and that the extensive experiments are convincing. We hope our response adequately addresses the remaining questions.
>
> **[W1: Handling Spatial-Temporal characteristics]**: We thank the reviewer for recommending the discussion of Graph2Route [1] and DRL4Route [2]. Both works have presented realistic scenarios of spatial-temporal dynamics of routing problems, and we will add their discussion in the final version. The MTVRP and MTMDVRP scenarios, in their current forms, are focused more on fixed locations governed by spatial coordinates since they reflect the majority of standard vehicle routing problems for effective multi-task model development. In order to handle more complex temporal distributions, it is possible to adapt similar encoding architectures shared in Graph2Route and DRL4Route. Both works discuss the evolution of edge features (such as travel time) as the problem is progressively solved, and present encoding methods such as GRU to capture such nuances. Indeed, this adds realism to the problem scenario, which is an important and exciting future work to expand the capability of SHIELD towards foundation optimization models for VRPs. At the same time, SHIELD works on problems with multiple constraints such as time-windows (e.g. VRPTW, VRPLTW etc), suggesting that it can handle some form of temporal characteristics as well.
>
> Alternatively, orthogonal works, such as [3], utilize POMO-styled architectures to encode the time-dependent VRP, where travel times on edges change as the solution construction progresses. This suggests that it is highly possible to integrate such encoding architectures into SHIELD to encode temporal variations in the data.
>
> **[Q1: Compute saved during inference]**: Thank you for the question! To answer this, we compare MVMoE-Deeper with SHIELD, as both contain the same number of decoder layers, where SHIELD only allows 10% of tokens while MVMoE-Deeper allows 100% for processing. Table 4 in Appendix F shows the number of parameters, floating operations (FLOPs) and runtime of the models. **From the results, SHIELD has almost 10 GFLOPs less and is 30% faster than MVMoE-Deeper during inference**, albeit with slightly more parameters. Additionally, **MVMoE-Deeper uses ~45Gb of memory, while SHIELD only uses ~6Gb during inference, reducing 87% of memory costs**. Finally, MVMoE-Deeper is untrainable on the MTMDVRP100, whereas SHIELD is able due to its sparse property in the decoder. We will clarify the computational savings more explicitly in the revised paper.
>
> **[Q2: Scaling laws]**: We agree that exploring the scaling laws of learned models would yield valuable insights for the community. However, we would like to clarify that this work focuses on developing more generalizable multitask models, which still remains at an early stage toward building foundation neural combinatorial optimization (NCO) models. As such, investigating scaling laws meaningfully would require training across larger datasets and model scales (e.g., 0.1B, 1B, 10B, 100B parameters), which is beyond the scope of this paper and the time constraints of the rebuttal, as such training demands substantial GPU resources over weeks. We will explore this direction by training models with varying sizes and data availability, and include further discussion in the revised paper. Our work represents a concrete step toward architectures that may underpin future foundation NCO models.
>
> Nevertheless, we opt to discuss the intuition of scaling laws for the inference stage of a trained model. For NCO solvers, we can allocate more test time to perform sampling and find better solutions during inference. Due to time constraints, we reduce the number of test instances to 100 instances per problem and performed inference with sampling widths 1x, 10x, 50x, and 100x. We plot the performance of the various widths [here](https://imgur.com/a/Dwo4EbE). As shown, as we increase the sampling width, the general performance of the model increases (lower gap is better) in a logarithmic fashion. This suggests that while we can allocate more test time for inference, its effectiveness eventually saturates.
>
> **[Q3: Influence of $\beta$]**: Sorry for the confusion. $\beta$ controls the number of tokens (or nodes) that is processed by a MoD layer. In Table 2 of the main paper, we highlight how varying $\beta$ influences the performance of the solver. Essentially, increasing the number of tokens processed improves the in-task in-distribution performance at the expense of **generalization**.
>
> [1] Graph2Route: A Dynamic Spatial-Temporal Graph Neural Network for Pick-up and Delivery Route Prediction. SIGKDD, 2022
>
> [2] DRL4Route: A Deep Reinforcement Learning Framework for Pick-up and Delivery Route Prediction. SIGKDD, 2023
>
> [3] SED2AM: Solving Multi-Trip Time-Dependent Vehicle Routing Problem using Deep Reinforcement Learning. TKDD, 2025

---

### Official Review · Reviewer_yZkq · 2025-03-13

**Overall Recommendation:** 3

**Summary:**

This paper proposes a novel problem, the Multi-Task Multi-Distribution Vehicle Routing Problem (MTMDVRP), which is an extension of the traditional Multi-Task Vehicle Routing Problem (MTVRP). The problem focuses on different node distributions of different geographical regions in the real world, further considering the generalizability. For this problem, the paper introduces SHIELD, a novel neural combinatorial optimization solver, which leverages sparsity and hierarchy principles through the Mixture-of-Depths (MoD) technique and a context-based clustering layer to improve efficiency and generalization. Experiments demonstrate that SHIELD outperforms existing methods across 9 real-world maps and 16 VRP variants, showing strong generalization capabilities, especially in cross-task and cross-distribution settings.

## update after rebuttal

In the rebuttal, authors have addressed most of my concerns with experiments. I believe this is a promising work to solve MTMDVRP. I would like to keep my postive rating. It would be beneficial if the code could be further made public.

**Claims And Evidence:**

The main claims made in the paper are supported by extensive experiments. The authors demonstrate the superior performance of the SHIELD model across 9 real-world maps and 16 VRP variants, showing its effectiveness in multi-task and multi-distribution scenarios. The experimental results highlight SHIELD's strengths in terms of optimization objectives (e.g., tour length) and generalization capabilities (e.g., cross-task and cross-distribution performance). Additionally, the ablation studies provide insights into the contributions of the sparsity and hierarchy designs.

**Essential References Not Discussed:**

The paper has correctly cited the relevant prior work.

**Experimental Designs Or Analyses:**

I have checked the validity of the experiment for SHIELD. The experiments are divided into in-task, out-task, in-distribution, and out-distribution categories. This setting allows for a comprehensive evaluation of the model's generalization capabilities.

**Methods And Evaluation Criteria:**

The proposed methods and evaluation criteria in the paper are largely reasonable and closely related to the research problem. The MTMDVRP problem aims to further consider the complexity of node distributions in the real world. The authors selected a range of tasks and distributions for training, and then tested on previously unseen tasks and distributions, achieving better performance than other baseline methods. This effectively demonstrates the generalization capability of the solver.

**Other Comments Or Suggestions:**

No other comments.

**Other Strengths And Weaknesses:**

Strengths:

S1. The MTMDVRP setting proposed is a novel and practical extension of the MTVRP, addressing the limitations of uniform distribution assumptions in prior works. This makes the problem formulation more relevant to real-world applications.

S2. The SHIELD model incorporates Mixture-of-Depths (MoD) and context-based clustering layers, which effectively balance computational efficiency and generalization.

S3. The authors conduct thorough experiments including in-distribution, out-distribution, in-task, out-task settings, demonstrating the model's generalization capability.

Weaknesses:

W1. The use of real-world maps, though diverse, primarily focuses on national-scale distributions. The inclusion of more granular urban or local distributions could provide a more comprehensive evaluation of the model’s adaptability. The paper mentions that the observed distribution differences are due to the company’s business expansion. However, the realistic scenario is better captured by the expansion of business across cities.

W2. Lack of comparison with non-neural solvers. This omission makes it difficult to assess how SHIELD performs relative to well-established non-neural methods.

W3. The code is not released, which may affect the reproducibility of this work.

**Questions For Authors:**

Q1. Can SHIELD scale to very large problem instances (e.g., >1000 nodes)?

Q2. The article tests SHIELD on 16 VRP variants, but some real-world problems may involve more complex constraints (e.g., multi-depot, heterogeneous fleets, stochastic demands). Can SHIELD handle such scenarios, or are there inherent limitations?

**Relation To Broader Scientific Literature:**

The paper expands current the MTVRP problem to the MDMTVRP problem. As part of the problem, the multi-task learning (MTL) has been an active research area in machine learning and deep learning, particularly in improving model generalization and reducing overfitting. Also, the Mixture-of-Depths (MoD) technique in the paper has also been widely used in other domains to improve generalization and reduce model complexity.

**Theoretical Claims:**

The paper only contains one theorem related to the VC (Vapnik-Chervonenkis) dimension, which is widely acknowledged. The idea derived from this theorem is relatively intuitive.

---

> ### Author Rebuttal · Authors · 2025-04-01
>
> We thank the reviewer for their positive recognition of the work's novelty, effectiveness, practical values, and thorough experimental validation. We hope our responses with new experiments address the remaining concerns.
>
> **[W1: Realistic Setup]**: While we present a national-level business expansion scenario, the distributional variations studied are also representative of city-level patterns. The national-level data serves as a realistic proxy to show SHIELD’s generalization ability. Our model and training process are agnostic to specific distributions. In practice, a company can train SHIELD on data from existing cities and apply it to new ones, where distributional shifts may occur. Given SHIELD’s flexibility in handling such variations, it is well-suited for the scenario noted by the reviewer.
>
> To illustrate our point that the proposed architecture is inherently generic, we refer the reviewer to Table 1 of the main paper and Table 11 of Appendix O. For Table 11, we trained the model on the Uniform distribution (the MTVRP scenario). The results show that even in a different distribution than the MTMDVRP, SHIELD presents itself as a superior model.
>
> **[W2: Comparison to Non-neural Solvers]**: Sorry for the confusion, yes, the results are benchmarked with classic solvers. Similar to [3][4], we solved the test instances with known solvers in HGS (for CVRP and VRPTW) and Google's OR-tools (for the rest). This allows us to compute the optimality gap of each solver compared to the best-performed non-neural solvers (run with reasonably longer solving times typical in industrial applications). Please see updated Table 1 [here](https://imgur.com/a/ihDUR0v) that includes the solver's performance across tasks and distributions.
>
> **[W3: Public Code]**: We plan to release our code and data upon acceptance of this paper after potential intellectual property review by the employer of our authors. Nevertheless, we have provided detailed descriptions of our architecture in Appendix I to support reproducibility.
>
> **[Q1: Scaling to large instances]**: Thank you for the question. The primary contribution of this paper is the introduction of a novel MTMDVRP setup and the SHIELD model, which significantly improves performance on small-scale instances. This serves as a foundational step toward future research on scaling to larger problem sizes. For example, several existing techniques orthogonal to our contributions can be integrated to enhance the scalability, such as self-improvement learning [1], and divide-and-conquer strategies (e.g. UDC[2]), all requiring a backbone neural solver for small-scale optimization.
>
> Nevertheless, we have evaluated SHIELD and the baselines on the CVRPLib Set-X containing CVRP instances of larger sizes. These instances range from 101 nodes to 1001 nodes. Tables 7 and 8 in Appendix L showcases their results. In general, SHIELD exhibits stronger generalization capabilities compared to the others.
>
> Additionally, we generated and labelled MTMDVRP200 datasets and performed inference using the trained MTMDVRP100 models. The results below illustrate that our approach is significantly more robust when generalizing beyond the trained problem size.
>
> |||MTMDVRP200||||
> |:-:|:-:|:-:|:-:|:-:|:-:|
> |||In-dist||Out-dist||
> ||Model|Obj|Gap|Obj|Gap|
> |In-task|POMO-MTVRP|14.5695|5.4613%|15.9036|7.0430%|
> ||MVMoE|14.6137|5.8753%|15.9391|7.3486%|
> ||MVMoE-Light|14.6420|6.0924%|15.9581|7.4784%|
> ||SHIELD-MoD|14.4123|4.7980%|15.7342|6.1487%|
> ||SHIELD|14.3648|3.7939%|15.6536|5.0516%|
> |Out-task| POMO-MTVRP|15.5735|8.5203%|17.1759|10.2531%|
> ||MVMoE|15.6040|8.8840%|17.2145|10.5085%|
> ||MVMoE-Light|15.6412|9.1470%|17.2423|10.7143%|
> ||SHIELD-MoD|15.5373|7.4336%|17.1948|8.8987%|
> ||SHIELD|15.3896|6.4856%|16.9555|7.8179%|
>
> **[Q2: Complex constraints]**: In this work, we extend generalization beyond the 16 VRP variants (from [3] and [4]) to include distributional shifts, building upon the POMO framework. The additional constraints, as suggested by the reviewer, can also be solved by POMO-styled neural solvers, such as multi-depot [5] and heteogeneous fleet [6]. This is orthogonal to our focus and represent valuable directions for future work where we can train SHEILD on even more tasks in MTMDVRP setup. To our knowledge, SHEILD is the first to generalize across both task and distribution **simultaneously**.
>
> [1] Boosting Neural Combinatorial Optimization for Large-Scale Vehicle Routing Problems. ICLR, 2025
>
> [2] UDC: A unified neural divide-and-conquer framework for large-scale combinatorial optimization problems. NeurIPS, 2024
>
> [3] Multi-task learning for routing problem with cross-problem zero-shot generalization. SIGKDD, 2024
>
> [4] Mvmoe: Multi-task vehicle routing solver with mixture-of-experts. ICML, 2024
>
> [5] Multi-type attention for solving multi-depot vehicle routing problems. ITS, 2024
>
> [6] Deep Reinforcement Learning for Solving the Heterogeneous Capacitated Vehicle Routing Problem. Cybernetics, 2021

---

### Official Review · Reviewer_Q6ps · 2025-03-19

**Overall Recommendation:** 3

**Summary:**

This paper introduces SHIELD, a framework with sparsity and hierarchy principles to address MTMDVRP problem.

**Claims And Evidence:**

Yes

**Essential References Not Discussed:**

No

**Experimental Designs Or Analyses:**

Yes

**Methods And Evaluation Criteria:**

Yes

**Other Comments Or Suggestions:**

See above.

**Other Strengths And Weaknesses:**

Strengths:
1. This paper is well-written and easy to follow.
2. The experimental results are solid.
3. The integration of MoD to reduce computation overhead is interesting.

Questions:
1. Why does the proposed framework utilize a MoE encoder?
2. Why the authors choose to use MoD to reduce computation overhead instead of other techniques such as linear attention.

**Questions For Authors:**

See above.

**Relation To Broader Scientific Literature:**

This paper advances the Multi-Task VRP (MTVRP) setting to the more realistic yet challenging Multi-Task MultiDistribution VRP (MTMDVRP) setting.

**Theoretical Claims:**

Yes

---

> ### Author Rebuttal · Authors · 2025-04-01
>
> We thank the reviewer for the positive comments and for recognizing our paper as solid and easy to follow, with an interesting use of MoD to reduce computational overhead. We hope our responses with new results address the remaining questions.
>
> **[Q1: Why MoE Encoder]**: Insightful question! In this paper, we find that MoD is more beneficial in the decoder as opposed to the encoder. Meanwhile, an existing work [3] (which only studies MTVRP instead of MTMDVRP in this paper) provides evidence that MoE is more effective in the encoder than in the decoder. **We believe this aligns with the distinct functional roles of the encoder and decoder in NCO models:**
> * In MTMDVRP, **the encoder** processes diverse multi-task contexts and learns meaningful representations from various task contexts which feature combinations of constraints. For example, CVRPTW combines capacity and time window constraints, while CVRPBLTW further adds backhaul and linehaul constraints. MoE is well-suited for the encoder as it leverages specialized expert subnetworks to handle the shared and combinatorial patterns in the input data.
> * In contrast, **the decoder** in MTMDVRP is focused on sequential solution construction with adaptive computation. While some node selections are straightforward, others require finer granularity and greater computational/reasoning capacity -- especially when dealing with clustered distributions or complex constraint-distribution interactions in MTMDVRP. Thus, dynamic control over depth and computation is essential. MoD naturally addresses this need by adaptively allocating resources across decoder layers.
> * **Together, their synergy enhances the model's ability to capture context-dependent, adaptive fine-grained decisions for MTMDVRP.**
>
> To verify our claims, experiments in Table 6 of Appendix J investigate the impact of MoD in the encoder. Even though we double the number of MoD encoder layers, the network is unable to learn effective representations for the problems.
>
> **[Q2: Why not Linear Attention]**: We agree with the reviewer that there are multiple alternate approaches for reducing the computational costs, such as linear attention [1]. It is an important direction for future work where such sparse attention methods could further improve the scalability of our SHIELD model. However, to the best of our knowledge, such approaches have not yet been shown to be effective for learning multi-task NCO solvers. This may be because, while sparse attention may work for simpler VRPs (e.g., TSP, CVRP), it can struggle with more complex variants (e.g., OVRPBLTW), where capturing complex dependencies and constraints with a simplified attention mechanism remains nontrivial, particularly in our MTMDVRP setting.
>
> To further support our claims, **we add new results by comparing with a recent model, INViT [2], that employs sparse attentions**, extending it from simple VRP variants to the more complex MTMDVRP setting. Essentially, INViT proposes to update embeddings by only paying attention to the current node's k-Nearest Neighbors (k-NN). Such a scheme is similar to ours, where the number of interactions amongst the nodes is reduced during decoding. **However, a key difference is that in INViT, the reduction is based on a heuristic -- the k-NN nodes, while in SHIELD, we opt to learn which nodes to focus on based on MoD**. We train INViT on our dataset and settings; our results are shown in the following table. Here, we see that SHIELD outperforms INVIT. One main reason is that the sparsity in INViT arises from selecting the k-NN nodes based on spatial coordinates, which is potentially unsuitable for MTMDVRP settings. Thus, such an approach prunes possibly important nodes to interact with, restricting the model's capabilities. In contrast, SHIELD offers two key advantages for MTMDVRP: 1) We reduce computational overhead by focusing on a smaller number of nodes; 2) We **learn** to prioritize task-relevant nodes for decision-making. These features make SHIELD significantly stronger and more generalizable.
>
> |||MTMDVRP50||||||MTMDVRP100||||||
> |:-:|:-:|:-:|:-:|:-:|:-:|:-:|:-:|:-:|:-:|:-:|:-:|:-:|:-:|
> |||In-dist|||Out-dist|||In-dist|||Out-dist|||
> ||Model|Obj|Gap|Time|Obj|Gap|Time|Obj|Gap|Time|Obj|Gap|Time|
> |In-task|INVIT|6.4082|9.1437%|66.48s|6.7462|9.0992%|66.84s|10.6057|17.2425%|66.65s|11.4286|18.4235%|68.06s|
> ||SHIELD|6.0136|2.3747%|6.13s|6.2784|2.7376%|6.11s|9.2743|2.4397%|19.93s|9.9501|3.1638%|20.25s|
> |Out-task|INVIT|6.2996|15.3570%|69.43s|6.6932|15.2064%|70.11s|11.1489|26.8217%|68.00s|12.1012|27.9947%|69.98s|
> ||SHIELD|5.7779|6.0810%|6.20s|6.1570|6.3520%|6.20s|9.2400|5.6104%|19.92s|9.9867|6.2727%|20.18s|
>
> [1] Linear attention is (maybe) all you need (to understand transformer optimization). ICLR, 2024
>
> [2] INViT: A generalizable routing problem solver with invariant nested view transformer. ICML, 2024
>
> [3] Mvmoe: Multi-task vehicle routing solver with mixture-of-experts. ICML, 2024

---

> > ### Comment · Reviewer_Q6ps · 2025-04-07
> >
> > Thanks for the rebuttal of the authors. I have no further questions from my side, and the given explanations help me learn something about this field.

---

> > > ### Author Response · Authors · 2025-04-08
> > >
> > > Dear Reviewer Q6ps,
> > >
> > > We thank you for the acknowledgement and your positive support of our work.
> > >
> > >
> > > Best Regards,
> > >
> > > Authors of 8616

---

### Official Review · Reviewer_7L8M · 2025-03-20

**Overall Recommendation:** 3

**Summary:**

This paper introduces the Multi-Task Multi-Distribution Vehicle Routing Problem (MTMDVRP), an extension of the MTVRP. The MTMDVRP effectively captures the complexities inherent in real-world industrial applications by incorporating various realistic customer distributions. To address these challenges, the authors propose a neural solver, SHIELD, which integrates soft clustering, Mixture of Experts, and Mixture-of-Depths (MoD). The authors further conducted experiment on 9 real-world maps with 16 VRP variants each.

**Claims And Evidence:**

The paper's claim is supported by their numerical experiments, in both time and suboptimality gap. Although I'm not an expert in VRP problems, it seems that MTMDVRP is hard to solve since multi-distributions are considered.

**Essential References Not Discussed:**

N/A

**Experimental Designs Or Analyses:**

See Methods And Evaluation Criteria.

**Methods And Evaluation Criteria:**

The authors proposed the innovative network learning architecture SHIELD, introduced the clustering layer to enhance the hierarchical expression ability of the model, and added the MoD layer in the decoding to take into account the sparsity, which is impressive in the field of machine learning to solve combinatorial optimization problems.

However, the introduction of MoD and soft clustering are meant to enhance generalization; but in this paper, the generalization capabilities for larger-scale problems have not yet been tested. All experiments are conducted on relatively small data, namely 50 nodes and 100 nodes problems. How the model behave on larger data (>= 200) should be tested.

**Other Comments Or Suggestions:**

N/A

**Other Strengths And Weaknesses:**

Strength:
1. By combining MoE and MoD, and soft clustering, the proposed neural solver SHIELD outperform existing methods across a range of VRP variant tasks.
2. The authors conducted solid experiments that involve nearly all modules, including MoD, MoE, and soft clustering, with detailed descriptions of the experimental procedures and results.


Weakness:
Although the paper studied a difficult question, I feel the contribution of adapting clustering nodes/MoE/MoD is limited, as it has already been thoroughly explored in previous works, therefore it's barely innovative.

**Questions For Authors:**

1. Is the proposed method SHIELD designed for the scenario of multi-task multi-distribution? What about its performance on the scenario of multi-distribution?

**Relation To Broader Scientific Literature:**

VRP is closely related to operation research. That said, MTMDVRP seems to be a novel problem, though I can hardly evaluate how important it is to OR community.

**Theoretical Claims:**

The theorem in the main paper was excerpted from Theorem 2.3 in Goldberg & Jerrum, 1993. Although I didn't check the book, I'm prone to believe it's correct.

---

> ### Author Rebuttal · Authors · 2025-04-01
>
> We would like to thank the reviewer for the positive review and recognizing the depth of experiments done to show the benefits of the proposed architecture. We hope our following responses will further address the reviewer's concerns about the work.
>
> **[W1: Contributions of Clustering Nodes/MoE/MoD]**: While we agree that this paper is not for proposing new general clustering/MoE/MoD techniques, **our work represents the first to reveal and verify their unique synergy towards learning foundation models for neural combinatorial optimization (NCO)**, which holds significant value in both academia and industrial applications.
>
> Specifically, inspired from the VC-dimension perspective, we first pose a key research question for neural VRP solvers: can generalizable multi-task solver be learned by regularizing the model through (1) dynamic compute allocation and (2) parameter size control? We thus bring the MoD and the adaptive clustering approaches to this field which respectively regulates these two aspects. To our knowledge, we are the first to explore these techniques in NCO, showing **insights to the community on how dynamic node selection per decoder layer and adaptive clustering enhance both efficiency and generalization**.
>
> Moreover, our contribution extends to **the introduction and study of the more realistic multi-task, multi-distribution VRP setting (MTMDVRP)**, which opens new directions and bridges the gap to real-world applications. Up till recently, the NCO community has been focused on *single-task* solvers on the *uniform* distribution. The two most recent works in [1] and [2] first addressed the possibility of constructing generalized multi-task models (but still on the *uniform* distribution). This is similar to how traditional solvers behave - the same solver can be used to solve multiple different tasks by introducing various constraints. A neural version of such a solver is highly important and intriguing to the NCO and OR community, as they are capable of extremely fast problem solving by exploiting modern architecture, and the learning aspect provides possible advantages by exploiting underlying structure. Pushing further toward practical relevance, our work is the first to evaluate multi-task neural solvers under cross-distribution generalization by introducing the MTMDVRP setup, which better reflects real-world settings (see Table 9 in Appendix M). We further demonstrate that our proposed SHIELD architecture enables the learning of **robust** foundational neural solvers for solving the introduced MTMDVRP.
>
> **[W2: Generalization to Larger Instances]**: Thanks for suggesting the evaluation on larger problem sizes. We evaluated all models on the CVRPLib Set-X, which contains CVRP instances ranging from **101 nodes to 1001 nodes**. The results can be found in Tables 7 and 8 of Appendix L. Additionally, we would like to direct the reviewer to our response to Reviewer yZkq regarding scaling to larger instances. We performed additional experiments of MTMDVRP200 using our trained MTMDVRP100 models. Both results show that SHIELD exhibits stronger generalization capabilities than the other baselines.
>
> **[Q1: Performance on MDVRP]**: We thank the reviewer for the insightful observation on whether the observation of SHIELD's performance is unique to the MTMDVRP scenario. **We observe in our experiments that in both the Multi-Task VRP (MTVRP) and Multi-Distribution (MDVRP) case, SHIELD is still the clear leader.** For MTVRP, Table 11 in Appendix O highlights the case where the underlying distribution is Uniform in nature, similar to the works done in [1] and [2]. From Table 11, it is clear that SHIELD has a sizable advantage over its counterparts, especially so when *generalization* across tasks is required. In the MTVRP100 case, it has a large gap of ~1.2% over the current known state-of-the-art MVMoE.
>
> Additionally, Table 9 in Appendix M showcases the **importance** of having varied distributions. Here, we trained all models on the MTVRP scenario (meaning we only draw data from the Uniform distribution) but apply them to the same test set in the MTMDVRP scenario (USA, JA, BM are considered "in-distribution", while the rest are considered "out-distribution"). We see that the model's performance on the varied distributions have **degraded** as compared to Table 1 in the main paper, highlighting the **importance** of exposing the models to varied distributions in training.
>
> As for MDVRP, Table 10 in Appendix N displays our experiments for this scenario. In this case, we fix the task to CVRP but retain the various distributions. We find that SHIELD still shows a sizable performance improvement over its fellow benchmark models. This exemplifies that our approach is not specific to only the MTMDVRP scenario.
>
> [1] Multi-task learning for routing problem with cross-problem zero-shot generalization. SIGKDD, 2024
>
> [2] Mvmoe: Multi-task vehicle routing solver with mixture-of-experts. ICML, 2024

---

### Decision · Program_Chairs · 2025-05-01

**Decision:**

Accept (poster)

**Comment:**

The paper presents SHIELD, a novel neural combinatorial optimization solver for the Multi-Task Multi-Distribution Vehicle Routing Problem.

SHIELD incorporates sparsity and hierarchy principles, effectively balancing computational efficiency and generalization.
The model is evaluated on 9 real-world maps with 16 VRP variants each, showing strong generalization. However, the paper lacks a comparison with non-neural solvers, making it difficult to assess SHIELD's performance relative to well-established methods.

Despite the cons, the reviewers acknowledge the novelty of the problem setting, the effectiveness of the SHIELD model, and the extensive experimental validation.  The authors also provide rebuttals addressing the reviewer concerns and providing additional experimental results.

Considering the overall assessment and the authors' efforts to address the concerns, the decision is a weak accept.